# Formation of polarity convergences underlying shoot outgrowths

Katie Abley[†], Susanna Sauret-Güeto[‡], Athanasius FM Marée, Enrico Coen*

John Innes Centre, Norwich Research Park, Norwich, United Kingdom

**Abstract** The development of outgrowths from plant shoots depends on formation of epidermal sites of cell polarity convergence with high intracellular auxin at their centre. A parsimonious model for generation of convergence sites is that cell polarity for the auxin transporter PIN1 orients up auxin gradients, as this spontaneously generates convergent alignments. Here we test predictions of this and other models for the patterns of auxin biosynthesis and import. Live imaging of outgrowths from *kanadi1 kanadi2 Arabidopsis* mutant leaves shows that they arise by formation of PIN1 convergence sites within a proximodistal polarity field. PIN1 polarities are oriented away from regions of high auxin biosynthesis enzyme expression, and towards regions of high auxin importer expression. Both expression patterns are required for normal outgrowth emergence, and may form part of a common module underlying shoot outgrowths. These findings are more consistent with models that spontaneously generate tandem rather than convergent alignments.

*For correspondence: enrico.coen@jic.ac.uk

Present address: [†]The Sainsbury Labratory, University of Cambridge, Cambridge, United Kingdom; [‡]Department of Plant Sciences, University of Cambridge, Cambridge, United Kingdom

Competing interests: The authors declare that no competing interests exist.

## Introduction

The development of plant shoots involves iterative formation of outgrowths. Shoot apical meristems produce leaf primordia, which in turn provide the setting for the initiation of new outgrowths such as serrations and leaflets. A common developmental module has been proposed to underlie the generation of both leaves and leaf-derived outgrowths (*Barkoulas et al., 2008*; *Hay et al., 2006*). A key feature of the module is an epidermal site of high intracellular auxin, located at the centre of convergence of the polarised auxin efflux carrier, PIN1 (*Barkoulas et al., 2008*; *Benková et al., 2003*; *Hay et al., 2006*; *Reinhardt et al., 2000*, *2003*; *Scarpella et al., 2006*).

The generation of polarity convergence sites has been most commonly explained by the up-the-gradient model, whereby cells localise PIN1 towards the neighbouring cell with the highest concentration of intracellular auxin (*Bilsborough et al., 2011*; *Jönsson et al., 2006*; *Smith et al., 2006*). This mechanism is parsimonious because it can spontaneously generate spaced centres of polarity convergence without additional assumptions about the presence of auxin sources or sinks. Molecular mechanisms accounting for up-the-gradient behaviour have been proposed to involve detection of auxin-induced stress gradients (*Heisler et al., 2010*) or auxin transport sensing (*Cieslak et al., 2015*).

An alternative model for formation of sites of polarity convergences assumes that PIN1 becomes localised to cell membranes in proportion to the rate of auxin efflux across the membrane (*Mitchison, 1980*; *Rolland-Lagan and Prusinkiewicz, 2005*). A possible molecular mechanism for how cells might sense auxin flux has recently been proposed (*Cieslak et al., 2015*). Although originally proposed to account for venation patterns, this flux-based model has also been shown to be compatible with the patterns of epidermal PIN1 polarity in the vegetative shoot apical meristem (*Stoma et al., 2008*). Polarity convergence formation can be accounted for by assuming that sub-epidermal provascular PIN1 strands are induced in regions of elevated auxin and cause a local depletion of auxin from the epidermis. This depletion causes PIN1 polarities to reorient and generate a site of convergence, which then raises auxin levels at its centre through transport. A problem with this model is

**eLife digest** Plants, unlike animals, are able to grow and develop throughout their lives. New leaves and flowers are made from outgrowths that constantly form at the tip of growing shoots. Groups of cells in the outer layer of the shoot tip arrange a protein called PIN1 so that it is more abundant on the cell surfaces that face towards the centre of the group. PIN1 transports a hormone called auxin out of plant cells and this "convergent" arrangement of PIN1 increases the levels of auxin in cells at the centre of the group, leading to the formation of a new outgrowth. However, it is not clear what causes these cells to position their PIN1 proteins in this way.

Several hypotheses have been proposed to explain how convergent patterns of PIN1 form. For example, according to the "up-the-gradient" hypothesis, PIN1 is allocated to the end of a cell that is next to a cell with a higher level of auxin. Abley et al. have now compared predictions from computer models with new experimental data from a plant called *Arabidopsis* to evaluate three hypotheses for how convergent PIN1 patterns form. A computer model based on the up-the-gradient hypothesis naturally creates convergent PIN1 patterns, even if each cell starts off with the same level of auxin. On the other hand, models based on two other hypotheses generate tandem alignments of PIN1 so that auxin is transported in the same direction along lines of cells.

Next, Abley et al. tested these models using mutant *Arabidopsis* plants that develop outgrowths from the lower surface of their leaves. These outgrowths form in a similar way to outgrowths at the growing shoot tip, but in a simpler context. The experiments show that the patterns of where auxin is produced in growing leaves were more compatible with the tandem alignment models than the up-the-gradient model. This suggests that plants use a tandem alignment mechanism to form convergences of PIN1 proteins that generate the local increases in auxin needed to make new outgrowths.

This study only examined a single layer of cells on the plant surface. Other cell layers also show highly organised patterns of PIN1 proteins, so a future challenge is to extend the approach to study the entire 3D structure of new shoot outgrowths.

that it predicts a transient drop in intracellular auxin concentration during early stages of convergence formation, which is not supported experimentally (*Brunoud et al., 2012*; *Heisler et al., 2005*).

A further type of model for coordinating PIN1 orientations requires neither responding to auxin gradients between cells nor sensing auxin flux. Instead, the indirect coupling model involves intracellular polarity components that can establish cell polarity without external asymmetries in auxin distribution (*Abley et al., 2013*). The two polarity components each exist in two forms: a diffusible cytoplasmic form (A and B) and a more slowly diffusing membrane bound form (A* and B*). Interactions between the components leads to A* and B* being localised at opposite ends of the cell. PIN1 is recruited to the membrane by one of the components (A*), causing a polarised PIN1 distribution. Polarities of neighbouring cells are coupled indirectly through a feedback mechanism in which extracellular auxin inhibits A* and thus PIN1 accumulation. This model results in polarities being oriented away from regions of high extracellular auxin and towards regions with low extracellular auxin. It can generate coordinated polarities for a field of cells, but it is unclear whether it can generate centres of PIN1 convergence.

A convenient system for testing the models is the formation of ectopic 3D outgrowths from the abaxial leaf surface of *kanadi1kanadi2 (kan1kan2)* mutants (*Eshed et al., 2004*). These outgrowths can be considered as intermediates between leaf primordia and serrations. Similar to leaf primordia, the outgrowths emerge perpendicular to the main plane of the tissue, and like serrations, they are derived from the leaf. Because of their emergence from the abaxial lamina which can be readily imaged, these outgrowths are more amenable to time-lapse imaging than serrations which are often obscured by neighbouring cotyledon tissue and curving of the leaf edge. *kan1kan2* outgrowths have elevated intracellular auxin at their tips, and their formation depends on specific patterns of auxin biosynthetic enzyme expression (*Wang et al., 2011*). They provide a test bed for studying convergence site formation in a starting context which, unlike the apex, is not complicated by the prior

patterns of continual primordium initiation. However, the dynamics of auxin accumulation, PIN1 polarity and expression of auxin biosynthesis genes at early stages of *kan1kan2* outgrowth emergence have not been described. Moreover, the role of *CUC* genes, needed for formation of primordia and serrations (*Nikovics et al., 2006*, *Aida et al., 1997*) has not been determined.

We show through time-lapse imaging that, similar to leaf primordia and serrations, *kan1kan2* outgrowths are preceded by centres of PIN1 polarity convergence with elevated intracellular auxin. These convergent polarities arise within the context of an initial proximo-distal PIN1 polarity field and are promoted by *CUC2* activity. An exploration of model behaviours reveals that models may be classified into two types: those that spontaneously generate convergent aligments (e.g. the up-the-gradient) and those that spontaneously generate tandem alignments (e.g. flux-based and indirect coupling). Both types of model can account for the generation of an initial proximo-distal polarity field, followed by the formation of convergences with elevated intracellular auxin. However, unlike the convergent alignment model, tandem alignment models require the appearance of local regions with elevated auxin import. In support of tandem alignment models, we show that expression of the *LAX1* auxin importer is elevated in regions of polarity convergence at the tips of *kan1-kan2* outgrowths, and that *AUX/LAX* importer genes are required for normal *kan1kan2* outgrowth development. Additionally, we show that expression of YUCCA1 and YUCCA4 auxin biosynthetic enzymes tends to be elevated in regions of polarity divergence in *kan1kan2* and WT leaves, an observation which is also most readily consistent with tandem alignment models. Thus, tandem alignment models provide parsimonious explanations for the developmental module underlying outgrowth emergence.

## Results

### Formation of polarity convergence centres in the context of a proximodistal polarity field

To characterise patterns of epidermal PIN polarity associated with the development of *kan1kan2* outgrowths, a *PIN1::PIN1:GFP* reporter (*Benková et al., 2003*) was imaged on the abaxial surface of wild-type (WT) and *kan1kan2* leaves. Here we focus on PIN1 since this is the predominant epidermally expressed PIN that shows polar intracellular distributions in wild-type leaves (*Guenot et al., 2012*). In young *kan1kan2* leaf primordia, the epidermal PIN1 polarity pattern was similar to that observed in WT: PIN1 polarities were oriented towards the leaf tip in both cases (*Figure 1A,B*). At later stages of development, WT leaves retained a proximo-distal polarity pattern, and then no-longer showed detectable expression of the *PIN1::PIN1:GFP* reporter (*Figure 2A*). The loss of *PIN1::PIN1:GFP* expression did not occur uniformly throughout the WT leaf because some patches of cells retained expression for longer than other regions of the leaf (*Figure 2A iii*). By contrast to WT, *kan1-kan2* leaves maintained PIN1 expression until later stages of development and formed centres of PIN1 polarity convergence, located at the tips of emerging ectopic outgrowths (*Figure 2B iv and C iv*). Cells at the centres of convergence had highly polarised PIN1, oriented towards the interface between three or four neighbouring cells (*Figure 2C iv*). To determine whether the centres of convergence preceded outgrowth emergence, leaves were imaged over several days and cell lineages that gave rise to convergences traced back through the time-course of the experiment (*Figure 2C*, yellow dots, arrows and lines). This revealed that cells closest to the centre of convergence at the outgrowth tip descended from one or two cells, which were already at a centre of polarity convergence prior to outgrowth emergence (*Figure 2B,C*). Taken together, these findings show that, prior to *kan1kan2* outgrowth emergence, centres of polarity convergence develop within a proximodistally oriented polarity field.

To investigate how the pattern of PIN1 polarity is related to the dynamics of intracellular auxin accumulation, a *DR5::GFP* reporter (*Benková et al., 2003*) was imaged in *kan1kan2* leaves over the course of outgrowth development. At early stages, *DR5::GFP* signal was detected exclusively at the leaf tip (*Figure 3A*), similar to the pattern described for the WT leaf (*Mattsson et al., 2003*; *Scarpella et al., 2006*). Approximately one day prior to the first observation of an outgrowth, locally elevated *DR5::GFP* signal was detected in groups of epidermal cells in the proximal region of the lamina, where outgrowths typically form (*Figure 3B i*, yellow arrow). At later stages, these regions of *DR5::GFP* expression persisted, and could be found throughout emerging outgrowths (*Figure 3C,D*,

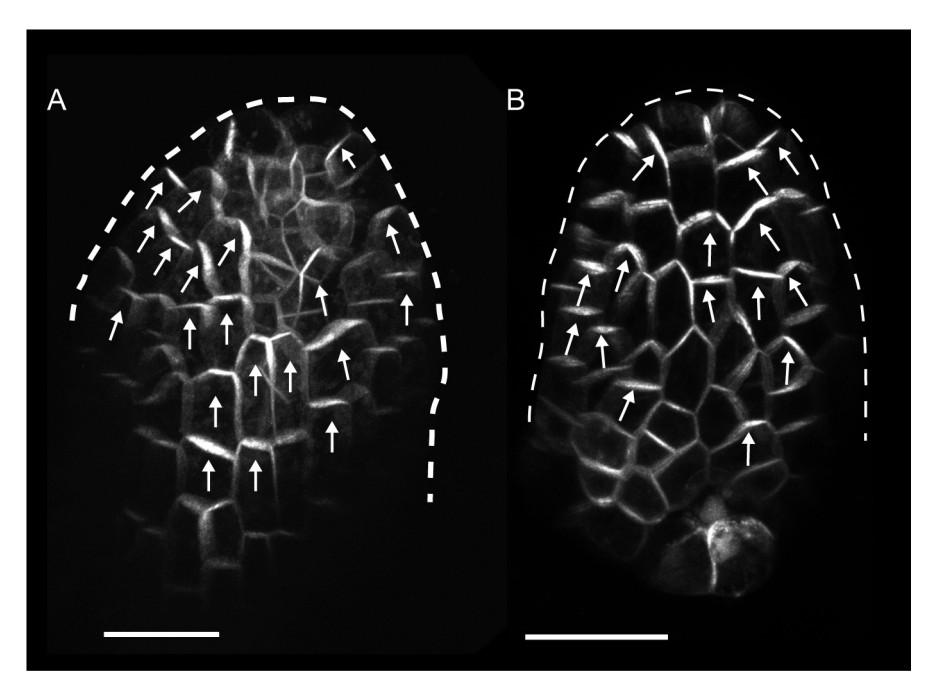

**Figure 1.** *PIN1::PIN1:GFP* expression in young WT and *kan1kan2* leaf primordia. (**A**) WT primordium of leaf 1, showing abaxial epidermis (a total of 10 leaves were imaged over 2 separate experiments). (**B**) As for **A**, but for a *kan1kan2* primordium (a total of 15 leaves were imaged over three separate experiments). White arrows indicate inferred polarities. Dashed white lines indicate leaf outlines. Scale bars = 20 μm.

yellow arrows). Thus, both intracellular auxin activity maxima and centres of PIN1 polarity convergence precede and predict sites of outgrowth emergence in *kan1kan2* leaves.

## CUC2 is required for normal *kan1kan2* outgrowth development and PIN1 convergence formation

*cuc2* mutants fail to form centres of PIN1 polarity convergence in the leaf margin and do not develop serrations (*Bilsborough et al., 2011*; *Nikovics et al., 2006*). To investigate whether, like leaf serrations, *kan1kan2* outgrowths depend on the CUC2 transcription factor, we generated *kan1-kan2cuc2* mutants. Leaves of *kan1kan2* mutants consistently produce ectopic, finger-like outgrowths from the abaxial leaf surface (97% of 160 leaves observed had at least one outgrowth, and the mean number of outgrowths per leaf was 12.4) (*Figure 4A,B*). By contrast, mature leaves of *kan1kan2cuc2* mutants only occasionally developed such outgrowths (only 8% of leaves observed developed outgrowths, and the mean number of outgrowths was 1.3) (*Figure 4C,D*). Some *kan1kan2cuc2* leaves also developed ridge-like thickenings of the abaxial surface or serrations in the leaf margin (*Figure 4—figure supplement 1A,B*).

To test if *CUC2* is required for the formation of epidermal sites of PIN1 convergence, we generated *kan1kan2cuc2* plants with the *PIN1::PIN1:GFP* reporter. Time-lapse imaging of PIN1::PIN1:GFP in the abaxial epidermis of the first two leaves of this mutant revealed that centres of convergence did not form (*Figure 5*). At early stages of leaf development, a proximodistal PIN1 polarity field was observed (*Figure 5 i*), but similar to WT leaves, at later stages the expression of *PIN1::PIN1:GFP* was lost throughout most of the abaxial epidermis (*Figure 5B i–iii*). Expression of the reporter was maintained in groups of epidermal cells which did not show centres of PIN1 convergence (*Figure 5B ii*).

In upper leaves of *kan1kan2cuc2* mutants (leaves 3 to 6), ectopic primordium-like bumps were occasionally observed on the abaxial leaf surface (in approximately 10% of leaves observed, n = 50) (*Figure 5C*). These bumps were associated with centres of PIN1 polarity convergence linked with sub-epidermal PIN1 strands. The frequency of centres of PIN1 convergence in *kan1kan2cuc2*

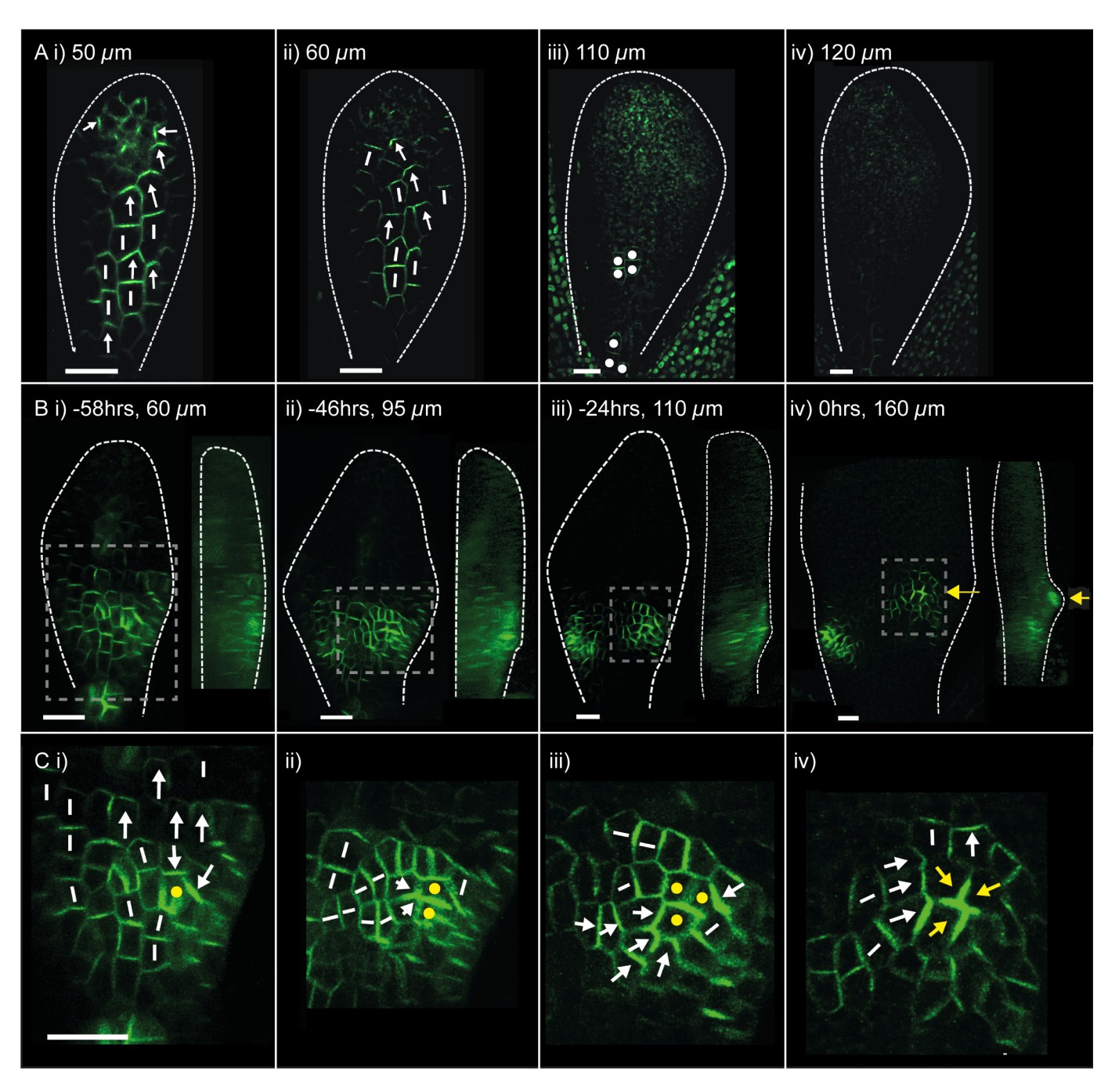

**Figure 2.** PIN1::PIN1:GFP polarity patterns in WT and *kan1kan2* leaf development. Confocal images of the *PIN1::PIN1:GFP* marker in the abaxial epidermis of the same WT leaf primordium imaged over a period of 2 days. Approximate leaf widths (measured from projections of the z stacks) are given above each image. Times from the beginning of the experiment are: i) 0 hrs, ii) 12 hrs, iii) 36 hrs, iv) 48 hrs. Cells indicated by white dots in iii) retain detectable expression of the marker for longer than other cells. This time-lapse data is supported by snapshot images taken for at least five leaves at each developmental stage in a separate experiment. (B) Confocal images of the *PIN1::PIN1:GFP* marker in the abaxial epidermis of a *kan1kan2* leaf primordium, prior to and during the emergence of an ectopic outgrowth. For each time point, a surface view of the abaxial epidermis (left) and a side view of a 3D rendering of the confocal z-stack (right) are shown. The side views allow the emergence of the outgrowth to be monitored. The time relative to when an outgrowth could clearly be observed and the estimated leaf width are given above each image. Yellow arrows in iv) indicate an ectopic centre of PIN1 polarity convergence at the tip of an emerging outgrowth. (C) Magnified images of the regions outlined by the dashed grey rectangles in B, showing the development of the centre of PIN1 polarity convergence at the tip of the emerging outgrowth. Yellow dots, arrows and lines indicate the cells that form the centre of convergence in iv). Arrows indicate inferred PIN1 polarities and lines indicate inferred axes of

*Figure 2 continued on next page*

*Figure 2 continued*

PIN1 distributions. Scale bars = 20 µm. The scale bar in C i) applies to all panels in C. This data is representative of tracking three out of three *kan1kan2* leaves that developed ectopic outgrowths, each in a separate experiment, and with snapshot images from at least three leaves at each developmental stage (in another separate experiment).

mutants was much lower than in the *kan1kan2* background where approximately 95% of leaves 3–6 were observed to have at least one centre of PIN1 polarity convergence (n = 50). Thus, formation of *kan1kan2* outgrowths and polarity convergence sites is largely dependent on *CUC2,* although this requirement can occasionally be circumvented.

## *CUC2* is ectopically expressed during PIN1 convergence formation in *kan1kan2* leaves

To gain insight into how *CUC2* might promote the formation of epidermal centres of PIN1 convergence, we characterised the expression pattern of *CUC2* at the time when centres of PIN1 polarity convergence form in *kan1kan2* leaves. The *CUC2::RFPer* reporter (which causes expression of endoplasmic reticulum-localised RFP) was time-lapse imaged together with *PIN1::PIN1:GFP* in the abaxial epidermis of the first two *kan1kan2* leaves. At early stages of leaf development, when PIN1 polarities are oriented distally, elevated expression of *CUC2::RFPer* was detected throughout the proximal half of the lamina (*Figure 6A,B*). Before outgrowth emergence, expression of the reporter was lost from cells towards the base of the leaf, and from early centres of PIN1 polarity convergence (*Figure 6C ii,D ii*). This resulted in groups of cells within the proximal half of the lamina with *CUC2::RFPer* expression, surrounded by non-*CUC2::RFPer* expressing cells. Later on these groups of *CUC2::RFP* expressing cells were seen to be at the distal side of each epidermal PIN1 polarity convergence (*Figure 6D iii*) and at the distal base of emerging outgrowths (*Figure 6D iii*, side view, blue arrow head). This region is the boundary between the outgrowth and the abaxial lamina and is analogous to the sinus region of serrations where *CUC2* is also expressed (*Nikovics et al., 2006*; *Bilsborough et al., 2011*).

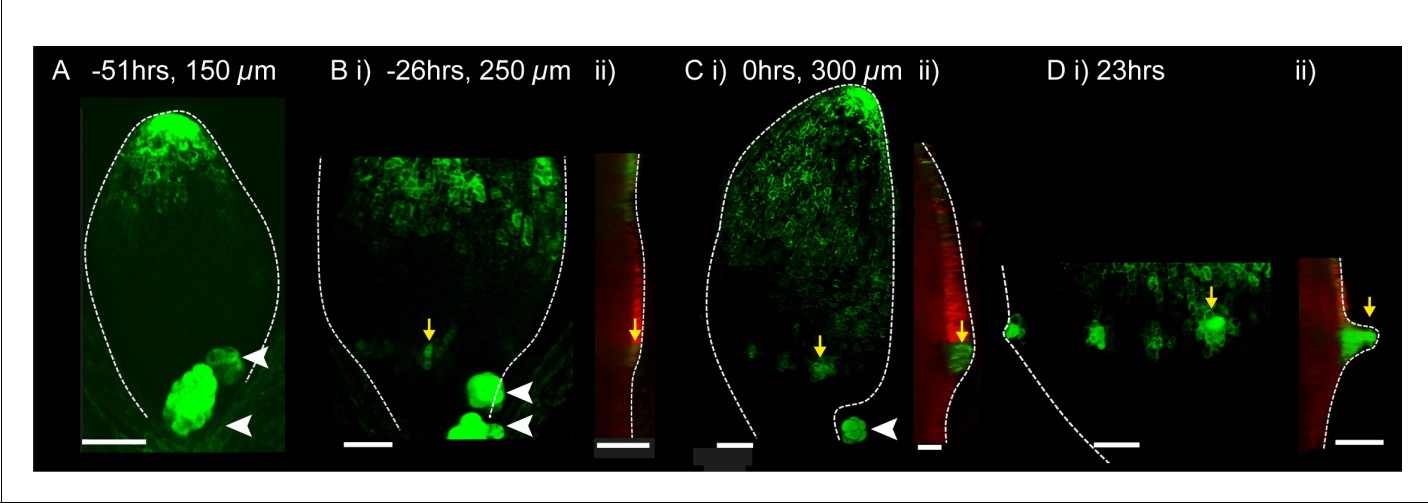

**Figure 3.** *DR5::GFP* expression in a *kan1kan2* leaf during outgrowth development. Confocal images of *DR5::GFP* in the same *kan1kan2* leaf imaged over a period of 3 days. Times relative to the first observation of an outgrowth, and leaf widths, are given above images. B ii), C ii) and D ii) show optical sections through 3D renderings of confocal z-stacks. *DR5::GFP* (green) and auto-fluorescence plus *CUC2::RFP* (red) channels are shown (*CUC2::RFP* is used to help show the leaf outline). Yellow arrows in D indicate the region of *DR5::GFP* activity at the tip of an outgrowth and yellow arrows in B and C indicate the same *DR5::GFP* expressing cells tracked back in time prior to outgrowth emergence. White arrow heads indicate high *DR5::GFP* signal in stipules. Scale bars = 50 µm. This data is representative of tracking 4 out of 4 *kan1kan2* leaves that developed outgrowths (in two experiments), and of snapshot data taken for at least 3 leaves at each of the developmental stages shown (in another separate experiment).

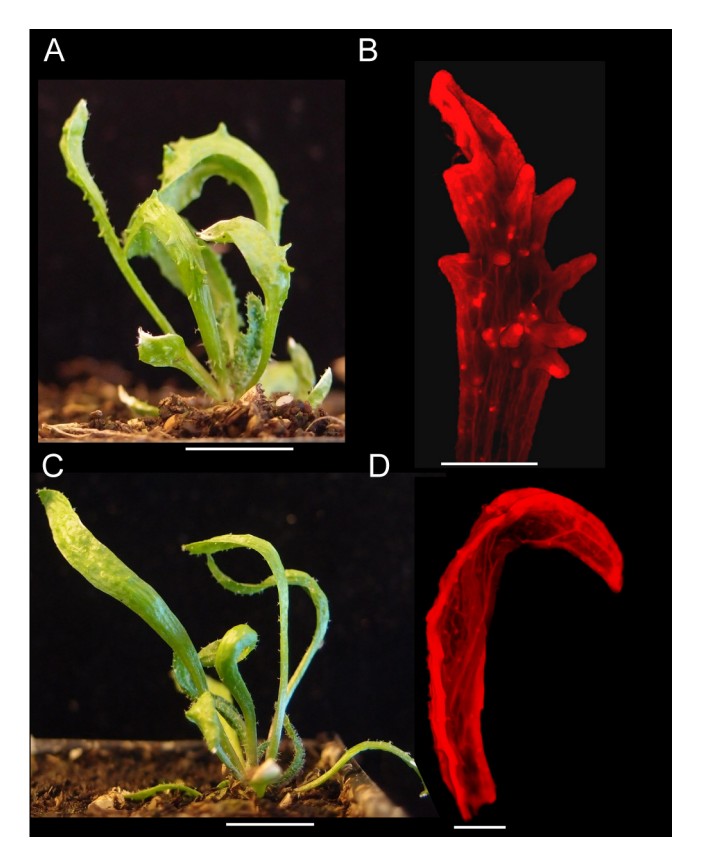

**Figure 4.** Leaf phenotype of the *kan1kan2cuc2* mutant. (**A**) Whole *kan1kan2* plant. (**B**) OPT images of a *kan1kan2* leaf showing abaxial leaf surface. (**C**) Whole *kan1kan2cuc2* plant. (**D**) OPT images of a *kan1kan2cuc2* leaf. Scale bars in **A** and **C** = 1 cm, scale bars in **B** and **D** = 1 mm. See *Figure 4—figure supplement 1* for more details of the *kan1kan2cuc2* phenotype.

The following figure supplement is available for figure 4:

**Figure supplement 1.** Abaxial ridges and serrations produced in *kan1kan2cuc2* mutants.

We next investigated how the expression pattern of *CUC2::RFPer* is related to the polarity of PIN1 in emerging outgrowths. Inspection of PIN1 polarities using the *PIN1::PIN1:GFP* reporter revealed that in developing outgrowths, cells close to the regions of elevated *CUC2::RFPer* expression frequently had PIN1 polarities oriented away the regions of high *CUC2* expression (*Figure 6E, F*). On the proximal side of the *CUC2::RFPer* expression domain (i.e. in cells of the emerging outgrowth), PIN1 polarities were oriented proximally, away from cells with high *CUC2::RFP* expression and towards a centre of convergence at the outgrowth tip; while on the distal side of the *CUC2::RFPer* expression domain, polarity appeared to point distally.

In contrast to *kan1kan2* leaves, in WT, *CUC2::RFPer* expression was mainly restricted to the leaf base and to regions of the margin associated with centres of PIN1 convergence involved in serration development (*Figure 6G*; *Bilsborough et al., 2011*).

Thus, ectopic outgrowth development in *kan1kan2* leaves is associated with the ectopic expression of *CUC2* in the abaxial lamina. Similar to leaf serrations, the normal development of *kan1kan2* outgrowths and the centres of PIN1 polarity convergence that precede them is dependent on *CUC2*. We next compare different models of PIN1 polarity for their ability to capture the observed PIN1 polarity patterns and for their consistency with the role of *CUC2* in PIN1 polarity convergence formation.

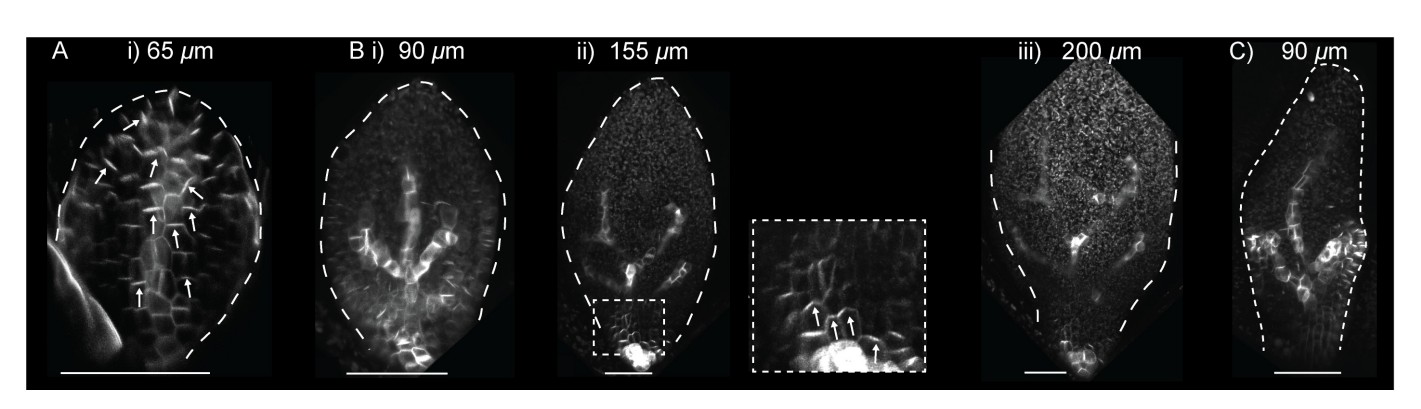

**Figure 5.** *PIN1::PIN1:GFP* in leaf one of the *kan1kan2cuc2* mutant. (**A**) Confocal image of *PIN1::PIN1:GFP* in the abaxial epidermis of the first leaf primordium of a *kan1kan2cuc2* mutant. (**B**) Time-lapse confocal images of the abaxial side of the first leaf of a *kan1kan2cuc2* mutant, taken at successive time points. B**i**, **ii** and **iii** are maximum intensity projections of the abaxial side of the leaf, and therefore signal from both epidermal and sub-epidermal cell layers is combined. The data set shown here is representative of that obtained by tracking three *kan1kan2cuc2PIN1::PIN1:GFP* leaves (from three different seedlings), and of snapshot images of 10 *kan1kan2cuc2* first leaf primordia taken at each of the developmental stages shown. (**C**) Maximum intensity projection of the abaxial side of leaf 5 of a *kan1kan2cuc2* mutant, showing an example of epidermal centres of PIN1 convergence (on left and right sides of the leaf). Scale bars = 50 $\mu$m.

## PIN1 polarity models can be classified into two groups with different behaviours

To test possible mechanisms that could underlie the observed epidermal polarity patterns, we compare the abilities of up-the-gradient, flux-based and indirect coupling models to account for them. We first characterise basic model behaviours, and then investigate the assumptions needed for each model to capture the observed polarity patterns and how they may be tested experimentally.

The flux and up-the-gradient models are implemented as in previous publications, using the simplifying assumption that auxin moves directly from cell to cell and omitting an explicit representation of the cell wall (*Bilsborough et al., 2011*; *Feugier et al., 2005*; *Jönsson et al., 2006*; *Rolland-Lagan and Prusinkiewicz, 2005*; *Smith et al., 2006*; *Stoma et al., 2008*). By contrast, we explicitly represent the cell wall in our implementations of the indirect coupling model. In this model, we also represent cell membranes and walls with several compartments per cell edge (allowing lateral diffusion of components to be simulated). However, as in previous work, flux and up-the-gradient models are implemented with a single compartment per edge and no lateral diffusion of components along the cell edge (*Bilsborough et al., 2011*; *Feugier et al., 2005*; *Jönsson et al., 2006*; *Rolland-Lagan, 2008*; *Smith et al., 2006*; *Stoma et al., 2008*). With the flux-based model, we assume a linear feedback between flux and PIN1 allocation, which tends to generate broad regions of coordinated polarisation rather than narrow canalised strands that arise with super-linear feedback (*Feugier et al., 2005*; *Stoma et al., 2008*). Details of all the model assumptions are given in the Materials and methods.

Previously simulations of up-the-gradient and flux-based models have considered polarity patterns in fields of cells all of which have the potential to polarise (*Jönsson et al., 2006*; *Rolland-Lagan, 2008*; *Smith et al., 2006*; *Stoma et al., 2008*). However, to get a clearer view of the properties of each model, we begin by considering a group of cells having initially uniform auxin concentration and PIN1 distribution, and with only the central cell having the ability to relocate its PIN1 according to the rules specified by each model (*Figure 7A–C*). We find that the central cell becomes polarised with both the up-the-gradient and flux-based models, assuming that there are small random fluctuations (noise) in the initial concentrations of PIN1 in the cell membranes and that auxin flux from the central cell can modify the initially homogeneous auxin concentration in the neighbouring tissue (*Figure 7A and B*). In the up-the-gradient model, a cell edge with elevated PIN1 causes an increase in auxin concentration in the neighbouring cell. This elevated auxin concentration in the neighbour feeds back to cause an increase in PIN1 allocation to the cell edge (*Figure 7A ii*, purple

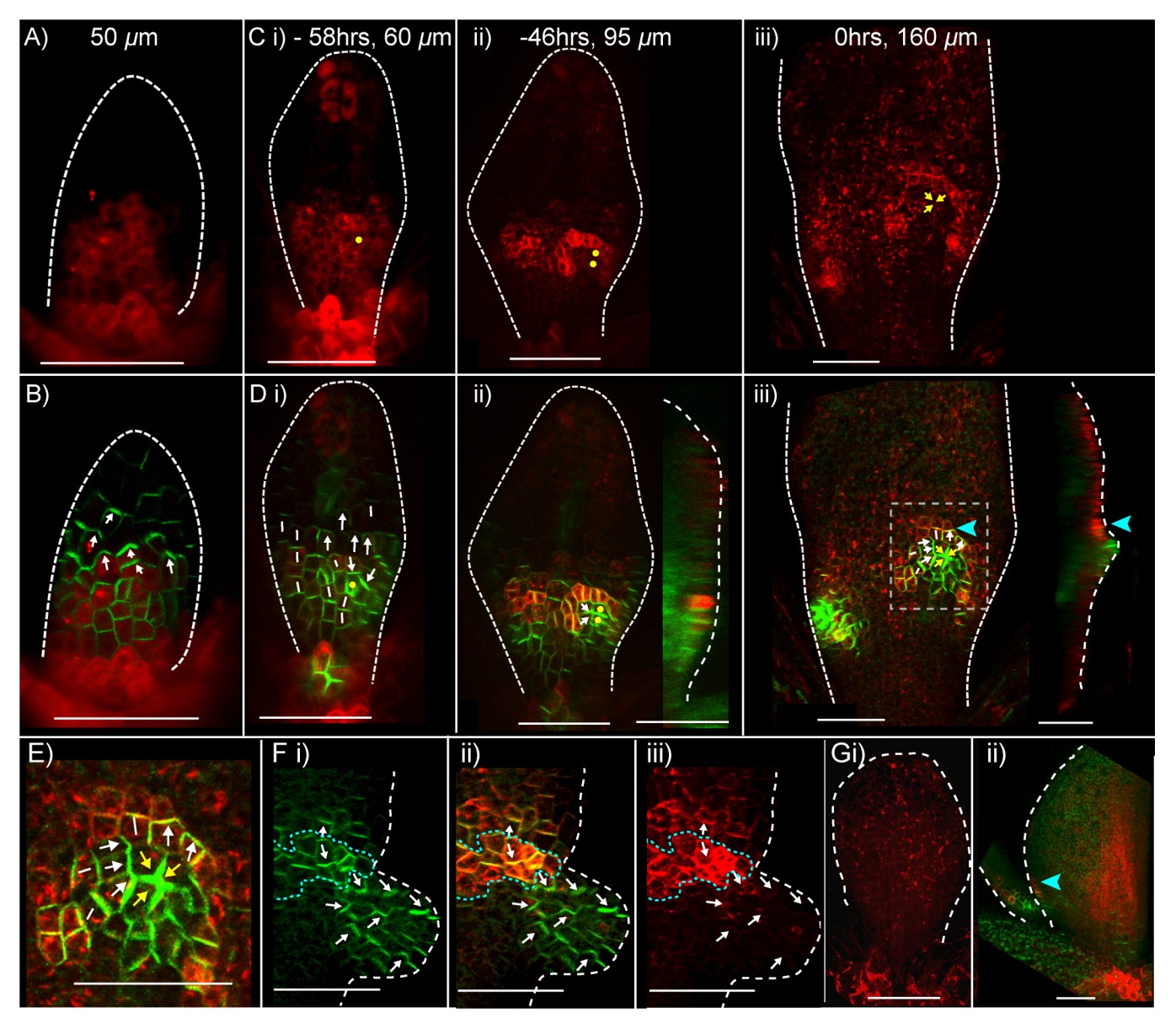

**Figure 6.** Expression of *CUC2::RFPer* and *PIN1::PIN1:GFP* during *kan1kan2* outgrowth development. (**A**) Confocal images of a *CUC2::RFPer* reporter in the abaxial epidermis of a young *kan1kan2* first leaf primordium. (**B**) Combined *CUC2::RFPer* (red) and *PIN1::PIN1:GFP* confocal channels for the same leaf as in **A**. White arrows indicate inferred polarity orientations. (**C**) and (**D**) Time-lapse imaging of *CUC2::RFP* and *PIN1::PIN1:GFP* in the abaxial epidermis of the first leaf of another seedling from that in **A** and **B**, over 58 hr prior to outgrowth development. C: *CUC2::RFPer*, D: combined channels, as in **B**. Times relative to outgrowth emergence and approximate leaf widths are given above each image. The right-hand images in **D ii** and **iii** show optical sections through 3D renderings of the confocal z-stacks at the position of the centre of convergence. White and yellow arrows indicate inferred PIN1 polarity orientations. Yellow dots and arrows indicate the cells that gave rise to the outgrowth tip in **C iii** and **D iii**. Blue arrow heads in **D iii** indicate elevated *CUC2::RFPer* expression distal to a centre of PIN1 convergence and at the base of the emerging outgrowth on its distal side. Data is representative of that obtained by tracking outgrowth development in five *kan1kan2 CUC2::RFPEer PIN1::PIN1:GFP* leaves, in three separate experiments. (**E**) Zoomed-in image of the region outlined by the dashed rectangle in **D iii**, showing PIN1 polarities within and surrounding the domain of elevated *CUC2::RFPer* expression. (**F**) Confocal images of *CUC2::RFPer* (red) and *PIN1::PIN1:GFP* (green) associated with an ectopic *kan1kan2* outgrowth at a later stage of development to that in **C–E**. Dotted blue line outlines the domain with elevated *CUC2::RFPer* expression at the distal boundary between an outgrowth and the main lamina. (**G**) *CUC2::RFPer* expression in two WT first leaf primordia (ii also shows PIN::PIN1:GFP in green). Blue arrow in **ii** indicates a site of elevated *CUC2::RFPer* expression in the leaf margin. Red signal in the centre of the lamina in **ii**) is non-ER localised and therefore due to auto-fluorescence. Data is representative of of that obtained by imaging nine leaves at each of the developmental stages shown, across two separate experiments. Scale bars = 50 $\mu$m. Dashed white lines indicate leaf outlines.

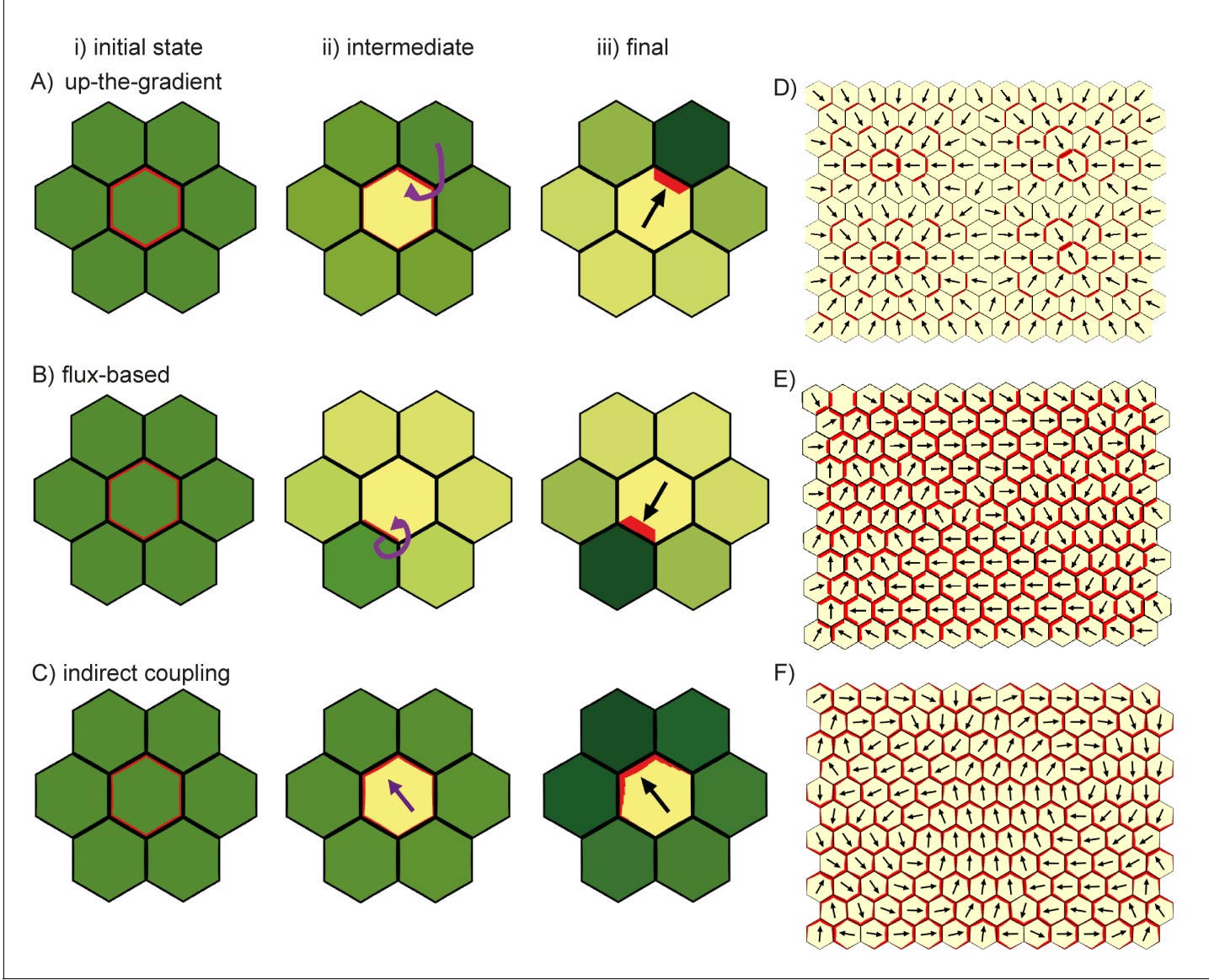

**Figure 7.** Comparison of basic behaviours of indirect coupling, flux-based and up-the-gradient models. (A–C) Investigating the ability of cells to polarise in the absence of pre-established external asymmetries or polarisable neighbours for up-the-gradient (A), flux-based (B) and indirect coupling (C) models. Panels i) show the initial state of the simulations. In each simulation, all cells initially have the same auxin concentration (indicated by the intensity of green, where darker green indicates a higher concentration). In the case of up-the-gradient or flux-based models (A, B), noise is present in the initial levels of PIN1 (shown by red lines at cell edges). In the case of indirect coupling (C), a single central cell has noise in the levels of A* and B*. Panels ii) show intermediate states of the simulations, and panels iii) show the final states. In this and subsequent simulations, PIN1 polarisation is indicated by black arrows, where the arrow points towards the region of the cell with highest PIN1. (A) For the up-the-gradient model, if a given cell edge has slightly elevated PIN, this causes the juxtaposed neighbour to have an increased auxin concentration. This increased auxin concentration in the neighbour feeds-back to cause increased PIN1 recruitment to the given cell edge (purple arrow in ii). (B) For the flux-based model, a cell edge with a slightly elevated PIN1 concentration causes an increased auxin efflux rate. This increased auxin efflux feeds-back to cause an increased recruitment of PIN1 to the given cell edge (purple arrow in ii). (C) For the indirect coupling model, polarity is generated independently from PIN1 and auxin through A* and B* polarity components (purple arrow in ii) which in turn causes polarisation of PIN1. (D–F) Behaviour of up-the-gradient (D), flux-based (E) and indirect coupling (F) models for 2D arrays of polarisable cells in the absence of pre-established asymmetries. D and E were initialised with noise in the initial concentrations of auxin and F was initialised with noise in the levels A* and B* in the membrane. Details of all simulations can be found in the Materials and methods. See *Figure 7—figure supplement 1* for a further comparison of the requirements for polarisation.

The following figure supplement is available for figure 7:

**Figure supplement 1.** Comparison of models' ability to generate polarity for a cell surrounded by a fixed environment.

arrow). Assuming a limited pool of PIN1 in the cell, there will be a corresponding reduction in PIN1 allocation to other cell edges. In the flux-based model polarity arises because if one cell edge has slightly elevated PIN1 (due to small random fluctuations), this edge will have a higher rate of auxin efflux, which feeds back to favour further recruitment of PIN1 to the given cell edge (*Figure 7B ii*, purple arrow). Recruitment of PIN1 to other cell edges is inhibited because the different edges of a cell compete to export auxin into surrounding cells.

As previously described, the indirect coupling model also generates a polarised distribution of PIN1 in the central cell (*Figure 7C*). This is due to the auto-activating and mutually inhibitory activities of membrane bound A\* and B\* polarity components, combined with relatively fast diffusion of their cytoplasmic forms, which generates cell polarity. PIN1 is then recruited to the membrane with high A\* (*Marée et al., 2006*; *Jilkine et al., 2007*).

Thus, all three models have the potential to generate polarity in a single polarisable cell surrounded by non-polarisable neighbours in an initially uniform field of auxin concentration. For the indirect coupling model, this polarity arises through partitioning of the polarity components, while for the flux-based and up-the-gradient models it arises because small fluctuations in PIN1 distribution create variations in auxin flux or concentrations which feed back to reinforce the polarity. There is, however, a key difference between the up-the-gradient and other models. If auxin concentration in the surrounding medium remains fixed throughout the simulation (equivalent to voltage-clamping in neurophysiology), then the up-the-gradient model does not generate polarity (*Figure 7—figure supplement 1*). In contrast, both the flux-based and indirect coupling model present polarity even under such settings (*Figure 7—figure supplement 1*).

In a previous paper we overlooked the potential effect of feedback in the flux-based and up-the-gradient models for single polarisable cells, and implicitly assumed that such cells would not polarise (*Abley et al., 2013*). Moreover, we did not distinguish between lack of pre-established asymmetric cues versus lack of being able to establish asymmetric cues (e.g. through auxin transport). The above simulations show that this distinction needs to be taken into account to appreciate that, whereas the flux-based model can present intracellular partitioning (i.e. establish polarity in the absence of asymmetric cues or polarisable neighbours), the up-the-gradient model relies on its ability to change its surrounding to establish and maintain polarization. Thus, both the flux-based and up-the-gradient model share commonalities with the indirect coupling model regarding intracellular partitioning, albeit through a more indirect mechanism involving auxin transport between cells.

We next consider an array of polarisable cells in the absence of pre-established asymmetries. Noise is present in initial auxin concentrations (for the up-the-gradient and flux-based models), or in the A\*-B\* polarity components (for the indirect coupling model), and all cells have the ability to relocate PIN1 (*Figure 7D–F*). In this situation the up-the-gradient model generates convergent polarities, in which spaced groups of cells orient their PIN1 polarity towards a central region (*Figure 7D*) (*Jönsson et al., 2006*; *Smith et al., 2006*). This is because fluctuations lead to competing centres of high auxin concentration which orient polarities towards them.

By contrast, the flux-based model tends to generate swirled patterns of polarity, in which polarities are coordinated in tandem between neighbouring cells (*Figure 7E*). Convergent polarities are disfavoured because if two cells have PIN1 oriented towards each other, both cells would experience a low net auxin efflux across their PIN1-rich ends due to transport towards them from the opposing neighbour. The low net auxin efflux would cause removal of PIN1 from the membranes and relocalisation to edges juxtaposed with a PIN1-free edge of a neighbouring cell, promoting tandem alignments.

Similar to the flux-based model, indirect coupling also generates swirled patterns of polarity (*Figure 7F*, *Abley et al., 2013*). Convergent alignments are disfavoured because if two cells have PIN1 oriented towards each other, both cells transport auxin to the intervening extracellular space. Accumulation of auxin in the extracellular space then inhibits A\*, and therefore PIN1, in adjacent membranes. This destabilises convergent polarities and favours tandem alignments.

Thus, in the absence of pre-established asymmetries, the models spontaneously generate two types of polarity pattern. The flux-based and indirect coupling models both generate tandem alignments, and are subsequently referred to as tandem alignment models; while the up-the-gradient model generates convergent alignments, and is subsequently referred to as a convergent alignment model.

We next explore the assumptions that need to be added to these models for them to account for the epidermal PIN1 polarity patterns observed in WT and *kan1kan2* leaves. With each model, we attempt to capture the initial proximodistal polarity field observed in both WT and *kan1kan2*, where polarities are aligned in tandem along the proximodistal leaf axis with high intracellular auxin at the tip (*Figure 1*). We then explore the assumptions required for the formation of centres of PIN1 convergence with elevated intracellular auxin, such as those observed on the abaxial side of the main lamina at later stages of *kan1kan2* leaf development (*Figure 2B,C*, *Figure 3*).

The convergent alignment model has been proposed to account for the initial centre of convergence at the distal end of the leaf primordium (*Jönsson et al., 2006*; *Smith et al., 2006*). For a proximodistal polarity field to be maintained within the growing primordium, the convergent alignment model requires maintenance of an increasing auxin concentration gradient from the leaf base to the leaf tip. One way this can be achieved is through net auxin removal at the leaf base. Net auxin removal from a region of tissue could be achieved through a decreased rate of auxin biosynthesis, an increased rate of transport away from the region, or an increased rate of auxin degradation or conjugation. Here net removal is achieved using elevated auxin degradation rates (*Figure 8A*). Combining this assumption with an initially elevated intracellular auxin concentration at the leaf tip, a proximodistal polarity field can be maintained within an array. The intracellular auxin concentration is kept high at the distal end of the array through transport towards this region (*Figure 8A*), similar to experimental observations using the *DR5::GFP* reporter (*Figure 3*, *Mattsson et al., 2003*; *Scarpella et al., 2006*). As the primordium increases in size (*Figure 8B*), the auxin concentration tends to become shallow at a distance from the leaf tip. This can result in the spontaneous formation of a centre of convergence with elevated intracellular auxin within the proximal region of the lamina (*Figure 8B*), similar to that observed in *kan1kan2* mutants (*Figure 2B,C*). The length of the tissue at which the polarity reversal occurs depends on the parameters values used in the model. The failure to form such a convergence centre in WT leaves could be explained by the PIN1 polarity pattern becoming fixed at early stages of development, preventing polarity reorientation. However, how this could be achieved is unclear, although it has been proposed that absence of *CUC2* activity may play a role in fixing polarity (*Bilsborough et al., 2011*).

In the flux-based model, polarities become oriented away from auxin sources and towards auxin sinks. A proximodistal polarity pattern may therefore be generated with net auxin production (a source) proximally and elevated net auxin removal (a sink) distally (*Figure 9A*). Here we simulate net auxin production in proximal regions with an increased auxin biosynthesis rate, but it could also be achieved with a reduced auxin degradation rate or increased auxin influx from (or decreased auxin efflux to) tissues beyond those represented in the simulation. Likewise, net removal from the leaf tip is simulated using elevated degradation, but could also occur through reduction in biosynthesis, increase in conjugation, or via transport into underlying tissue layers (*Bayer et al., 2009*; *Scarpella et al., 2006*).

With these assumptions the flux-based model generates a proximodistal pattern of polarities. However, intracellular auxin concentrations tend to be lowest at the distal end of the tissue (*Figure 9A*), inconsistent with experimental observations. This inconsistency can be avoided if the leaf tip acts as a weak instead of a strong auxin sink. The initial drop in intracellular auxin at the tip is then followed by a rise to a high level because of the transport towards this region (*Stoma et al., 2008*; *Walker et al., 2013*). Another possibility is that the leaf tip has an elevated auxin import rate as well as an elevated rate of auxin removal (*Figure 9B*). Elevated auxin import in the distal-most cells causes increased auxin flux towards them, encouraging polarities to point distally. The elevated import also causes the distal-most cells to accumulate intracellular auxin. Although auxin is synthesised at the highest rate in the proximal-most files of cells, in the final state of the simulation, maximal auxin concentrations are found at the distal end (*Figure 9B*). This is due to polar transport of auxin away from the base, towards the leaf tip.

Similar to the flux-based model, with indirect coupling an initial proximodistal alignment of polarities can be generated if the proximal region (leaf base) has net auxin production (here due to elevated auxin biosynthesis generating high extracellular auxin), while the distal region (leaf tip) has elevated net auxin removal (here due to elevated auxin degradation generating low extracellular auxin). As in the flux-based model, elevated auxin degradation at the distal end of the tissue can lead to low intracellular auxin in this region (*Figure 9C*). Combining an elevated rate of auxin import at the leaf tip with the elevated degradation ensures low extracellular auxin concentrations in this

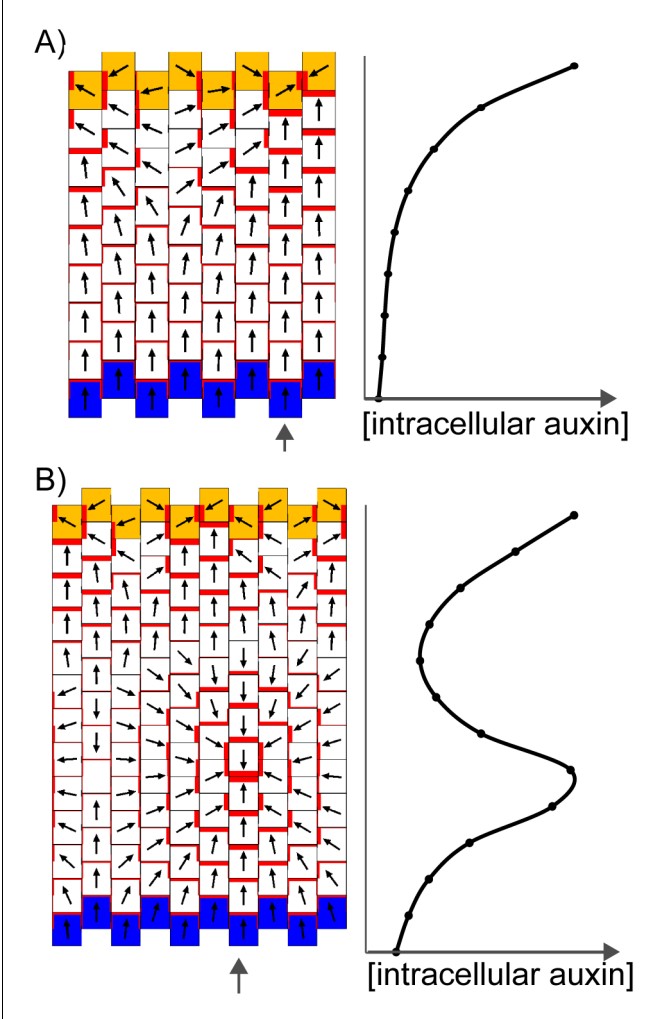

**Figure 8.** Formation of a proximo-distal polarity field and centres of convergence in the convergent alignment model. (**A**) Formation of a proximo-distal polarity field due to the presence of an elevated initial auxin concentration at the leaf tip (orange cells) and an elevated rate of auxin removal from the leaf base (blue cells). The graph shows the concentration of intracellular auxin for each cell in the column marked with the grey arrow. (**B**) As for **A**, but for a larger tissue. A centre of polarity convergence forms within the proximal half of the lamina.

region (encouraging polarities to point distally) and allows distal cells to accumulate intracellular auxin (*Figure 9D*). Thus, the two tandem alignment models both have similar requirements for the generation of a proximodistal polarity field with an intracellular auxin maximum in the distal region.

The predictions of the tandem alignment models for the generation of a proximodistal polarity field differ from those of the convergent alignment model. Tandem alignment models predict that the leaf base has net production of auxin, whilst the convergent alignment model predicts that this region has net auxin removal. Also, tandem alignment models require that the leaf tip has an elevated rate of auxin import and removal to account for high auxin in this region, whereas the convergent alignment model has no such requirement.

A further difference is that unlike the convergent alignment model, the two tandem alignment models do not spontaneously generate centres of polarity convergence upon an increase in tissue size (the results of simulations shown in *Figure 9A–D* are similar for a range of tissue sizes). Thus, with tandem alignment models, further assumptions are needed to generate centres of PIN1 polarity convergence. With tandem alignment models, a site of convergence could arise by a cell having higher levels of auxin removal, caused by export to underlying cells. This cell could be positioned in

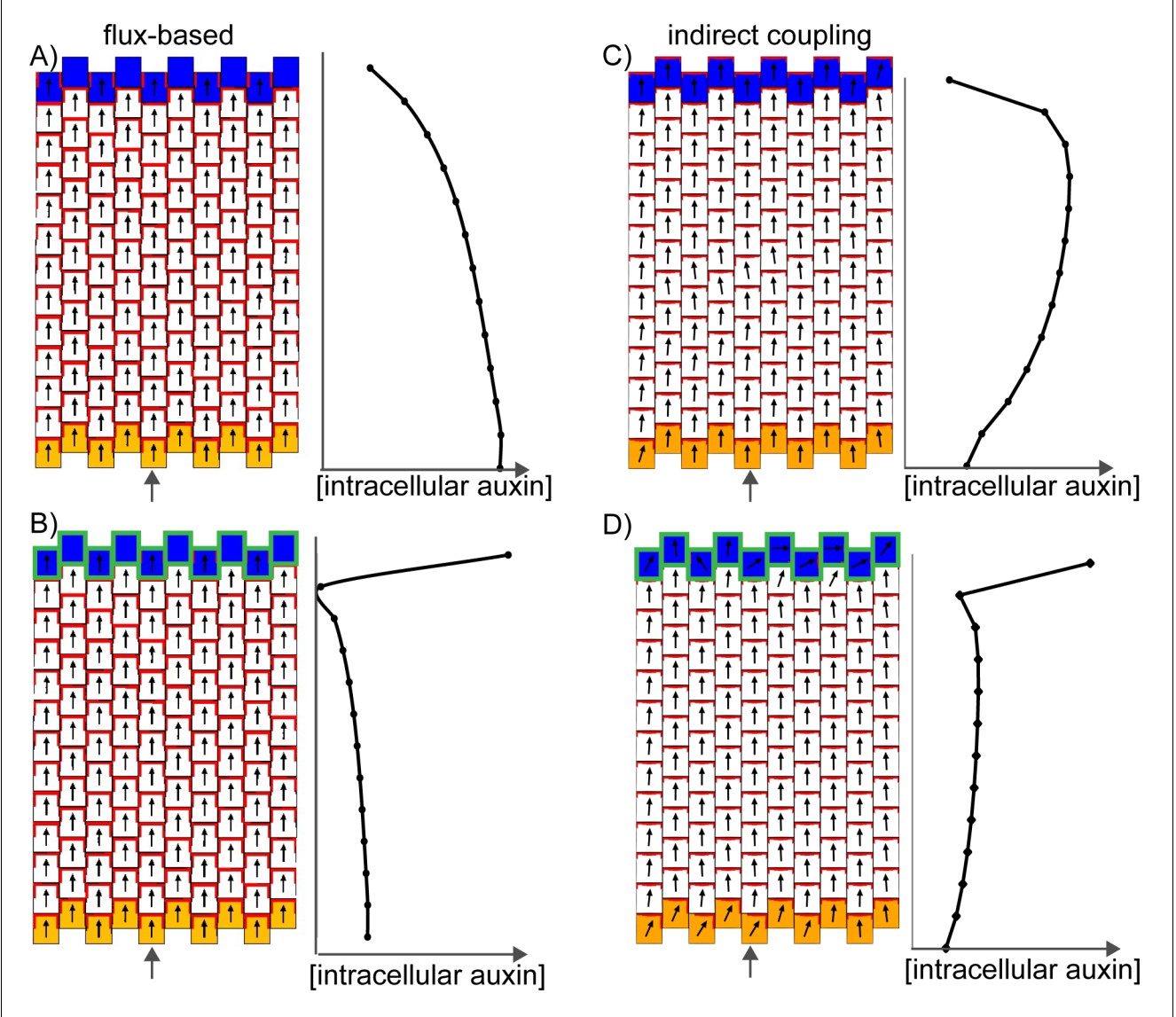

**Figure 9.** Formation of a proximo-distal polarity field in the flux-based and indirect coupling models. (**A**) Formation of a proximo-distal polarity field in the flux-based model due to the presence of elevated auxin biosynthesis at the leaf base (orange cells) and an elevated rate of auxin removal from the leaf tip (blue cells). The graph on the right shows the concentration of intracellular auxin for the column marked with the grey arrow. (**B**) As for **A**, but with elevated auxin import (green cell outlines) at the distal end of the tissue. (**C**) As for **A**, but for the indirect coupling model. (**D**) As for **B**, but for the indirect coupling model.

regions of elevated intracellular auxin, as proposed by *Stoma et al, 2008*. However, induction of auxin removal alone can result in a drop in intracellular auxin concentration before convergence formation (*Stoma et al., 2008*), and it is unclear how such regions of elevated intracellular auxin could emerge in the proximal half of the *kan1kan2* abaxial lamina.

One possible explanation for the generation of convergence sites in tandem alignment models is that the proximal region of the abaxial lamina has an extended region of auxin biosynthesis. This region corresponds to the region of ectopic *CUC2::RFP* expression in the proximal region of young *kan1kan2* primordia (*Figure 6A,C i*). In the tandem alignment models, auxin biosynthesis in this region leads to a shallow peak in intracellular auxin proximal to the distal limit of auxin biosynthesis (*Figure 10A,C*). This peak arises as a consequence of elevated auxin biosynthesis in this domain, together with polarised transport which shifts the position of the peak distally. If intracellular auxin

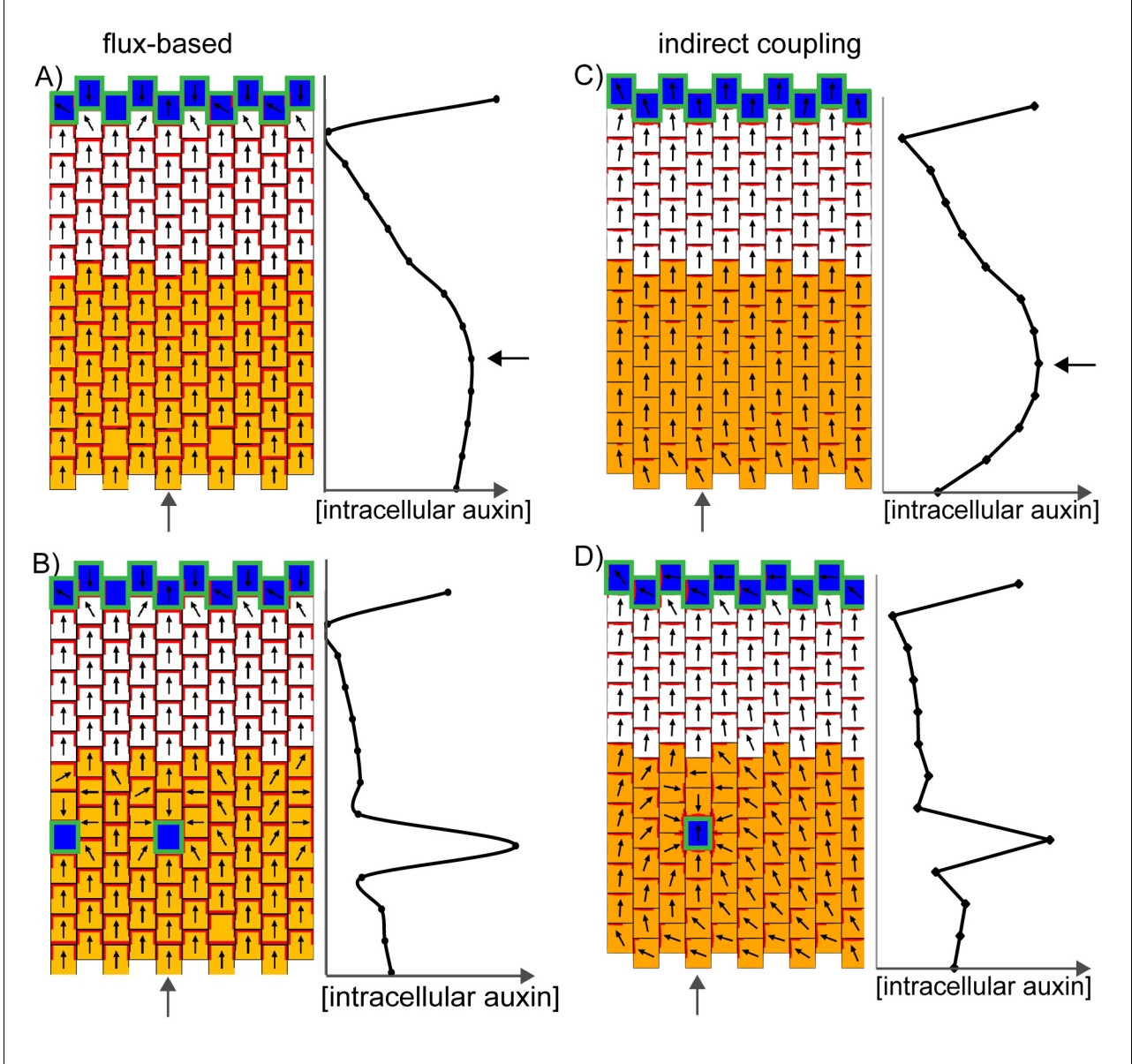

**Figure 10.** Formation of centres of polarity convergence in the flux-based and indirect coupling models. (A) A proximo-distal polarity field is initially established with the flux based model due to the presence of elevated auxin biosynthesis in the proximal half of the leaf (orange cells) and elevated rates of auxin import and removal from the leaf tip (blue cells with green outlines). The graph on the right shows the concentration of intracellular auxin for the column marked with the grey arrow. Note that there is a broad peak of intracellular auxin concentration within the proximal domain of elevated auxin synthesis (black arrow). (B) Subsequent stage of the simulation in A. If it is assumed that intracellular auxin above a threshold concentration induces elevated auxin import and removal, one or more cells within the proximal domain of elevated auxin biosynthesis are induced to have elevated levels of import and removal. Neighbouring cells reorient their polarity to point towards the high import cells, which accumulate elevated levels of intracellular auxin. (C) As for A), but for the indirect coupling model. (D) As for B), but for the indirect coupling model. See *Figure 10—figure supplement 1* for an alternative version of the indirect coupling model which incorporates D6-kinase like activity.

The following figure supplement is available for figure 10:

**Figure supplement 1.** Incorporation of a D6 protein kinase-like activity into the indirect coupling model.

above a threshold concentration activates auxin import, a region with elevated import arises, leading to a centre of polarity convergence (*Figure 10B,D*). Elevated auxin removal needs to arise in addition to elevated auxin import since import alone causes very high levels of intracellular auxin to accumulate at the centre of convergence. This can disrupt convergence formation by preventing the maintenance of low extracellular auxin (in the indirect coupling model), and by increasing total auxin efflux from the central cell (in the flux-based model). Unlike the model of *Stoma et al., 2008*, where only auxin removal is induced by high intracellular auxin, the combination of elevated auxin import and removal helps to ensure that a dip in intracellular auxin concentration does not occur prior to convergence formation.

D6 protein kinases are polarly localized in cells independently of PIN proteins and their phosphorylation of PIN proteins is required for PIN-mediated auxin efflux (*Barbosa et al., 2014*, *Zourelidou et al., 2009*). We added a representation of D6 kinase activity to the indirect coupling model, as cell polarity in this system does not depend on PIN activity, and found that this does not qualitatively affect model behavior (*Figure 10—figure supplement 1*).

This comparison of models gives rise to specific predictions that may be used to distinguish between them experimentally. Tandem alignment models require that, at the leaf tip and at the tips of outgrowths (where centres of convergence are located), the epidermis has elevated rates of auxin import and auxin removal (which could be removal into underlying tissue layers). Locally elevated auxin import is not required for the convergent alignment model, as sites with slightly elevated intracellular auxin reinforce themselves. The tandem alignment models predict that the epidermis towards the leaf base has elevated net auxin production, whilst the convergent alignment model predicts that it has net auxin removal. Moreover, a broader band of auxin biosynthesis at the base of the abaxial side of the *kan1kan2* leaf provides one possible mechanism for inducing convergence site formation with tandem alignment models.

## Expression of auxin importers is elevated at the tips of *kan1kan2* outgrowths and is required for their normal development

To test whether auxin import is elevated at centres of polarity convergence, we determined the expression pattern of auxin importers in *kan1kan2* mutant leaves. The *AUX/LAX* family of auxin importers includes four genes in *A.thaliana* (*AUXIN RESISTANT 1* (*AUX1*), *LIKE AUXIN RESISTANT 1* (*LAX1*), *LIKE AUXIN RESISTANT 2* (*LAX2*), and *LIKE AUXIN RESISTANT 3* (*LAX3*)) which encode proteins that actively transport auxin from the extracellular space into the cytoplasm (*Parry et al., 2001*; *Péret et al., 2012*; *Yang et al., 2006*). We focus on *AUX1* and *LAX1* expression since both genes are expressed in the leaf epidermis and therefore may contribute towards the auxin import activity predicted by the models (*Kasprzewska et al., 2015*). Consistent with previous reports, expression of *LAX2::GUS* was detected in leaf vascular tissue (*Figure 11—figure supplement 1A*) and expression of *LAX3::GUS* was absent from leaves but present in vascular tissue of hypocotyls and roots (*Bainbridge et al., 2008*; *Kasprzewska et al., 2015*) (*Figure 11—figure supplement 1B,C*).

As expected from the tandem alignment models and consistent with a previous report, *LAX1::GUS* was expressed at the tips of young wild-type leaf primordia, at the tips of serrations, and in the midvein, but was absent from the rest of the leaf lamina (*Kasprzewska et al., 2015*) (*Figure 11A*). In *kan1kan2* leaves, *LAX1::GUS* was expressed in the same regions as in WT, but was also ectopically expressed in groups of a few cells in proximal regions of the abaxial epidermis (*Figure 11B*). These groups of cells were at the tips of developing outgrowths (*Figure 11B ii and iii*). To investigate whether *LAX1::GUS* expression preceded outgrowth emergence, sections of *kan1kan2LAX1::GUS* seedlings stained to reveal GUS activity were imaged. This revealed that local epidermal sites of elevated *LAX1::GUS* expression were present in the abaxial lamina prior to outgrowth emergence (*Figure 11C i*), consistent with a role of *LAX1* in the generation of centres of polarity convergence. Transverse sections also confirmed that *LAX1::GUS* expression at the tips of outgrowths is located in the epidermis (*Figure 11C ii*).

In wild-type primordia, *AUX1::AUX1:YFP* was expressed in all cells of the abaxial epidermis, with strongest expression at the leaf margin (*Figure 11Di*). On the adaxial leaf surface, expression was excluded from most of the epidermis, but detected in cells close to and within the leaf margin (*Figure 11Dii*). *kan1kan2* leaves showed a similar pattern of *AUX1::AUX1:YFP* expression on their adaxial surface, but the expression pattern in the abaxial epidermis differed from that in wild type. Expression was found throughout distal regions of the leaf, but was concentrated in cells of

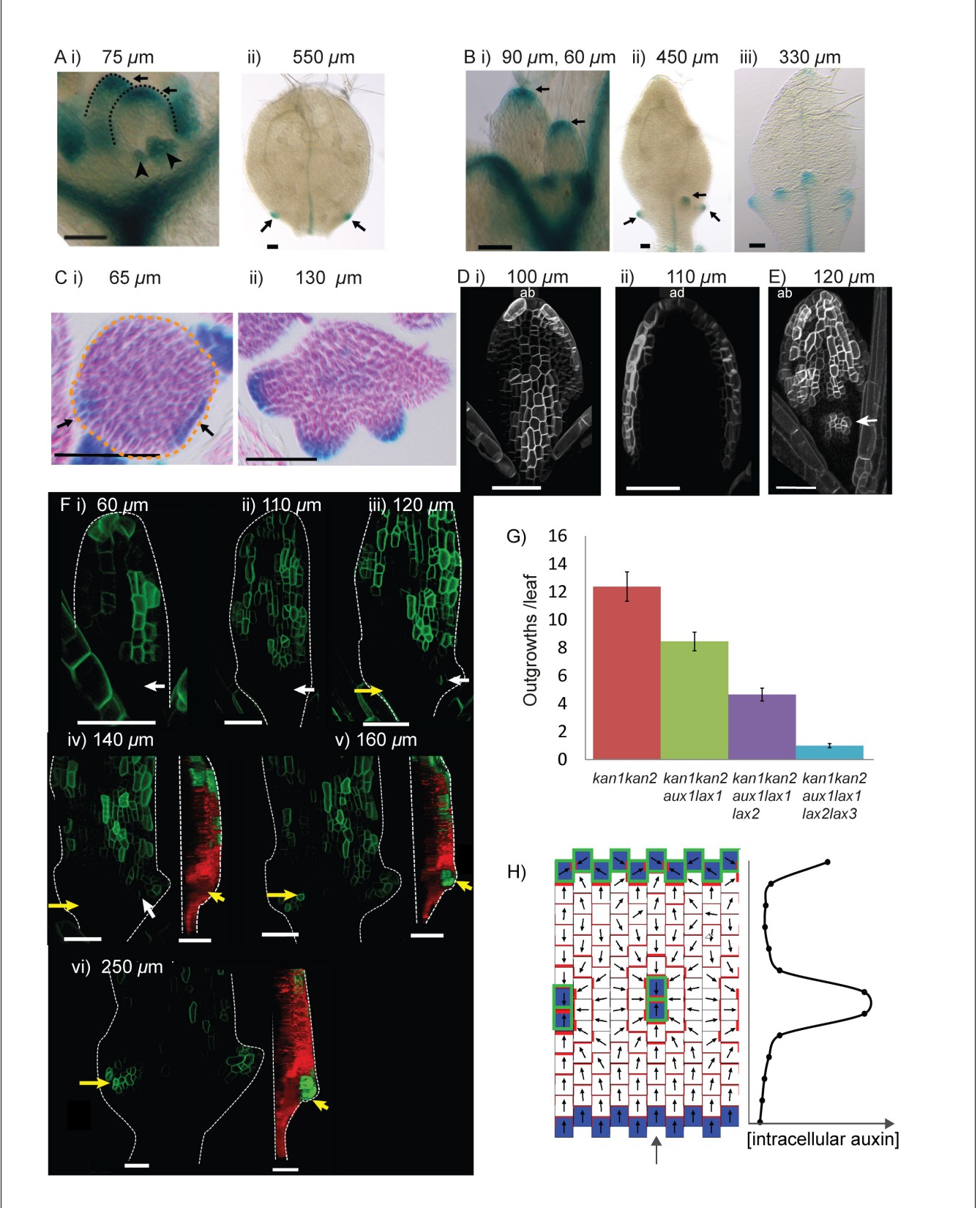

**Figure 11.** Expression of *LAX1::GUS* and *AUX1::AUX1:YFP* in WT and *kan1kan2* leaves. (**A**) Expression pattern of *LAX1::GUS* in WT leaves one and two. *LAX1::GUS* was expressed at the tips of developing primordia (arrows in (**i**), black dashed lines indicate leaf outlines, arrow heads indicate stipules) (a

*Figure 11 continued on next page*

*Figure 11 continued*

total of 6 plants (with 12 young leaves) were imaged across two separate experiments) and at the tips of serrations (ii) (arrows indicate serrations) (a total of 9 plants (18 leaves) were imaged across three separate experiments). Leaf widths are given above images. (B) Expression pattern of *LAX1::GUS* in *kan1kan2* leaves. *LAX1::GUS* was expressed at the tips of primordia (i) (a total of 7 plants (14 leaves) were imaged across two separate experiments) and at the tips of outgrowths (ii and iii) (arrows) (20 out of 21 leaves imaged across three separate experiments). ii) shows leaf 1, iii) shows leaf 3. (C) Transverse sections through GUS stained *kan1kan2 LAX1::GUS* seedlings, showing points of *LAX1::GUS* expression before outgrowths have emerged (black arrows in i) (data supported by serial sections of 3 other leaves at similar developmental stages in a separate experiment) and at the tips of developing outgrowths (ii) (a total of 8 leaves were imaged across two separate experiments). Dashed orange line in i) indicates leaf outline. (D) *AUX1::AUX1:YFP* expression in leaf 1 of WT, showing abaxial surface (i) (a total of 15 leaves were imaged across two separate experiments) and adaxial surface (ii) (a total of 4 leaves were imaged in two separate experiments) of two different leaves. (E) *AUX1::AUX1:YFP* expression in leaf 1 of a *kan1kan2* mutant, showing abaxial surface. Arrow points to the tip of an emerging outgrowth with locally elevated *AUX1::AUX1:YFP* signal (a total of 9 leaves were imaged, across three different experiments). (F) Time-lapse confocal imaging of *AUX1::AUX1:YFP* in the abaxial epidermis of the first leaf of a *kan1kan2* seedling. White arrows mark the positions of cells that eventually gave rise to the *AUX1::AUX1:YFP* expressing cells in the developing serration on the right side of the leaf. Yellow arrows mark the positions of cells which eventually gave rise to the tip of the ectopic outgrowth on the left side of the leaf. Red and green images in iv), v) and vi) show side views, from the left hand side of the leaf (showing that the outgrowth emerged at the time point shown in v). Red shows auto-fluorescence and a *CUC2::RFP* marker (used to show the leaf contours), and green shows *AUX1::AUX1:YFP* signal. Times from the beginning of the experiment at which images were taken are: i) 0 hrs, ii) 22 hr 40 min, iii) 31 hr 10 min, iv) 46 hr 40 min, v) 55 hr 40 min, vi) 74 hr 40 min. Data is consistent with tracking experiments performed for two other *kan1kan2* leaves that developed ectopic outgrowths, and with snapshot images of at least 6 leaves before and after outgrowth emergence. (G) Mean number of outgrowths (+/- standard error of the mean) in rosette leaves of *kan1kan2* plants carrying mutant alleles of *AUX/LAX* genes. n numbers are: *kan1kan2*: 84 leaves from 10 plants, *kan1kan2aux1lax1*: 110 leaves from 10 plants, *kan1kan2aux1lax1lax2*: 136 leaves from 10 plants, *kan1kan2aux1lax1lax2lax3*:200 leaves from 10 plants. (H) Convergent alignment model where elevated intracellular auxin causes elevated rates of auxin import and removal. The model is initialised with elevated auxin degradation at the leaf base (blue cells) and an elevated rate of auxin import and degradation at the leaf tip (blue cells with green outlines). Centres of convergence form within the proximal half of the lamina and when the intracellular auxin concentration exceeds a threshold level, an elevated rate of auxin import and removal is induced (blue cells with green outlines). Scale bars = 50 µm. See *Figure 11—figure supplement 1* for *LAX2::GUS* and *LAX3::GUS* expression.

The following source data and figure supplement are available for figure 11:

**Source data 1.** Counts of outgrowths used to generate *Figure 11G*.

**Figure supplement 1.** Expression patterns of *LAX2::GUS* and *LAX3::GUS*.

emerging outgrowths in proximal regions (*Figure 11E*). Time-lapse confocal imaging revealed that, prior to outgrowth formation, *AUX1::AUX1:YFP* expression was absent from proximal regions (*Figure 11Fiii,iv*, yellow arrow), but became detectable as outgrowths emerged (*Figure 11Fv–iv*, yellow arrows). Since centres of PIN1 polarity convergence form at least one day before outgrowth emergence, these observations suggest that strong *AUX1* expression is not involved in their initial development. However, local elevation of *AUX1* expression at outgrowth tips could play a role in the stabilisation and maintenance of convergent polarities.

To investigate whether AUX/LAX auxin importers are required for the generation of ectopic outgrowths, we generated *kan1kan2aux1lax1lax2lax3* hextuple mutants. Loss of all four *AUX/LAX* genes caused a reduction in the mean number of outgrowths per rosette leaf from 12.4 (in *kan1kan2* mutants) to 1.6 (in *kan1kan2aux1lax1lax2lax3* mutants) (*Figure 11G*). To determine the contribution of *lax2* and *lax3* to the loss of outgrowths, *kan1kan2aux1lax1* quadruple and *kan1kan2aux1lax1lax2* quintuple mutants were also generated. Loss of function of *aux1* and *lax1* alone caused a reduction in the number of outgrowths compared with *kan1kan2* plants (*Figure 11G*). This reduction was further increased by a loss of *LAX2* activity (*kan1kan2aux1lax1lax2* mutants have a further loss of outgrowths [*Figure 11G*]). Since *LAX2::GUS* expression was only detected in sub-epidermal tissue, this finding supports a role of sub-epidermal auxin importer expression in the generation of outgrowths. *LAX2*-mediated sub-epidermal auxin import below centres of convergence could contribute to removal of auxin from the centre of convergence. Despite the absence of *LAX3::GUS* reporter expression in leaves, loss of *LAX3* caused a further reduction in the number of outgrowths (*kan1kan2aux1lax1lax2* mutants generated more outgrowths than *kan1kan2aux1lax1lax2lax3* mutants, [*Figure 11G*]). This suggests that *LAX3* may function redundantly with other members of the *AUX/LAX* family in the generation of ectopic outgrowths, or that differences in genetic background between the multiple mutants has an effect on outgrowth frequency (see Materials and methods).

The findings that auxin importers are expressed in the epidermis at the tips of leaves and emerging *kan1kan2* outgrowths, and play a role in outgrowth development, meet the requirement of tandem alignment models (*Figures 9* and *10*). In its most parsimonious form, there is no need for elevated auxin import at the leaf tip or at subsequent centres of convergence with the convergent alignment model (*Figure 8B*). However, this does not make the model incompatible with elevated auxin import at these sites. If it is assumed that intracellular auxin above a threshold concentration induces elevated auxin import and removal, elevated import and removal may be induced after convergence formation (*Figure 11H*). This does not cause disruption of the centres of convergence. Distally oriented polarities near the leaf tip may also be achieved if these cells have an elevated rate of auxin import and removal (rather than an elevated initial auxin concentration) (*Figure 11H*). Thus, induction of auxin import by elevated intracellular auxin concentrations is compatible with the convergent alignment model, though it renders the model less parsimonious.

## *CUC2* is required for locally elevated auxin import at *kan1kan2* outgrowth tips

To test whether *CUC2* could promote the formation of *kan1kan2* outgrowths by influencing expression of auxin importers, we examined the expression of *LAX1::GUS* and *AUX::AUX1:YFP* reporters in the *kan1kan2cuc2* background. Sites of detectable *LAX1::GUS* expression were lost from the *kan1kan2cuc2* lamina (*Figure 12A*). Additionally, in *kan1kan2cuc2* leaves, *AUX1::AUX1:YFP* was expressed uniformly throughout the abaxial lamina (*Figure 12B*), suggesting that *CUC2* represses the expression of *AUX1* in the region surrounding outgrowths. In support of this conclusion, *AUX1::AUX1:YFP* and *CUC2::RFP* were found to be expressed in mutually exclusive domains surrounding outgrowths, with *AUX1::AUX1:YFP* expression being excluded from the region of high *CUC2::RFP* expression in the outgrowth axil (*Figure 12C*).

These findings suggest that *CUC2* is required for both the regions of elevated *LAX1* expression at outgrowth tips and the regions of low *AUX1::AUX1:YFP* expression surrounding emerging outgrowths. Thus, *CUC2* contributes to locally elevated auxin import at the outgrowth tip. These findings are compatible with a role for *CUC2* in promoting convergence formation in the context of a tandem alignment polarity mechanism.

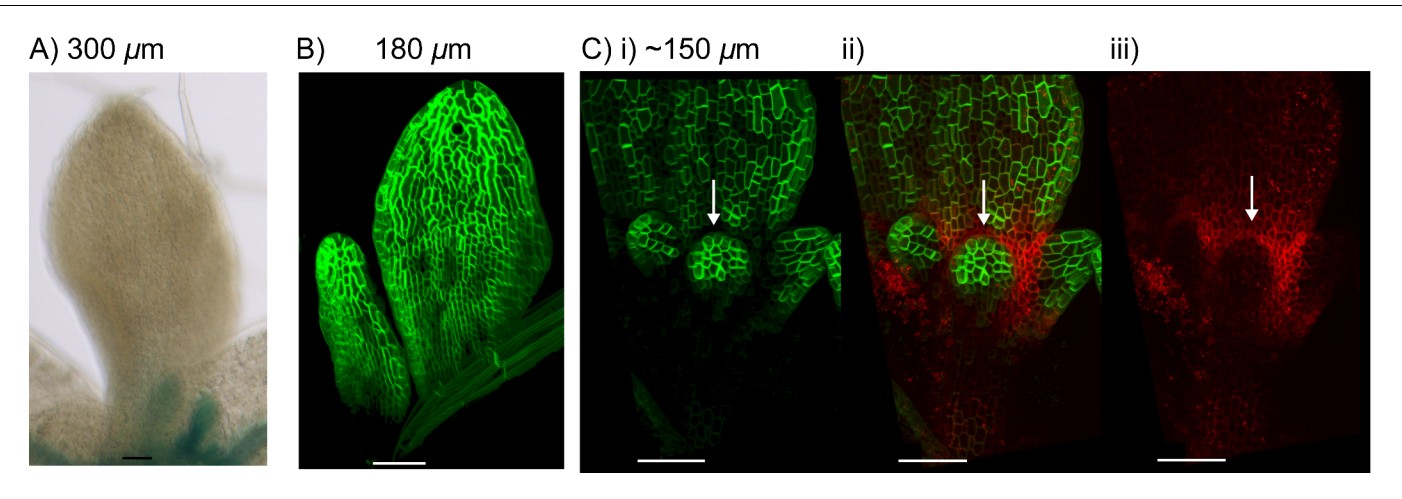

**Figure 12.** *LAX1* and *AUX1* expression in *kan1kan2cuc2* mutants. (**A**) Expression of *LAX1::GUS* in the abaxial surface of leaf 1 of a *kan1kan2cuc2* mutant. Data representative of images from ten seedlings in two separate experiments. (**B**) *AUX1::AUX1:YFP* expression in leaf 3 and 4 of a *kan1kan2cuc2* mutant. Images representative of those taken from fifteen seedlings in two separate experiments. (**C**) *CUC2::RFPer* and *AUX1::AUX1:YFP* expression in leaf 3 of a *kan1kan2* mutant leaf i) *AUX1::AUX1:YFP*, ii) *CUC2::RFPer* (red) and *AUX1::AUX1:YFP* (green), iii) *CUC2::RFPer*. White arrows indicate region of low *AUX1::AUX1:YFP* and elevated *CUC2::RFPer* expression at the distal base of developing outgrowths. Images representative of those taken from six seedlings in two experiments. Approximate leaf widths are given above each image. Scale bars = 50 $\mu$m.

## *kan1kan2* epidermal centres of convergence are coupled with sub-epidermal PIN1 strands

We next test whether centres of convergence have elevated auxin removal rates as well as elevated rates of import. One way that auxin could be removed from epidermal centres of convergence is through transport into underlying tissue layers. In the shoot apical meristem, and in serrations, epidermal centres of PIN1 convergence are linked with sub-epidermal strands of PIN1 expression with PIN1 oriented auxin away from the epidermis (*Hay et al., 2006*; *Reinhardt et al., 2003*; *Scarpella et al., 2006*). To test whether this is also the case for *kan1kan2* outgrowths, we immuno-localised PIN1 in transverse cross-sections of *kan1kan2* leaves before (*Figure 13A*) and after (*Figure 13B*) outgrowth emergence. At both time points, epidermal centres of convergence were coupled with sub-epidermal strands of cells with elevated PIN1 expression (*Figure 13*). PIN1 in these strands was polarised away from the epidermis, towards underlying tissue (*Figure 13A*), suggesting that the strands provide a route for the removal of auxin from the epidermis. This observation

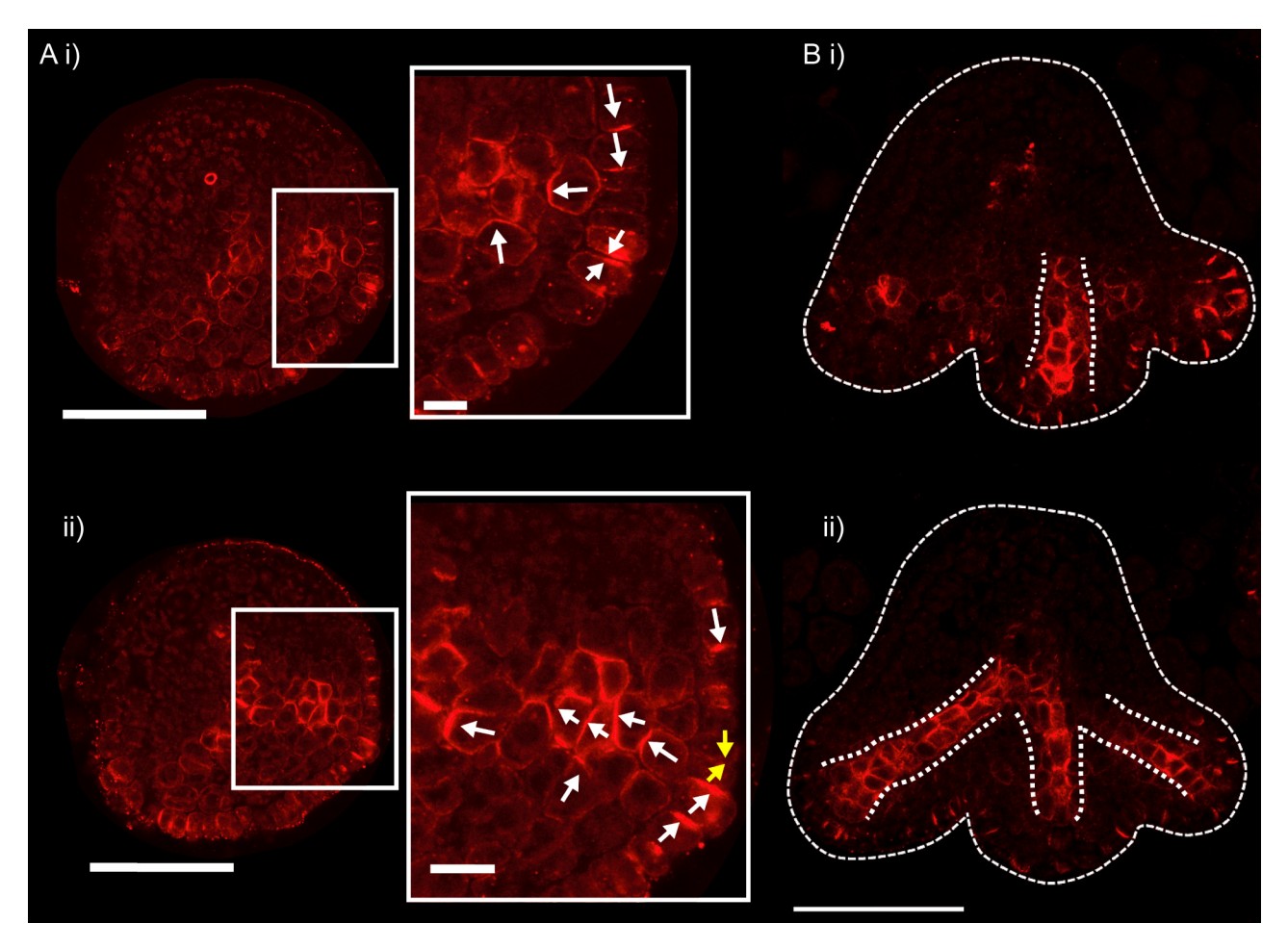

**Figure 13.** PIN1 immuno-localisation in transverse cross-sections of *kan1kan2* leaves. (**A**) PIN1 immuno-localisation in a *kan1kan2* leaf before outgrowth emergence. i) and ii) are consecutive sections through the tissue (each section is 8 μm thick). A centre of PIN1 polarity convergence can be seen in i) and the site of this convergence is shown with yellow arrows in ii). Arrows show the inferred directions of PIN1 polarity. The adaxial side of the leaf is at the top of the image. 4 leaves with convergences were imaged across two separate experiments. (**B**) PIN1 immuno-localisation in a *kan1kan2* leaf in which outgrowths have begun to emerge. Dotted white lines indicate the strands of cells with elevated levels of PIN1 and the leaf outline. 3 leaves with emerging outgrowths were imaged across two separate experiments. All scale bars = 50 μm, except those in the zoomed in panels in **A**, which are 10 μm.

provides a mechanism for auxin removal required for the tandem alignment models (*Figure 10B,D*). It is compatible with the convergent alignment model (*Figure 11H*) but not required (*Figure 8B*).

## Patterns of *YUC1* and *YUC4* expression are more consistent with tandem alignment models

The epidermis at the leaf base is predicted to have net auxin production by the tandem alignment models, and one way this could be achieved is through elevated auxin biosynthesis. To identify likely locations of elevated auxin production in the leaf, we analysed the expression patterns of genes encoding YUCCA auxin-biosynthesis enzymes in wild-type and *kan1kan2* leaves. *A.thaliana* has six *YUCCA* genes, three of which (*YUCCA1* (*YUC1*), *YUCCA2* (*YUC2*) and *YUCCA4* (*YUC4*)) are expressed in the leaf and are redundantly required for *kan1kan2* outgrowth development (*Cheng et al., 2007*, *2006*; *Wang et al., 2011*). *YUC1::GUS* was previously reported to be expressed at the base of wild-type leaf primordia, in all cell layers (*Wang et al., 2011*). Using altered staining conditions to restrict the diffusion of the coloured GUS product, we found that the expression of this reporter was restricted mainly to outer cell layers at the base of the wild-type leaf, with strongest expression on the adaxial side (*Figure 14A i,ii*). This adaxial expression was detected in very young wild-type leaf primordia emerging from the shoot apical meristem (*Figure 14B*), and expression at the leaf base persisted during leaf development (*Figure 14C*).

The expression of *YUC1::GUS* in the epidermis at the base of primordia matches the predictions of the tandem alignment models. When this region of auxin production is added to the convergent alignment model along with auxin import and removal at the leaf tip, the formation of a proximo-distal polarity field may be disrupted. A divergent polarity field may be established, with proximally oriented polarities towards the base of the tissue, and distally oriented polarities towards the tip (*Figure 14D*). This pattern emerges because cells with sufficiently elevated rates of auxin synthesis at the leaf base acquire increased intracellular auxin concentrations, causing polarities to orient towards them. However, it is possible that auxin biosynthesis in regions of *YUC1* expression does not contribute sufficiently to influence polarity.

Compared to wild-type leaves, *kan1kan2* mutants tended to have a broader distribution of *YUC1::GUS* in the proximal half of young leaf primordia (*Figure 15A i*). Transverse sections through these primordia revealed that the expression domain extended further along the proximodistal axis on the abaxial side of the leaf (the region where centres of convergence form) than on the adaxial side (*Figure 15A ii and iii*). This is consistent with the tandem alignment models for convergence formation, where a broad domain of auxin biosynthesis in the proximal region of the leaf induces cells with elevated auxin import and removal, causing convergence formation (*Figure 10*). As in WT, *YUC1::GUS* expression in the basal-most cells of the leaf persisted and could still be detected in leaves of around 500 µm in width (*Figure 15B iv*). Consistent with a role of auxin biosynthesis in inducing cells with elevated auxin import (*Figure 10*), the majority (98%) of *kan1kan2yuc1yuc4* mutant rosette leaves failed to form outgrowths and lacked sites of elevated *LAX1::GUS* expression within the lamina (*Figure 15C*, compare with *Figure 11B*).

Before *kan1kan2* outgrowths were detected, *YUC1::GUS* was expressed in an additional band, a few cells wide, running across the abaxial side of the leaf, approximately one-third of the way from the leaf base (*Figure 15B ii*, black arrow). As outgrowths emerged, the band of *YUC1::GUS* expression was present at their base on their distal side (the outgrowth axil) (*Figure 15B iii and iv*, black arrow) and absent from the outgrowth tips (*Figure 15B iii*, black arrow head).

The expression pattern of *YUC1::GUS* is similar to that of *CUC2::RFP* (*Figure 6*). To investigate whether *CUC2* functions upstream of *YUC1* expression in *kan1kan2* leaves, we generated *kan1kan2-cuc2* mutants with the *YUC1::GUS* reporter. In young *kan1kan2cuc2* leaves, the proximal domain of *YUC1::GUS* expression was reduced in size (*Figure 15D i*) and at later stages of leaf development the ectopic band of *YUC1::GUS* expression was lost (*Figure 15D ii*). Thus, *CUC2* is required for the ectopic expression of *YUC1::GUS* in *kan1kan2* leaves.

To test the potential effects of the pattern of *YUC1::GUS* expression close to outgrowths on PIN1 polarity patterns, we introduced a pre-pattern of auxin biosynthesis similar to that observed for *YUC1* into simulations of the different models. In tandem alignment models, a band of cells with elevated auxin production introduced after the establishment of a proximodistal polarity field generates a region of divergent polarities centred on the band with elevated production (*Figure 16A,C*). If the band of elevated auxin biosynthesis is introduced together with a cell with elevated auxin

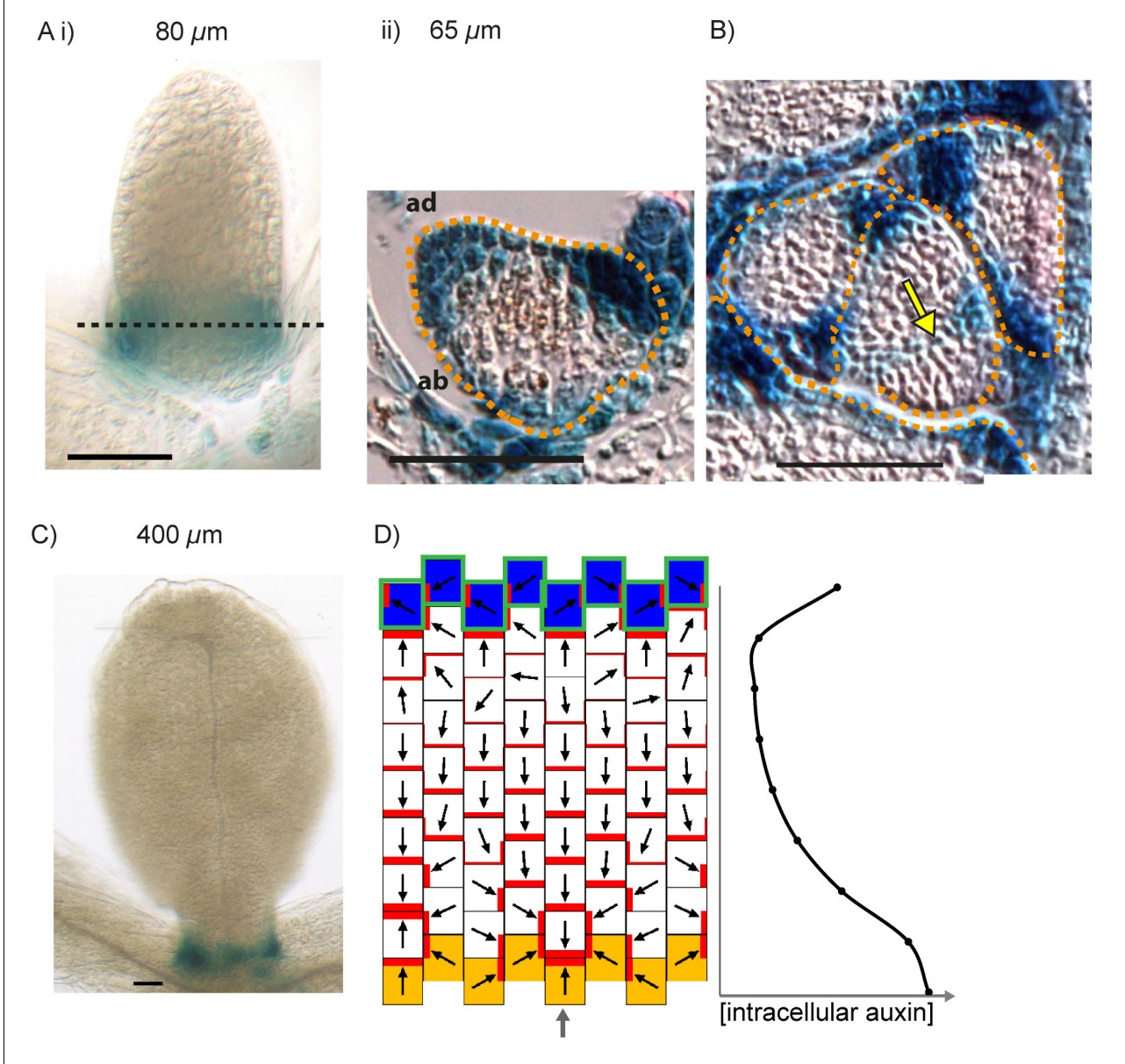

**Figure 14.** *YUCCA1::GUS* expression in WT leaves. (**A**) Expression of *YUC1::GUS* in a WT leaf 1 primordium. **i**) abaxial epidermis (10 plants were imaged across two separate experiments) **ii**) transverse cross-sectional view of leaf 1 at a similar developmental stage (adaxial (ad) is towards the top) at the position marked by the dashed black line in **i**). Dashed orange line indicates leaf outline. Data consistent with serial sections taken through 3 other leaves. (**B**) Transverse cross-sectional view of a *YUC1::GUS* expressing wild-type vegetative meristem. Yellow arrow with black outline indicates region of *YUC1::GUS* expression at the boundary of an emerging primordium. Dashed orange lines indicate leaf primordia outlines. Data consistent with sections from 3 meristems, across two separate experiments. (**C**) Expression of *YUC1::GUS* at a later stage of WT leaf development. Scale bars = 50 μm. 9 leaves imaged across two separate experiments. (**D**) In the presence of elevated auxin production in the proximal row of cells (orange), and elevated auxin import and removal in the distal row of cells (blue cells with green outline), the convergent alignment model generates a divergent polarity field, with proximally oriented polarity in the proximal half of the tissue, and distally oriented polarity in the distal half.

import and removal located on the proximal side of the band, a polarity convergence tends to form centred on the cell with elevated import and removal (*Figure 16B,D*). This behaviour is consistent with the finding that centres of PIN1 polarity convergence are located at the tips of outgrowths, where auxin importer expression is elevated.

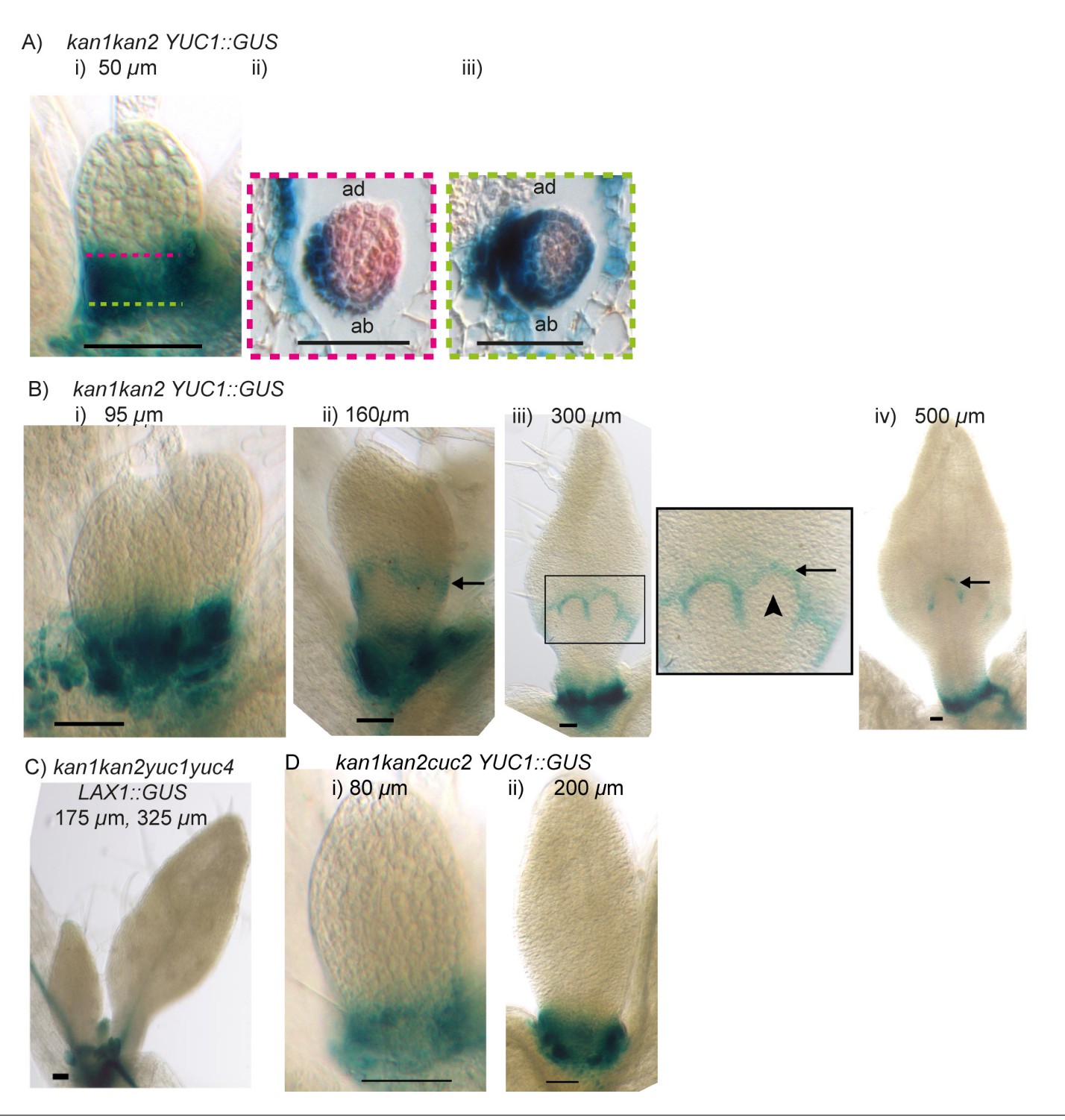

**Figure 15.** Expression of *YUCCA1::GUS* in *kan1kan2* leaves. (**A**) Expression of *YUC1::GUS* in *kan1kan2* leaf one primordia. **i**) abaxial epidermis (a total of 15 leaves imaged, in three separate experiments), **ii**) transverse cross-sectional view of leaf 1 at a similar developmental stage, at an approximate position along the proximo-distal axis marked with the pink dashed line in **i**, **iii**) transverse cross section through the same leaf as shown in **ii**), at the approximate position marked with the green dashed line in **i**. Data consistent with serial sections for 3 other *kan1kan2* leaves. (**B**) Expression of *YUC1::GUS* in *kan1kan2* mutant leaf 1 primordia at progressive stages of development. Black arrows indicate the band of *YUC1::GUS* expression, which is at the distal base of outgrowths in **iii**) and **iv**). Arrow heads indicate an outgrowth tip which does not express *YUC1::GUS*. At least 8 leaves were imaged for each developmental stage, across two experiments. (**C**) Expression of *LAX1::GUS* in first two leaves of a *kan1kan2yuc1yuc4* seedling (a total of 20

*Figure 15 continued on next page*

Figure 15 continued

leaves were imaged in three separate experiments). (**D**) Expression of *YUC1::GUS* in leaf one of *kan1kan2cuc2* mutant seedlings (15 seedlings were imaged at each developmental stage across three separate experiments). Scale bars = 50 μm.

In contrast to the tandem alignment models, when a band of elevated auxin synthesis is introduced in the convergent alignment model, polarities tend to orient towards the band (*Figure 16E*). This may occur even in the presence of a cell with locally elevated auxin import and removal on the proximal side of the band if the rate of auxin import in this cell is not sufficiently high (*Figure 16F*). This behaviour is inconsistent with the observation that PIN1 polarities orient towards regions with elevated *LAX1* and *AUX1* expression, even in the presence of a band with elevated *YUC1* expression in the outgrowth axil.

According to these simulations, the patterns of *YUC1*-mediated auxin biosynthesis close to *kan1-kan2* outgrowths are more compatible with the tandem rather than convergent alignment models. However, the presence of YUC expression in regions of polarity divergence at the leaf base and outgrowth axil may be reconciled with the convergent alignment model if it is assumed that YUC expression appears as a consequence of low auxin in these regions following polarity reorientation. Consistent with this possibility, the expression of *YUC1, YUC2, YUC4* and *YUC6* has been shown to be down-regulated by auxin, and upregulated by expression of an auxin biosynthesis inhibitor (*Suzuki et al., 2015*). Given such negative feedback acting upon *YUC* expression, it is possible that local auxin-biosynthesis by *YUC*s does not raise auxin concentrations sufficiently to affect PIN1 polarities.

*YUC2::GUS* and *YUC4::GUS* were previously reported to be expressed at the tips of wild-type leaves, in contrast to expression of *YUC1::GUS* which is expressed at the base (*Wang et al., 2011*). This expression pattern of *YUC2::GUS* and *YUC4::GUS* is more consistent with the convergent alignment model. However, at early stages of leaf development, when *YUC1::GUS* is expressed at the leaf base, *YUC4::GUS* and *YUC2::GUS* were found to be absent from the leaf tip (*Wang et al., 2011*). The proximo-distal PIN1 polarity pattern was only observed in young WT primordia (*Figure 1A*, *Figure 2A*). This suggests that, at the time that the proximodistal PIN1 polarity field is observed, the leaf base is the main site of *YUC*-mediated auxin biosynthesis.

*YUC2::GUS* and *YUC4::GUS* were also previously reported to be expressed at the tips of *kan1-kan2* outgrowths (whether this expression domain is epidermal or sub-epidermal was not shown). However, whether expression of these genes in regions of polarity convergence precedes outgrowths or appears later was not determined. We therefore performed time-lapse imaging of a *YUC4::GFP* reporter to capture the dynamics of *YUC4* expression (*Figure 17*). This revealed that, approximately one day prior to outgrowth emergence, when centres of PIN1 convergence were previously determined to form, this reporter was expressed in the epidermis, distal to where outgrowths subsequently emerged (*Figure 17 i,ii*). As outgrowths emerged, they maintained epidermal expression of *YUC4* in their axils, the region where *YUC1* is also expressed (*Figure 17 iii,iv*, compare with *Figure 15B ii–iv*). Emerging outgrowths also showed a second region of expression of *YUC4::GFP* in epidermal and sub-epidermal tissue at the outgrowth tip (*Figure 17 iii,iv*). These findings suggest that, at the time that centres of PIN1 polarity convergence form, *YUC4* is expressed in the epidermis distal to the centre of convergence. *YUC4* expression at the outgrowth tip then appears after convergence formation. The epidermal expression domain distal to polarity convergences in the outgrowth axil is similar to that inferred for *YUC1* and is more consistent with tandem than convergent alignment models. Similar to *YUC1*, expression of *YUC4::GFP* within the abaxial lamina was lost in the *kan1kan2cuc2* background (*Figure 17B*).

## Discussion

*kan1kan2* outgrowths arise ectopically as a consequence of reduced abaxial identity, and thus differ from leaf primordia and serrations. Despite these differences, we show that general elements of outgrowth formation apply to shoot outgrowths (*kan1kan2* outgrowths, primordia and serrations) regardless of their specific context. 1) Similar to leaf primordia and serrations (*Bilsborough et al., 2011*; *Hay et al., 2006*; *Reinhardt et al., 2003*; *Scarpella et al., 2006*), *kan1kan2* outgrowths are

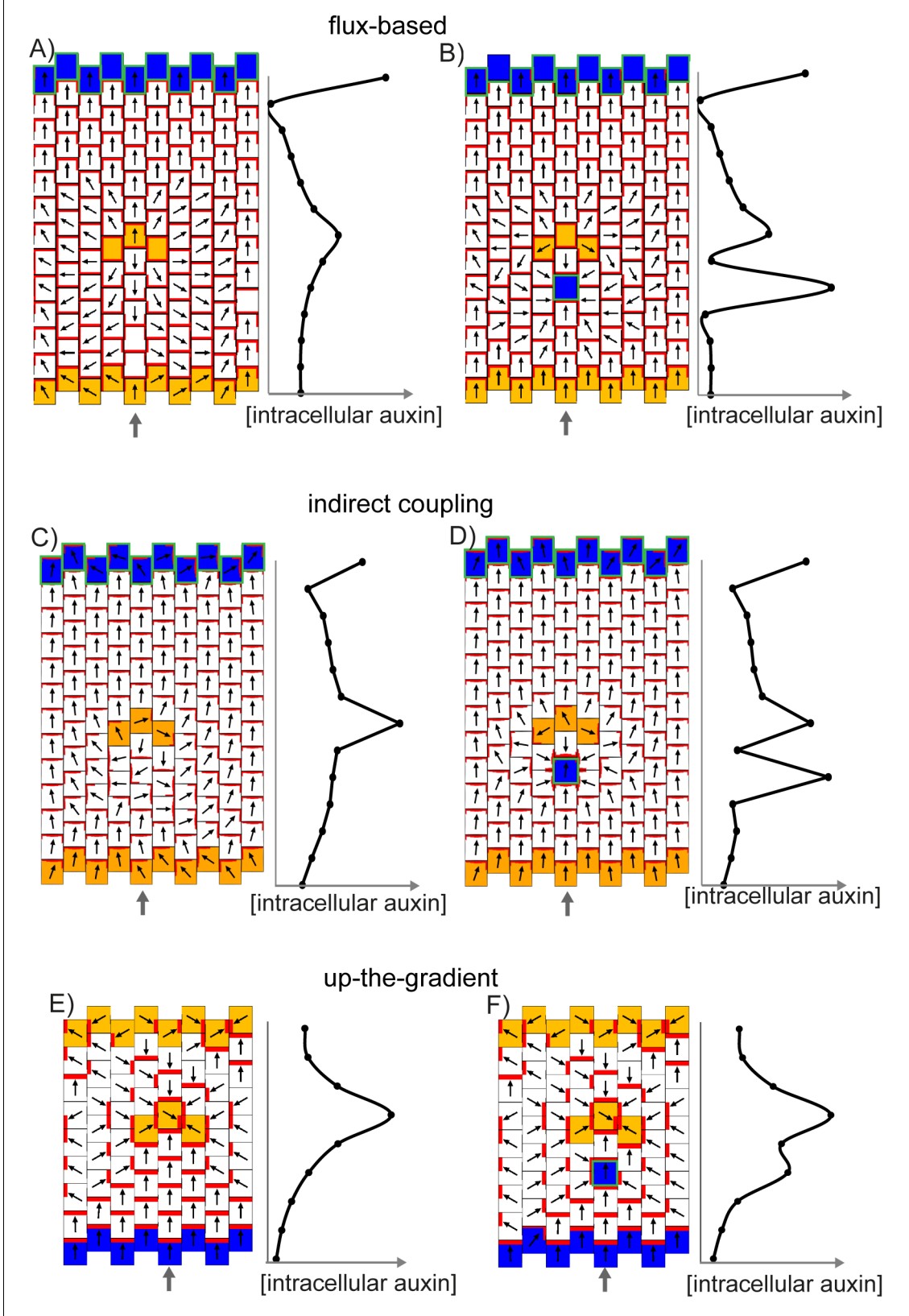

**Figure 16.** Effect of a band of locally elevated auxin biosynthesis on flux-based, indirect coupling and up-the-gradient models. (**A**) Flux-based model. A proximodistal polarity field is initially established due to elevated auxin production at the leaf base (orange cells) and elevated auxin import and

*Figure 16 continued on next page*

*Figure 16 continued*
removal at the leaf tip (blue cells with green outline). After distally oriented polarities are established, 3 cells are given an elevated rate of auxin biosynthesis (orange cells), causing polarities to diverge away from this region. Graph shows intracellular auxin concentrations for the column of cells marked with the grey arrow. (B) As for A, but with a cell with elevated auxin import and removal (blue cell with green outline) on the proximal side of the band with elevated auxin synthesis. (C) As for A, but for the indirect coupling model. (D) As for B, but for the indirect coupling model. (E) Up the gradient model. A proximo-distal polarity field is initially established through elevated auxin removal at the leaf base (blue cells) and an initially elevated auxin concentration at the leaf tip (same initial conditions as *Figure 8A*). Three cells (isolated orange cells) are then given an elevated auxin biosynthesis rate, causing polarities to orient towards them. (F) As for E, but a cell with elevated auxin import and removal is also added on the proximal side of the band with elevated auxin synthesis.

preceded by epidermal centres of PIN1 polarity convergence. By tracking cells, we show that these centres go on to form outgrowth tips, similar to results from tracking emerging floral primordia (*Heisler et al., 2005*). Thus, as the outgrowth develops, the centre of convergence corresponds to the distal end of a proximodistal field. 2) Like primordia and serrations (*Bilsborough et al., 2011*; *Heisler et al., 2005*), *kan1kan2* outgrowths are preceded by locally elevated intracellular auxin. 3) Expression of *YUCCA* auxin biosynthesis enzymes is elevated in the boundaries that separates leaves, serrations and *kan1kan2* outgrowths from the tissues that they emerged from. The axillary boundary (located in the axils of primordia and distal sides of serrations and *kan1kan2* outgrowths) has divergent PIN1 polarity orientations (*Hay et al., 2006*; *Heisler et al., 2005*; *Wang et al., 2011*). In the case of *YUC4*, we show that axillary expression in *kan1kan2* outgrowths precedes outgrowth emergence. 4) Centres of polarity convergence at the tips of leaves (*Bainbridge et al., 2008*), serrations (*Kasprzewska et al., 2015*) and *kan1kan2* outgrowths are associated with regions of elevated epidermal expression of the auxin importer, *LAX1*. 5) Epidermal centres of PIN1 polarity convergence are coupled to sub-epidermal strands of cells with elevated PIN1 expression and inwardly oriented polarity (*Bayer et al., 2009*; *Hay et al., 2006*; *Reinhardt et al., 2003*). 6) Like primordia and serrations (*Nikovics et al., 2006*; *Aida et al 1997*), formation of outgrowths is promoted by *CUC* genes which become strongly expressed distal to the outgrowth. We show that in *kan1kan2* leaves, CUC2 promotes formation of PIN1 convergence sites and both elevated YUCCA and auxin importer expression.

Taken together, these results support the hypothesis that a common module is associated with the development of outgrowths. This outgrowth module operates iteratively, first in the creation of the leaf with a proximodistal polarity field, and then in the creation of leaf outgrowths such as serration and leaflets (*Hay et al.,2006*; *Bilsborough et al., 2011*), or ectopic *kan1kan2* outgrowths. Although centres of convergence associated with elevated intracellular auxin and sub-epidermal PIN1 strands were previously shown to be associated with primordia and serrations, here we extend the notion of the outgrowth module to include elevated auxin biosynthesis and auxin import (features 3, 4 and 6 above) and show that it also applies to outgrowths emerging from the plane of the leaf. The observation that a loss of PIN1 convergences and outgrowths in *kan1kan2cuc2* mutants is correlated with a loss of ectopic auxin biosynthesis and import supports an important role of these features in the outgrowth module.

These observations raise the question of how the various features of the outgrowth module are generated. We found that models for PIN1 convergence site formation (feature 1) fall into two groups. Convergent alignment models (e.g. up-the-gradient) spontaneously generate spaced centres of polarity convergence in a in a field of cells with initially uniform auxin distribution (*Figure 7D*, *Jönsson et al., 2006*; *Smith et al., 2006*; *Cieslak et al., 2015*; *Heisler et al., 2010*). By contrast, tandem alignment models (e.g. flux-based and indirect coupling) generate swirled patterns of tandemly aligned polarity in this context (*Figure 7E,F*).

Convergent alignment models provide the most parsimonious explanation for the observed PIN1 polarity (feature 1), intracellular auxin (feature 2), and spacing of outgrowths, as all of these aspects arise naturally in a uniform field (*Bilsborough et al., 2011*; *Jönsson et al., 2006*; *Smith et al., 2006*). However, there is no expectation of elevated expression of auxin biosynthesis enzymes in the regions of polarity divergence in the axils of leaves and their outgrowths (feature 3). Indeed, auxin biosynthesis in these regions can cause polarities to orient towards regions of elevated auxin biosynthesis, rather than away from them as is observed experimentally (*Hay et al., 2006*; *Heisler et al.,*

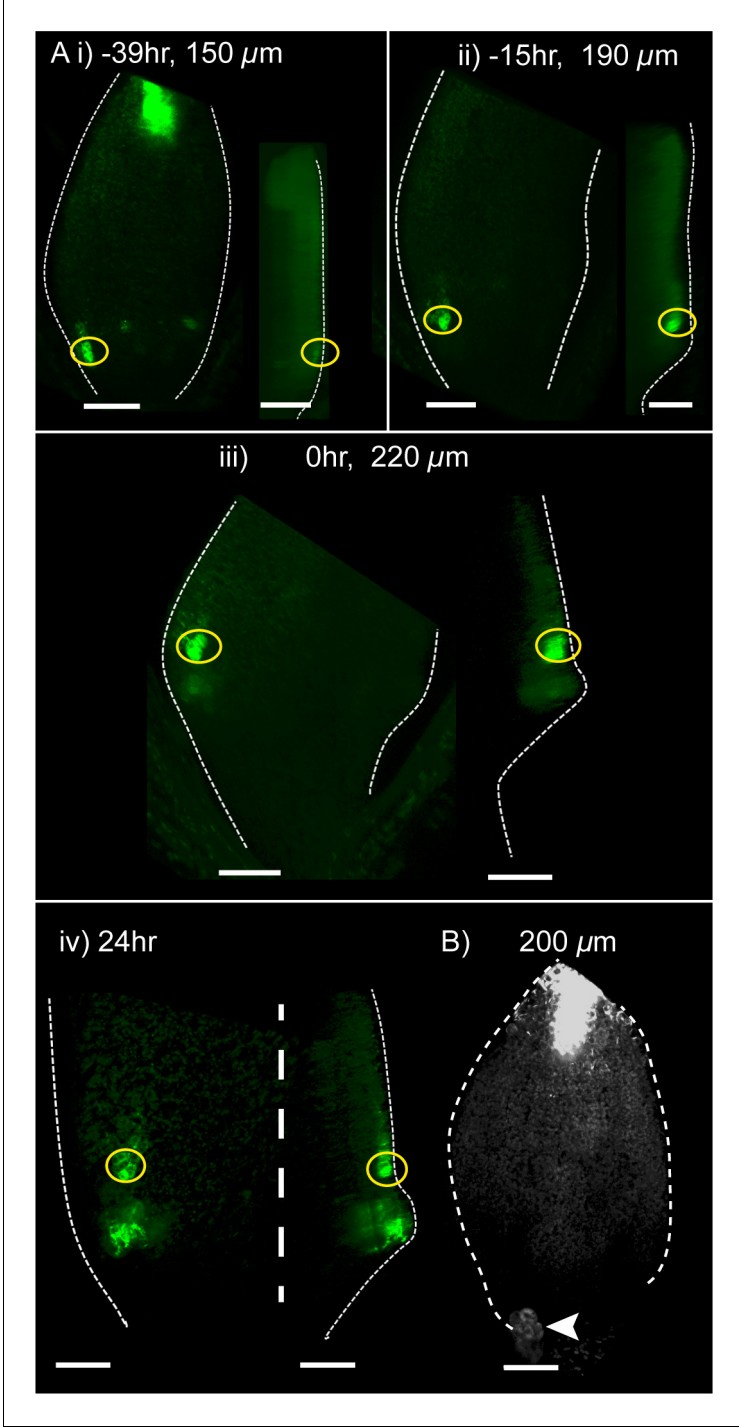

**Figure 17.** Time-lapse imaging of *YUC4::GFP* in *kan1kan2* leaves. (**A**) Confocal images of *YUC4::GFP* in the same *kan1kan2* leaf imaged over a period of 3 days. Times relative to outgrowth emergence, and leaf widths, are given above images. Yellow circles indicate cells located distal to the outgrowth after its emergence, and their ancestors prior to outgrowth emergence. Note that these cells express *YUC4::GFP* before and after outgrowth emergence, but the region of expression at the outgrowth tip only appears following outgrowth emergence. Data is representative of tracking 5 out of 5 *kan1kan2* leaves that developed ectopic outgrowths across two experiments. (**B**) *YUC4::GFP* expression in leaf one of a *kan1kan2cuc2* mutant. Data representative of 13 leaves imaged in two separate experiments. Arrow head indicates stipule. Scale bars = 50 μm.

*2005*; *Wang et al., 2011*). The convergent alignment model may be reconciled with the observed expression of *YUC1* and *YUC4* auxin biosynthesis genes in regions of polarity divergence if it is assumed that this expression does not raise auxin levels sufficiently to influence polarity. Rather than playing a role in polarity reorientation, *YUC* expression may then be a consequence of low auxin in these regions following polarity reorientation, as expression of *YUC* genes has been shown to be down-regulated by auxin (*Suzuki et al., 2015*).

There is also no expectation of elevated auxin import in regions of polarity convergence (feature 4) for the convergent alignment model. The parsimony of the convergent alignment model arises from the reinforcement of auxin levels at centres of convergence through auxin transport without any need for additional mechanisms, such as elevated auxin import. Elevated auxin importer expression can be incorporated within a convergent alignment model, by postulating that high levels of intracellular auxin induce auxin import. This approach was adopted to account for the effect of auxin import on leaf serration development, and with suitable parameters auxin import can play a role in reinforcing convergent alignment (*Kasprzewska et al., 2015*).

However, with some convergent alignment models, incorporation of auxin import may have a disruptive effect. It has been proposed that cells may orient polarity up auxin gradients by responding to auxin levels in their neighbours indirectly by sensing auxin flux (*Cieslak et al., 2015*). Assuming all cells in a tissue have an equal capacity to import and export auxin, cells with elevated intracellular auxin concentration will have higher net rates of auxin flux away from them, into neighbouring cells. This flux could be sensed to orient PIN1 allocation up the auxin gradient. However, this behaviour would be disrupted by elevated levels of auxin importer expression at centres of convergence, because elevated auxin import in the high auxin cells would reduce the net flux from them into their neighbours.

Based on PIN1 data and intracellular auxin levels alone (features 1 and 2), tandem alignment models provide a less parsimonious explanation than convergent alignment models. Special mechanisms need to be postulated for tandem alignment models for PIN1 convergence sites to form, such as induction of regions with elevated auxin import and removal. The observation of sub-epidermal strands of cells with inwardly oriented PIN proteins (feature 5) fits the requirement of auxin removal for convergence formation in the tandem alignment models. However, auxin removal alone, through formation of sub-epidermal PIN1 strands (*Stoma et al., 2008*) would lead to a transient dip in epidermal auxin concentrations, which is not observed experimentally (*Brunoud et al., 2012*; *Heisler et al., 2005*). Here, in the simulations we present, the transient dip does not occur if centres of convergence have elevated auxin import in addition to removal. The observed elevated levels of auxin import at convergence centres (feature 4) is therefore an expected feature for tandem alignment models. A functional role for auxin import is also supported by the observed disruption of outgrowth initiation in *aux1lax1lax2lax3* meristems (*Bainbridge et al., 2008*) and *kan1kan2aux1lax1lax2lax3* leaves. A further property of tandem alignment models is that polarities tend to become oriented away from auxin sources. It is therefore expected that sites of divergent polarity, such as the axillary regions of outgrowths would be auxin sources, compatible with elevated expression of auxin biosynthesis enzymes in these regions (feature 3). Also supporting these predictions of tandem alignment models, loss of ectopic auxin biosynthesis and import in the *kan1kan2-cuc2* mutant correlates with a loss of PIN1 convergence points and ectopic outgrowths.

Thus, when the auxin import and biosynthesis data are considered along with the PIN1 polarity and intracellular auxin levels, tandem alignment models may give a more a parsimonious explanation for the data as a whole.

The contribution of *CUC* gene activity (feature 6) can be incorporated into both convergent and tandem alignment models. Our data suggest that in wild-type leaves *CUC2* expression is excluded from the abaxial lamina and confined to serrations by the action of abaxial identity determinants including *KAN1* and *KAN2*. According to the convergent alignment model, CUC2 may be required to enable PIN1 polarity to become reoriented rather than remaining fixed in the abaxial lamina as occurs in wild type (*Bilsborough et al., 2011*). According to the tandem alignment models, ectopic expression of *CUC2* could drive auxin synthesis (YUCCA expression) and induction of ectopic sites of elevated auxin import and removal, and thus centres of convergence. *CUC2* expression distal to the outgrowth could also maintain these regions as sites of high auxin synthesis and polarity divergence.

The results presented here highlight distinctive elements in models for tissue cell polarity in plants and animals. The ability of all three plant models analysed to generate cell polarity in a non-polarising context (*Figure 7A–C*) contrasts with models of animal planar cell polarity (*Strutt and Strutt, 2009*), where polarity is proposed to either be dependent on pre-established asymmetric cues (*Amonlirdviman, 2005*; *Fischer et al., 2013*; *Le Garrec et al., 2006*) or polarisable neighbours (*Burak and Shraiman, 2009*). Models may therefore be classified into two broad categories: those in which polarity of a cell depends on a polarising environment and those that do not (*Figure 18*). Although most animal models belong to the first category, a version that does not require a polarising environment, termed direct coupling, has been proposed (*Abley et al., 2013*).

For plant models, cells may polarise in a non-polarising environment in two ways. One is through small fluctuations in polarity leading to altered auxin distribution or flux which then feeds back to enhance polarity. This occurs in the up-the-gradient and flux-based models (*Figure 7A,B* [*van Berkel et al., 2013*]). It contrasts with current animal models in which polarity does not feed back to influence the distribution of the extracellular ligand. Another class of plant models proposes that cells become polarised without requirement for altered distribution or flux of auxin, through processes internal to the cell, as assumed for indirect coupling (*Figure 7C*). It remains to be seen whether the indirect coupling model can also capture the generation of sub-epidermal strands of cells with elevated levels of membrane-localised PIN1 (feature 5). Such strands are observed experimentally during pro-vascular development and can be accounted for with a flux-based model assuming superlinear feedback between flux and PIN1 allocation (*Scarpella et al., 2006*; *Bayer et al., 2009*; *Rolland-Lagan and Prusinkiewicz, 2005*; *Feugier et al., 2005*; *Stoma et al., 2008*).

Our analysis illustrates the value of comparing multiple models when analysing experimental data rather than focusing on one in isolation. Each model alone can be modified to account for experimental observations, showing that fitting to data is not conclusive. However, models may differ in

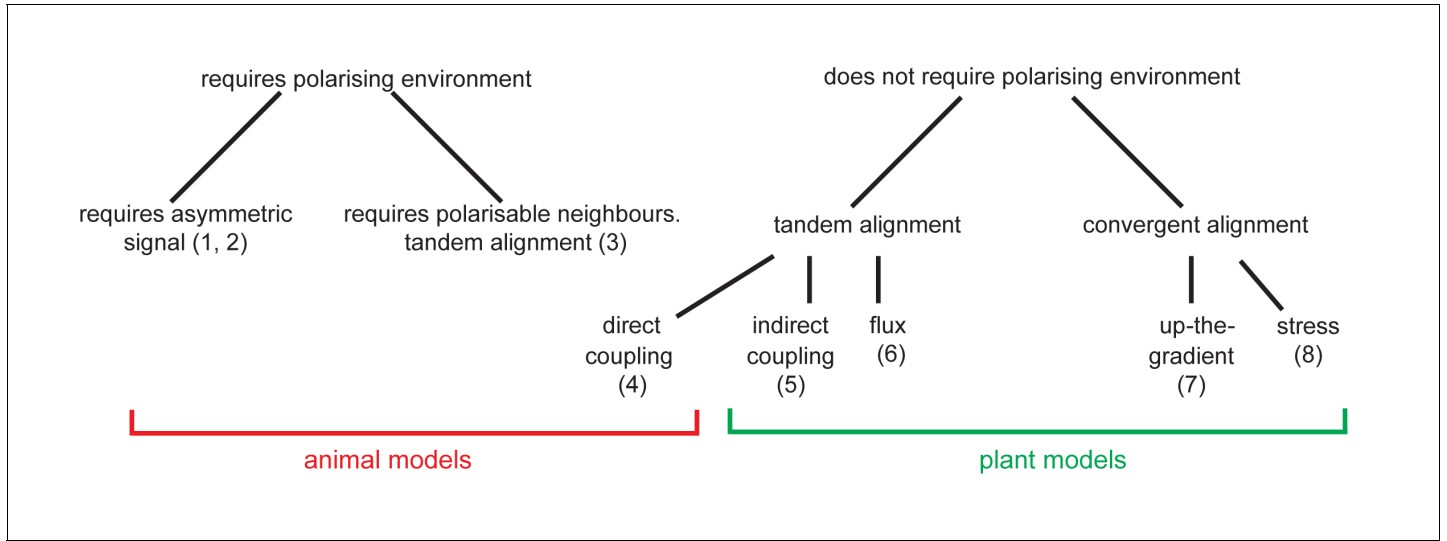

**Figure 18.** Classification of plant and animal polarity models. Models of plant and animal epidermal polarity may be classified according to whether they require the presence of pre-established asymmetries or polarisable neighbours (referred to as a polarising environment) for cell polarity to arise. All plant models tested can generate cell polarity without a polarising environment, whilst animal models except the direct coupling model ([4] [*Abley et al., 2013*]) either require an asymmetric signal ([1] [*Le Garrec et al., 2006*] [2] [*Amonlirdviman, 2005*]) or polarisable neighbours ([3] [*Burak and Shraiman, 2009*]) for cell polarity to arise. A further classification is made according to the behaviours of groups of cells in the absence of pre-established asymmetries. In this context, the indirect coupling ([5] [*Abley et al., 2013*]) and flux-based ([6] [*Mitchison, 1980*; *Rolland-Lagan and Prusinkiewicz, 2005*; *Stoma et al., 2008*]) models of plant polarity generate swirled patterns of polarity with tandem alignments between cells. In contrast, the up-the-gradient model ([7] [*Bilsborough et al., 2011*; *Jönsson et al., 2006*; *Smith et al., 2006*]) generates convergent alignments of cell polarities. The stress-based model of PIN1 polarity ([8] [*Heisler et al., 2010*]) also fits in the convergent alignment category as it generates similar behaviours with respect to auxin concentrations to the up-the-gradient model. The animal models that are capable of polarising in the absence of pre-established asymmetries generate tandem alignments of polarities ([4] direct coupling model [*Abley et al., 2013*], and (3) *Burak and Shraiman, 2009*).

the predictions they make prior to the data being obtained, and the parsimony with which experimental data can be accounted for. Moreover some models may make very similar predictions even though underlying molecular mechanisms are different. For example, two types of tandem alignment model - flux-based and indirect coupling - give similar predictions for auxin import and synthesis. Similarly, a concentration-based or stress-based version of the convergent alignment model gives similar predictions for auxin turnover. Other types of experiment will be needed to discriminate between these further levels of explanation.

## Materials and methods

### Plant material and growth conditions

All *kan1kan2* mutants carry *kan1-2* and *kan2-1* alleles in the Landsberg erecta (Ler-0) background (*Eshed et al., 2001*) except in the *kan1kan2yuc1yuc4* background, where plants carry *kan1-11* and *kan2-5* alleles (*Wang et al., 2011*). The *cuc2* mutant allele used is *cuc2-3* and was initially in the Columbia (Col-0) background (*Hibara et al., 2006*). All marker lines were originally in the Col-0 background. *PIN1::PIN1:GFP, DR5::GFP* (*Benková et al., 2003*); *LAX1::GUS, LAX2::GUS, LAX3::GUS* (*Bainbridge et al., 2008*); *YUC1::GUS, YUC2::GUS, YUC4::GUS* (*Cheng et al., 2006*) were described previously. The *AUX1::AUX1:YFP* line corresponds to the AUX1-YFP116 construct described by (*Swarup et al., 2004*). The *CUC2::RFP* line was kindly provided by Patrick Laufs prior to publication.

The *YUC4::GFP* reporter was provided by Yunde Zhao and Youfa Cheng. The *YUC4* promoter was amplified with the following primers: Y4pro3p_XmaI: cccggggtcgactaataaaagcgaaagagag and Y4pro5p_HindIII: aagcttatgtccaacatgcatgcg. The 2916 bp PCR product was digested with HindIII and XmaI, and cloned into pBJ36-ERGFP vector. The promoter was subcloned from pBJ36-ERGFP into the pART27 vector and transformed into Col-0 background by floral dipping and subsequently selecting for kanamycin resistant transgenic plants.

All images of marker construct expression in WT were obtained using the original marker lines in the Col-0 background. The *kan1kan2cuc2* mutant was generated by crossing *cuc2* mutants with *kan1kan2+/-* plants (*kan1kan2* mutants are sterile). In the F2 of this cross, *kan1kan2+/- cuc2* individuals were identified by screening for the *kan1kan2+/-* abberent fruit phenotype, and by PCR-based screening for *cuc2-3* homozygous individuals. The presence of a T-DNA insertion in the *CUC2* locus was detected by using a forward primer, which binds to the T-DNA: 5'-TCCATAACCAATCTCGA TACAC-3' and a reverse primer which binds within the *CUC2* locus downstream of the T-DNA insertion (referred to as CUC2-R): 5'- GGAGGCTAAAGAAGTACCATTC-3'. The presence of a WT copy of *CUC2* was detected using a forward primer CUC2-F: 5'-AATATCCATCCACATTATTACCAC-3', which binds to *CUC2* upstream of the site of the T-DNA insertion in *cuc2-3*, along with CUC2-R. One quarter of the offspring of *kan1+/- kan2 cuc2* individuals were *kan1kan2cuc2* mutants, and could be identified at the seedling stage (and therefore selected for experiments) based on the upwardly curled cotyledon phenotype of *kan1kan2* mutants.

Marker constructs were introduced into the *kan1kan2* and *kan1kan2cuc2* backgrounds by carrying out the corresponding crosses using *kan1kan2+/-* or *kan1kan2+/-cuc2* mutants. The *kan1kan2aux1-lax1lax2lax3, kan1kan2aux1lax1lax2* and *kan1kan2aux1lax1* mutants were generated by crossing *kan1kan2+/-* plants with the *aux1lax1lax2lax3* quadruple mutant (*Bainbridge et al., 2008*). In segregating F2s from this cross, *kan1kan2+/-* plants with desired mutant alleles of aux/lax genes were selected as described in *Bainbridge et al., 2008*.

The *kan1kan2+/-* line used to generate *the kan1kan2aux1lax1lax2lax3* mutants was in the Ler-0 background, and the *aux/lax* quadruple mutant was in the Col-0 background. To check that the loss-of-outgrowth phenotype in hextuple mutants was not due to the introduction of an unknown genetic element from the Col-0 background, we determined whether the reduction in the number of outgrowths segregated with the mutant *LAX1* allele. An F3 family of plants was used that segregated for mutations in *KAN2* and *LAX1*, but was homozygous for *kan1, aux1, lax2* and *lax3* mutant alleles. Of 156 *kan1kan2aux1lax2lax3* plants segregating for *lax1*, 120 (77%) developed many outgrowths and 36 (23%) had a reduced number and size of outgrowths. All of the plants with few outgrowths were homozygous for the *lax1* mutation, whilst all siblings with many outgrowths that were genotyped (40 in total) all had a wild-type copy of *LAX1*. Such segregation analyses were not carried out for *kan1kan2aux1lax1* and *kan1kan2aux1lax1lax2* mutants thus it is possible that the reduction in

outgrowths in these lines is due to introduction of an unknown genetic element during the cross between Col-0 and Ler backgrounds. However, two independent families of *kan1kan2aux1lax1lax2* plants (derived from different F2 plants) had similar losses of outgrowths (data provided in source data file for *Figure 11G*). The loss of outgrowths in *kan1kan2aux1lax1lax2lax3* plants was also observed in two independent families quantified in two different experiments (see source data file for *Figure 11G*).

Plants were sown on plates for confocal imaging experiments and to select those to be used for crosses. Seeds were surface-sterilized before plating on Petri dishes with solid Murashige and Skoog medium as described (*Sauret-Güeto et al., 2013*) and grown in controlled environment room at 20°C in long day conditions (8 hr dark and 16 hr light under fluorescent white light at a photon fluence rate of 100 µmol·m$^{-2}$·s$^{-1}$). When appropriate, medium was supplemented with Kanamycin 50 µg/ml (to select for *PIN1::PIN1:GFP, AUX1::AUX1:YFP, LAX1::GUS, LAX2::GUS, LAX3::GUS, YUC1:: GUS* and *YUC4::GFP* constructs).

*kan1kan2* plants carrying *aux/lax* or *cuc2* mutant alleles were sown on soil prior to characterisation of their phenotype. Seeds were stratified directly (without sterilisation) and then sown in John Innes Centre *A.thaliana* Soil Mix (Levington F2 compost with Intercept and grit at a 6:1 ratio) and were grown under long-day conditions (16 hr light and 8 hr dark) in a glasshouse, supplemented with artificial light, at approximately 22°C.

The *kan1kan2yuc1yuc4* mutants used are as described in (*Wang et al., 2011*). *kan1kan2+/-yuc1 +/-yuc4 LAX1::GUS* plants were selected in an F2 of a cross between *kan1kan2+/- yuc1+/- yuc4* plants and *LAX1::GUS* plants. F2 seedlings with the *LAX1::GUS* construct were by selected for sowing seeds on plates containing 50 µg/ ml Kanamycin. Those that were *kan1kan2+/-* mutants were identified based on their fruit phenotype and the presence of the *yuc1+/-yuc4* genotype was selected for by PCR-based genotyping. The *yuc1* and *yuc4* alleles have T-DNA insertions, and mutant alleles were screened for using the following primers for *yuc1*: LBb1: 5'-GCGTGGACCGC TTGCTGCAAC-3' and YUC1-R: 5'-CCTGAAGCCAAGTAGGCACGTT-3' and the following primers for *yuc4*: LBb1: 5'-GCGTGGACCGCTTGCTGCAAC-3' and YUC4-R: 5'-GCCCAACGTAGAATTAG-CAAG-3'. WT copies of *YUC1* were screened for using YUC1-F: 5'-GGTTCATGTGTTGCCAAGGGA-3' and YUC1-R and WT copies of *YUC4* were screened for using YUC4-F 5'-CCCTTCTTAGACCTAC TCTAC-3' and YUC4-R. To analyse the expression of *LAX1::GUS* in *kan1kan2yuc1yuc4* mutants, the offspring of *kan1kan2+/-yuc1+/-yuc4 LAX1::GUS* or *kan1kan2+/yuc1+/-yuc4 LAX1::GUS+/-* plants were sown on plates containing 50 µg/ ml Kanamycin and seedlings with a *kan1kan2* leaf phenotype (narrow, upwardly pointing leaves) were selected, genotyped for *YUC1* and *YUC4*, and stained to reveal GUS activity. Of these seedlings, one quarter did not have outgrowths, and genotyping showed they were *kan1kan2yuc1yuc4* mutants.

## Confocal and OPT imaging, propidium iodide and GUS staining

For tracking experiments, seedlings were first grown on plates (until 5 days after stratification for *kan1kan2* and *kan1kan2cuc2* mutants, and until 4 days after stratification for WT seedlings). Seven seedlings were then transferred into a tracking chamber (*Sauret-Güeto et al., 2013*; *Calder et al., 2015*) where they were kept for the duration of the imaging experiment. During this period, there was a constant flow of liquid media (1/4 strength Murashige and Skoog, 0.75% sucrose, 1.1 $\mu g$ / ml MES, pH 5.8) at 1 $\mu$l /s through the growth chamber. Seedlings in the chamber were imaged using a Zeiss EXCITER Laser Confocal Microscope every 6 to 24 hr (times are provided for specific experiments in figure legends). To image GFP and YFP markers, a 488-nm line of an argon ion laser was used. Emitted light was filtered through a 500–550-nm band-pass filter. To image RFP markers, a 543 nm helium-neon laser was used and emitted light was filtered using a 560–615-nm band-pass filter. A 40x oil objective was used for all experiments. Between imaging, the chamber containing seedlings was kept at 20°C, with 16 hr light, 8 hr dark cycles. In confocal imaging experiments without tracking, seedlings were mounted in water on microscope slides.

Confocal z-stacks were converted into individual PNG images using Bioformats converter (http://cmpdartsvr3.cmp.uea.ac.uk/wiki/BanghamLab/index.php/BioformatsConverter). Z-stacks of PNG images were then rendered in 3D using Volviewer (http://cmpdartsvr3.cmp.uea.ac.uk/wiki/Bangham-Lab/index.php/VolViewer#Download). All snapshots shown in figures were taken from Volviewer. Leaf widths were measured from maximum intensity projections in image J and thus do not take into account the 3D curvature of the leaf.

OPT imaging was performed as described by (*Lee et al., 2006*, *Sharpe et al., 2002*). Following PI staining (described below), plants were embedded in 1% low melting point agarose and embedded samples were kept in methanol overnight to dehydrate. To clear the agarose before scanning, the methanol was replaced with 2 parts (v/v) Benzyl Benzoate, 1 part Benzyl Alcohol and samples were left for 12 hr until almost transparent. A prototype OPT device was used to image embedded leaves (*Lee et al., 2006*; *Sharpe et al., 2002*). A 20-W halogen lamp connected to the OPT device was used to collect visible light transmission images. Images were reconstructed into png slices and visualised in 3D using *VolViewer*.

For propidium iodide staining, dissected leaves were placed into 10% (v/v) acetic acid, 50% (v/v) methanol solution and kept at 4°C overnight. Samples were then washed with water and dehydrated with two washes each in 40% (v/v) ethanol, 60% (v/v) ethanol and then 80% (v/v) ethanol. The samples were boiled in 80% (v/v) ethanol at 80°C for 2–10 min in a water bath and then rehydrated with two washes each in 60% (v/v) ethanol, 40% (v/v) ethanol, 20% (v/v) ethanol, and water. Samples were then treated with alfa-amylase [3 mg of alpha-amylase (SIGMA, UK) in 10 ml of 20 mM phosphate buffer (pH 7), 2 mM NaCl, 0.25 mM $Ca_2Cl$] overnight at 37°C, followed by three water washes, and then treated with 1% periodic acid [1% in solution from SIGMA, UK] (an oxidising agent) for 40 min at room temperature. Following two washes with water, PI staining solution [333 mM sodium metabisulphite, 0.5 M HCl, 148 $\mu$M PI] was added and incubated for 1–2 hr at room temperature until the material appeared pink in colour. PI staining solution was then removed and the samples were washed with water.

GUS staining with and without wax embedding was performed as described by *Sessions et al. (1999)*, but here using 6 mM Potassium Ferrocyanide and 6 mM Potassium Ferricyanide and with an 18 hr incubation with the GUS staining solution.

## Protein in situ localisation of PIN1

Immuno-localisation of PIN1 was performed as described (*Conti and Bradley, 2007*), using a goat anti-PIN1 (aP-20) polyclonal primary antibody (Santa Cruz Biotechnology, TX, USA) (RRID:AB_670528) diluted 1:300 and a Cy3 conjugated rabbit anti-goat secondary antibody (Jackson Immunoresearch, PA, USA), diluted 1 in 300.

## Model descriptions

All models were created using the VVe modelling environment, an extension of the VV system (*Smith et al., 2003*), which in turn is an extension of L-systems. All models are provided as supplementary source code files. Each model can be run using L-studio with VVe, which can be downloaded from http://algorithmicbotany.org/virtual_laboratory/. Instructions on how to run models are available as supplementary information (*Supplementary file 1*).

## Indirect coupling

The indirect coupling model was implemented based on the description in *Abley et al., 2013* except in some simulations we use rectangular rather than hexagonal cell geometries and we introduce an explicit representation of PIN1 to all models. The details of these differences are described below. All parameter values are as described in *Abley et al., 2013*, unless otherwise detailed in *Table 1*.

Each rectangular cell is represented by a single central cytoplasmic compartment, 22 membrane compartments (6 membrane compartments for each long edge and 5 membrane compartments for each short edge), and surrounding cell wall compartments. Each cell wall compartment links two membrane compartments located in adjacent cells.

The system is initialised with a default concentration of PIN1 in the cytoplasmic compartment of all cells. The rate of change of PIN1 concentration in a given membrane compartment depends on a background binding rate of PIN1 from the cytoplasm to the membrane, plus default unbinding from the membrane into the cytoplasm. It is also assumed that A* in a given membrane compartment promotes the binding of PIN1 to that membrane compartment and that PIN1 can diffuse between adjacent membrane compartments of the same cell. The equation describing the rate of change of PIN1 concentration in a given membrane compartment is:

**Table 1.** Parameter values used in simulations of the indirect coupling model.

| Symbol | description | unit | value |
|---|---|---|---|
| $\Delta t$ | numerical time step | s (seconds) | 0.05 |
| $R_c$ | area of cytoplasmic compartment | $\mu m^2$ | 260 |
| $R_w$ | area of cell wall compartment | $\mu m^2$ | 2.88 (for long cell edges); 3.0 (for short cell edges) |
| $l_n$ | length of membrane compartments | $\mu m$ | 2.88; 3.0 |
| $l_w$ | length of wall compartments | $\mu m$ | 2.88; 3.0 |
| $c_{PIN}$ | default initial concentrations of PIN1 in cytoplasmic compartments | $A_u/\mu m^2$ | 0.003 |
| $\rho_{PIN}$ | PIN1 default membrane binding rate | $\mu m/s$ | 0.03 |
| $\tau$ | A*-dependent promotion of PIN1 binding | $\mu m^2/A_u.s$ | 2 |
| $\mu_{PIN}$ | PIN1 default membrane unbinding rate | /s | 0.004 |
| $D_{PIN}$ | PIN1 diffusion in cell membrane | $\mu m^2/s$ | 0.1 |
| $\Psi_{PIN}$ | PIN-dependent active auxin efflux rate | $\mu m^2/A_u.s$ | 40 |
| $\varepsilon$ | limit for noise addition during initialisation of A* and B* concentrations | dimensionless | 0.0166 |
| $\gamma_{Aux}$ | Auxin-dependent promotion of A* to A conversion | $\mu m^2/A_u.s$ | 0.75* |
| $\rho_{Aux}$ | production rate of Auxin | $A_u/\mu m^2.s$ | $1.0 \times 10^{-4}$* |
| $\mu_{Aux}$ | degradation rate of Auxin | /s | 0.01* |
| $v_{in}$ | influx auxin permeability | $\mu m/s$ | 0.75* |

All parameters relate to either equations specified here or in **Abley et al., 2013**.

* In the simulations used to generate **Figure 7C** and **Figure 7—figure supplement 1C**, $\gamma_{Aux}$ = 0.9 $\mu m^2/A_u.s$, $\rho_{Aux}$ = 1.3 $\times$ 10$^{-4}$ $A_u/\mu m^2.s$ and $\mu_{Aux}$ = 0.02/s. Also, in some simulations, at tissue boundaries or centres of polarity convergence, auxin production, degradation and influx auxin permeability rates may vary from the background rates. Modulation of the influx auxin permeability rate is used to simulate elevated rates of auxin import. In **Figure 9C**, the auxin production rate in the bottom-most row of cells = 2 $\times$ 10$^{-4}$ $A_u/\mu m^2.s$, and the auxin degradation rate in the top most row of cells = 0.07/s. In **Figure 9D**, and at the beginning of the simulation used to generate **Figure 10C and D** the auxin production rate in the bottom-most row of cells = 2 $\times$ 10$^{-4}A_u/\mu m^2.s$. Throughout both simulations, in the top-most row, the auxin degradation rate = 0.015/s and the auxin influx permeability = 4.5 $\mu m/s$. In the simulation used to generate **Figure 10C and D**, changes in auxin production occur during the simulation and are described below. In this simulation, in the cell that forms the centre of convergence, the inwards permeation of auxin = 37.5 $\mu m/s$ and the auxin degradation rate = 0.15/s. In **Figure 16C and D**, the proximo-distal polarity field is established as described for **Figure 9D**. In the three cells with elevated auxin production, the auxin production rate is 1.4 $\times$ 10$^{-3}A_u/\mu m^2.s$. In the cell with elevated auxin import and removal (**Figure 16D**), the auxin degradation rate = 0.1 /s and the auxin influx permeability = 37.5 $\mu m/s$. In the simulations used to generate **Figure 10—figure supplement 1A and B**, parameter values are as for Figures 9D and 10D, respectively except $\Psi_{PIN}$ = 80 $\mu m^2/A_u.s$ and $\gamma_{Aux}$ = 0.85 $\mu m^2/A_u.s$. Here, c, the concentration of A* at which half of the PINs are activated = 0.22 $A_u/\mu m$ and the hill coefficient, h = 2.

$$\frac{\partial PIN}{\partial t} = (\rho_{PIN} + \tau A^*)PIN_c - \mu_{PIN}PIN + D_{PIN}\nabla^2 PIN \tag{1}$$

where *PIN* is the concentration of PIN1 in the focal membrane compartment, with units of Arbitrary units ($A_u$)/ $\mu m$ and $PIN_C$ is the concentration of PIN1 in the cytoplasmic compartment of the same cell as the focal membrane compartment, with units of $A_u/\mu m^2$. $\rho_{PIN}$ is the default binding rate of PIN1 to the membrane, with units of $\mu m/s$, and $\tau$ is a constant describing the rate at which membrane-bound A* promotes the binding of PIN1 to the membrane, with units of $\mu m^2/A_u.s$. A* is the concentration of the A* polarity component in the focal membrane compartment, with units of $A_u/\mu m$. $\mu_{PIN}$ is the default unbinding rate of PIN1 from the membrane into the adjacent cytoplasmic compartment, with units of /s. $D_{PIN}$ is the diffusion constant of PIN1 in the membrane, with units of $\mu m^2/s$.

The corresponding equation describing the rate of change of PIN1 concentration for a given cytoplasmic compartment is:

**Table 2.** Parameter values used in simulations of the flux-based model.

| Symbol | description | unit | value |
|---|---|---|---|
| $\Delta t$ | numerical time step | s (seconds) | 0.01 |
| $R$ | area of cytoplasmic compartment | $\mu m^2$ | 260 |
| $l$ | length of cell edge compartment | $\mu m$ | 15 for short cells edges; 8.6 for the each of the two compartments of a long edge * |
| $C_A$ | default initial concentration of auxin in cytoplasmic compartments | $A_u/\mu m^2$ | 0.01 |
| $C_{PIN}$ | default initial concentration of PIN1 at cell edge compartments | $A_u/\mu m$ | 0.01[†] |
| $\varepsilon_{PIN}$ | limit for noise addition during initialisation of PIN1 concentrations at cell edges | dimensionless | 0.025** |
| $\varepsilon_{Aux}$ | limit for noise addition during initialisation of Auxin concentrations | dimensionless | 0.025[‡] |
| T | PIN-dependent active auxin efflux rate | $\mu m^2/A_u.s$ | 1 |
| $\gamma$ | Unbinding rate of PIN1 from the cell edge | /s | 0.1 |
| $\rho$ | production rate of Auxin | $A_u/\mu m^2.s$ | 0.0001[§] |
| $\mu$ | degradation rate of Auxin | /s | 0.02 |
| $\alpha$ | Flux-dependent promotion of PIN1 allocation to a cell edge | dimensionless | 1[#] |
| D | Passive permeation rate of auxin | $\mu m/s$ | 5 |
| $P_{max}$ | Maximum concentration of PIN1 at a cell edge | $A_u/\mu m$ | 0.01[¶] |
| I | Auxin import rate | $\mu m/s$ | 0 |

* For cells with hexagonal geometries, each cell edge has a length of 10 $\mu m$.

[†] Value given is that used in simulations used to generate **Figure 7.B** and **Figure 7—figure supplement 1B**. In all other simulations, $C_{PIN} = 0$.

[‡] Values apply to simulations used to generate **Figure 7B** ($\varepsilon_{PIN}$) and **Figure 7—figure supplement 1B**, and **Figure 7E** ($\varepsilon_{Aux}$). All other simulations are initialised without noise addition.

[§] The value given for the auxin production rate applies to simulations used to generate **Figures 9A,B**, **10A, B** and **16A,B**. In the simulation used to generate **Figure 7B** and **Figure 7—figure supplement 1B**, $\rho = 0.0025\ A_u/\mu m^2.s$ and in the simulation used to generate **Figure 7E**, $\rho = 0.002\ A_u/\mu m^2.s$.

[#] The value given for, $\alpha$, the flux-dependent promotion of PIN allocation to cell edges, applies to simulations used to generate **Figures 9A,B**, **10A, B** and **16A,B**. In the simulation used to generate **Figure 7B**, $\alpha = 4 \times 10^{-3}$ and in the simulation used to generate **Figure 7E**, $\alpha = 3.2 \times 10^{-3}$.

[¶] In the simulation used to generate **Figures 7B**, **Figure 7—figure supplement 1B and 7E**, $P_{max} = 0.04 A_u/\mu m$.

$$\frac{\partial PIN_c}{\partial t} = \frac{-1}{R_c} \sum_{n \in N(c)} l_n \left( \left( \rho_{PIN} + \tau A_n^* \right) PIN_c - \mu_{PIN} PIN_n \right) \tag{2}$$

Where $PIN_C$ is the concentration of PIN1 in the focal cytoplasmic compartment and $PIN_n$ is the concentration of PIN1 in the membrane compartment $n$, in the neighbourhood of the cell c ($N(c)$). $R_c$ is the area of the cytoplasmic compartment and $l_n$ is the length of the $n$th membrane compartment. $A_n^*$ is the concentration of A* in the membrane compartment n. $\rho_{PIN}$, $\tau$ and $\mu_{PIN}$ are as described above for *Equation 1*.

The export of auxin from a cytoplasmic compartment to a wall compartment depends on the concentration of PIN1 in the intervening membrane compartment. The equation presented by *Abley et al., 2013*, describing the rate of change of auxin (referred to as mediator (M) in *Abley et al., 2013*), in cytoplasmic compartment i becomes:

$$\frac{dAux_i}{dt} = \rho_{Aux} - \mu_{Aux} Aux_i + \frac{1}{R_c} l_w \sum_{n \in N(c)} \left( v_{in} Aux_w - v_{out} Aux_i - \psi_{PIN} PIN_n Aux_i \right) \tag{3}$$

Where $Aux_i$ is the concentration of auxin in the cytoplasmic compartment i, $Aux_w$ is the concentration of auxin in the wall compartment neighbouring the membrane compartment $n$ in the neighbourhood of the cell c ($N(c)$), $\rho Aux$ is the production rate of auxin with units of $A_u/\mu m^2.s$, $R_c$ is the area of the cytoplasmic compartment, and $\mu_{Aux}$ is the degradation rate of auxin with units of /s. $PIN_n$ is the concentration of PIN1 in the $n$th membrane compartment of the cell. $v_{in}$ is the background

permeation rate of auxin into the cytoplasm from the wall with units of μm/s and $v_{out}$ is the background permeation rate of auxin into the wall from the cytoplasm with units of μm/s. $l_w$ is the length of the cell wall compartment into/out of which auxin flux is occurring and $R_c$ is the area of the cytoplasmic compartment. $\psi_{PIN}$ is the rate of PIN1-dependent active efflux of auxin from the cytoplasm into the wall with units of μm$^2$/ A$_u$.s.

The equation describing the rate of change of auxin in wall compartments becomes:

$$\frac{\partial Aux_w}{dt} = -\frac{1}{R_w} l_w \sum_{n \in N(w)} \left( v_{in}Aux_w - v_{out}Aux_c - \psi_{PIN}PIN_nAux_c \right) + D_{Aux}\nabla^2 Aux_w \tag{4}$$

where $R_w$ is the area of the wall compartment, $l_w$ is the length of the wall compartment, $Aux_w$ is the concentration of auxin in the wall compartment, $A_n$* is the concentration of A* in the membrane compartment $n$ in the neighbourhood of the given wall compartment $w$ ($N(w)$) and $Aux_c$ is the concentration of auxin in the cytoplasm of the same cell as the membrane compartment $n$. $\psi_{PIN}$ is as described above and $PIN_n$ is the concentration of PIN1 in the membrane neighbor, n, in the neighbourhood of the wall compartment w (($N(w)$). $D_{Aux}$ is the diffusion constant for auxin within the cell wall with units of μm$^2$/s (this constant relates to lateral diffusion between wall compartments as it is assumed that the concentration of auxin is uniform across the thickness of the wall).

In the simulation used to generate *Figure 10—figure supplement 1*, we introduce D6 kinase-like activity into the model by assuming that only a fraction of PINs are active and able to export auxin, and that this fraction depends on the local concentration of A* in the membrane. A* therefore represents D6 kinase-like activity. In this scenario, the term describing PIN1-mediated auxin efflux from the cell to the wall in *Equations 3 and 4* is modified to become:

$$\frac{A_n^{*h}}{A_n^{*h} + c^h} \Psi_{PIN}PIN_nAux_c \tag{5}$$

Where $A$*$_n$ is the concentration of A* in the same membrane compartment as PIN, $h$ is the Hill coefficient (here a value of 2 was used, giving a sigmoidal function), and $c$ is the concentration of A* at which half of the PINs are activated. $\psi_{PIN}$, $PIN_n$ and $Aux_c$ are as described above.

In all simulations of the indirect coupling model, there is no flux of auxin across the boundaries of the tissue.

In the simulation used to generate *Figure 10C and D* (positioning of cells with elevated auxin import and removal), in the first 6500 s of the simulation, the production rate of auxin, $\rho_{Aux}$, in all cells in the proximal half of the array (except those in the proximal-most row) is $1.5 \times 10^{-4}$ A$_u$/ μm$^2$.s (in the proximal most row $\rho_{Aux} = 2 \times 10^{-4}$A$_u$/μm$^2$.s). Then, between 6500 and 10 000 s of the simulation, the auxin production rate of all cells in the proximal half of the array (including those in the proximal-most file) is increased at every time step of the simulation:

$$\frac{d\rho_{Aux}}{dt} = \beta \tag{6}$$

Where $\rho_{Aux}$ is the auxin production rate of a given cell and β is a constant describing the rate of increase of the auxin production rate, with units of A$_u$/ μm$^2$.s$^2$ (β = $5 \times 10^{-6}$ A$_u$/ μm$^2$.s$^2$). As a consequence of this increase in the auxin production rate, after 10 000 s of the simulation, in the proximal-most file of cells, $\rho_{Aux} = 3.75 \times 10^{-4}$ A$_u$/ μm$^2$.s, and in the other cells in the proximal-most half of the leaf, $\rho_{Aux} = 3.25 \times 10^{-4}$ A$_u$/ μm$^2$.s.

After 6 500 s of the simulation, if the auxin concentration of a cell exceeds a threshold, T$_{import}$, the inwards permeability of the cell to auxin is increased 50-fold ($v_{in}$ = 37.5 μm /s) and the auxin degradation of the cell is increased 10-fold so that μ$_{Aux}$ = 0.1 /s. T$_{import}$ = 0.0285 A$_u$/ μm$^2$.

In the simulation used to generate *Figure 10C and D*, a noise term is added to the concentrations of auxin every 0.05 s of the simulation, according to the following equation:

$$Aux_n = Aux + \left( \theta_{Aux} * \sqrt{Aux} \right) \tag{7}$$

where $Aux_n$ is the concentration of auxin in a given cytoplasmic compartment after the addition of noise with units of A$_u$/μm$^2$, Aux is the concentration of auxin in that compartment before the addition of noise (with the same units), $\theta_{Aux}$ is a random number drawn from a normal distribution, with

mean 0, and standard deviation $5 \times 10^{-4}$. Noise is added to the concentration of auxin in proportion to the square root of the auxin concentration.

## Flux based model

The flux-based model was implemented using a similar discretisation of cells as in previous implementations (**Rolland-Lagan and Prusinkiewicz, 2005**; **Stoma et al., 2008**). Each cell is represented by a single central cytoplasmic compartment surrounded by six peripheral compartments representing cell edges (one for each cell-cell interface). There is no representation of the cell wall. Each edge compartment of a cell is connected to the cytoplasmic compartment of the same cell and to the juxtaposed edge compartment of the neighbouring cell, unless the vertex is at a border of the tissue.

The model is implemented based on previously described implementations (**Rolland-Lagan and Prusinkiewicz, 2005**; **Stoma et al., 2008**). At the beginning of all simulations, the system is initialised with auxin in cytoplasmic compartments and PIN1 at edge compartments.

The system may be initialised with noisy auxin concentrations in each cytoplasmic compartment, i:

$$Aux_i(t=0) = c_A(1 + \theta_{Aux}) \tag{8a}$$

$$\theta_{Aux} \in [-\varepsilon_{Aux}, \varepsilon_{Aux}] \tag{8b}$$

where $Aux_i(t = 0)$ is the initial concentration of auxin in a given cytoplasmic compartment, i, with units of $A_u/\mu m^2$ and $c_A$ is the default initial concentration of auxin at an edge compartment. $\theta_{Aux}$ is a random number uniformly distributed between an upper and lower limit, $\varepsilon_{Aux}$.

The system may also be initialised with noisy PIN1 concentrations at each cell edge:

$$PIN_{edge}(t=0) = c_{PINedge}(1 + \theta_{PIN}) \tag{9a}$$

$$\theta_{PIN} \in [-\varepsilon_{PIN}, \varepsilon_{PIN}] \tag{9b}$$

where $PIN_{edge}(t = 0)$ is the initial concentration of PIN1 at a given edge compartment, with units of $A_u/\mu m$ and $c_{PINedge}$ is the default initial concentration of PIN1 at an edge compartment. $\theta_{PIN}$ is a random number uniformly distributed between an upper and lower limit, $\varepsilon_{PIN}$.

PIN1 is recruited to a cell edge depending on the rate of auxin efflux across that edge, and PIN1 is removed from the edge at a background rate. The equation describing the rate of change of PIN1 concentration for a given cell edge, between cells i and j, is

$$\frac{dPIN_{ij}}{dt} = \begin{cases} \alpha\phi_{i\to j} - \gamma PIN_{ij} & if\, \phi_{i\to j} \geq 0 \\ -\gamma PIN_{ij} & if\, \phi_{i\to j} < 0 \end{cases} \tag{10}$$

Where $PIN_{ij}$ is the concentration of PIN1 in cell i, at the cell edge between cells i and j, $\alpha$ is a dimensionless constant determining the extent to which flux promotes PIN1 allocation to the membrane and $\Phi_{i\to j}$ is the auxin flux across the interface between cells i and j, with units of $A_u/\mu m.s$. Outgoing fluxes, from cell i, to cell j are considered to be positive, and incoming fluxes are considered to be negative. $\gamma$ is the unbinding rate of PIN1 from the cell edge, with units of /s.

As previously described (**Rolland-Lagan and Prusinkiewicz, 2005**), we assume that once the concentration of PIN1 at the cell edge reaches a threshold value, $P_{max}$, PIN1 can no-longer be allocated to the edge. This is equivalent to assuming that there is a maximum density of PIN1 proteins which may be present in the plasma membrane.

In all versions of the model, the flux across a given cell edge, from cell i to cell j, at a given time point depends on passive permeability of auxin between cells and PIN-mediated auxin export. In the simulations used to generate **Figures 9B**, **10A,B** and **16A,B**, we assume that auxin may also be actively imported into cells, and that the rate of import depends on the auxin import rate of a given cell, and the auxin concentration in its neighbour. The flux across a given cell edge, from cell i to cell j, at a given time point is therefore given by

$$\Phi_{i\to j} = D(A_i - A_j) + T(PIN_{ij}A_i - PIN_{ji}A_j) + I_jA_i - I_iA_j \tag{11}$$

Where $\Phi_{i\to j}$ is the flux across a given cell-cell interface, from cell i to cell j, $D$ is a constant describing

the passive permeation rate of auxin, with units of μm / s, $A_i$ is the concentration of auxin in cell i, $A_j$ is the concentration of auxin in cell j, $PIN_{ij}$ is the concentration of PIN1 in cell i, at the cell edge between cells i and j, and $PIN_{ji}$ is the concentration of PIN1 in cell j, at the cell edge between cell j and i. $T$ is a constant describing the rate of PIN-mediated auxin transport, with units of $\mu m^2/ A_u.s$. $I_j$ and $I_i$ are parameters describing the auxin import rates of cells j and i respectively, with units of μm / s.

The rate of change of auxin concentration in the cytoplasm of a given cell, i, depends on the production and degradation rates of auxin, and its flux into adjacent cells and is given by:

$$\frac{dA_i}{dt} = \rho - \mu A_i - \frac{1}{R_i} \sum_{ij \in N(i)} \left( \Phi_{i \to j} \right) l_{ij} \tag{12}$$

Where $\rho$ is the production rate of auxin, with units of $A_u / \mu m^2.s$, μ is the degradation rate of auxin, with units of /s, $A_i$ is the concentration of auxin in the cytoplasm of cell i and $R_i$ is the area of the cell i. $\Phi_{i \to j}$ is the flux of auxin across the interface between cells i and j, in the neighbourhood of the cell i (($N(i)$) and $l_{ij}$ is the length of the cell edge at the interface between cells i and j.

In the simulations used to generate *Figure 7B,E* and *Figure 7—figure supplement 1B*, there is no flux of auxin across the boundaries of the tissue. To minimise boundary effects, in the simulations used to generate *Figures 9A,B*, *10A,B* and *16A,B*, the left and right boundaries of the tissue are connected so auxin flux can occur between them. All changes in concentrations were solved numerically using an explicit Euler integration method.

In all simulations where the row of cells at the base of the tissue has an elevated rate of auxin production (orange cells at the base in *Figures 9A,B*, *10A,B* and *16 A,B*), the auxin production rate in these cells = 0.0008 $A_u /\mu m^2.s$. In *Figure 16A,B*, in the band of three cells with elevated auxin production, the auxin production rate = 0.032 $A_u /\mu m^2.s$. In the simulation used to generate *Figure 9A*, in the distal row of cells, the auxin degradation rate = 0.5/s. In the simulations used to generate *Figures 9B*, *10A,B* and *16 A, B*, in the distal most row of cells with elevated auxin import and removal, I = 300 μm / s and $\mu$ = 0.04/s. In the simulation used to generate *Figure 10B*, in the induced cells with elevated auxin import and removal, I = 30 μm / s and $\mu$ = 0.5 /s. In the simulation used to generate *Figure 16B*, I = 100 μm / s and $\mu$ = 0.8 /s in the single cell with elevated auxin import and removal.

In the simulation used to generate *Figure 10A and B* (positioning of cells with elevated auxin import and removal), in the first 165 s of the simulation, the production rate of auxin, $\rho_{Aux,}$ in the proximal-most file of cells, = 0.0008 $A_u/ \mu m^2.s$. In this first phase of the simulation, the production rate of auxin in all other cells in the proximal half of the array is 0.0005$A_u/ \mu m^2.s$. Then, between 165 and 200 s of the simulation, the auxin production rate of all cells in the proximal half of the array (including those in the proximal-most file) is increased at every time step of the simulation according to *Equation 6*. Here, β = 0.0001 $A_u/ \mu m^2.s^2$. As a consequence of this increase in the auxin production rate, after 200 s of the simulation, in the proximal-most file of cells, ρ = 0.0043 $A_u/ \mu m^2.s$, and in the other cells in the proximal-most half of the leaf, ρ = 0.004 $A_u/ \mu m^2.s$.

After 165 s of the simulation, if the auxin concentration of a cell exceeds a threshold, $T_{import}$, the auxin import of the cell is increased so that I = 30 μm / s and the auxin degradation rate is increased so that $\mu$ = 0.5 /s. $T_{import}$ = 0.09 $Au/ \mu m^2$ .

After 20 s of the simulation, noise is added to the concentrations of auxin every 0.01 s of the simulation, according to *Equation 7*. In this case, $\theta_{Aux}$ is a random number drawn from a normal distribution, with mean 0, and standard deviation $5 \times 10^{-3}$.

## Up-the-gradient model

The up-the-gradient model is implemented based on the description by *Bilsborough et al., 2011*. The tissue is represented as described for the flux-based model. The system is initialised with auxin in cytoplasmic compartments and PIN1 at edge compartments. In some simulations, the system is initialised with noisy PIN1 concentrations at each cell edge, as described by *Equations 9a,b*, or noisy auxin concentrations in each cell, as described by *Equations 8a,b*.

PIN1 only exists in cell edge compartments and each cell has a fixed total concentration of PIN1 available to its cell edges. This total concentration of PIN1 is distributed between edge compartments at each time step according to an exponential function of the auxin concentrations in

neighbouring cells. The equation describing the concentration of PIN1 in a given edge compartment in cell i, at the interface between cell i and j, at a given time step is:

$$PIN_{ij} = PIN_i \frac{b^{A_j}}{\sum_k b^{A_k}} \tag{13}$$

Where $PIN_{ij}$ is the concentration of PIN1 at the cell edge between cells i and j, in cell i. $PIN_i$ is the total PIN1 concentration available to be divided between all the edges of cell i (with units of $A_u$/μm), $A_j$ is the concentration of auxin in the neighbour j and $A_k$ is the auxin concentration in the neighbour k of cell i. The exponentiation base, $b$, controls the extent to which the auxin distribution in neighbouring cells influences PIN1 protein distribution at cell edges.

Previous simulations have differed in the implementation of auxin production. Some assume that intracellular auxin feeds back to inhibit its own production (*Smith et al., 2006*; *Bayer et al., 2009*; *Bilsborough et al., 2011*; *Feugier et al., 2005*), whilst other assume that auxin production occurs independently of intracellular auxin concentration (*Rolland-Lagan and Prusinkiewicz, 2005*; *Jonsson et al., 2006*; *Stoma et al., 2008*). We use the latter assumption in all models.

The rate of change of auxin concentration for a given cytoplasmic compartment depends on the rates of auxin production and degradation, and the rates of auxin flux between the given cell and its neighbours. This is described by the following equation:

$$\frac{dA_i}{dt} = \rho - \mu A_i - \frac{1}{R_i} \sum_{ij \in N(i)} \left( \Phi_{i \rightarrow j} \right) l_{ij} \tag{14}$$

Where $A_i$ is the auxin concentration of cell i, $\rho$ is the auxin production rate, with units of $A_u$ /μm$^2$.s, μ is the auxin degradation rate, with units of / s and $R_i$ is the area of the cell i, with units of μm$^2$. $\Phi_{i \rightarrow j}$ is the auxin flux (with units of $A_u$ / μm.s) out of cell i, across the interface between cells i and j, in the neighbourhood of cell i (($N(i)$). $l_{ij}$ is the length of the cell edge at the interface between cells i and j, with units of μm.

The flux of auxin between neighbouring cells is calculated as described by *Equation 11*. Changes in auxin concentrations were solved numerically using an explicit Euler integration method.

As for the flux-based model, in the simulations used to generate *Figure 7A and D* and *Figure 7— figure supplement 1A*, there is no flux of auxin across the boundaries of the tissue. To minimise boundary effects, in the simulations used to generate *Figures 8A,B*, *11H, 14D* and *16 E,F*, the left and right boundaries of the tissue are connected so auxin flux can occur between them.

In simulations used to generate *Figure 8A,B* and *Figure 16E,F*, where the distal-most row of cells has an elevated initial auxin concentration (row of orange cells in *Figures 8A,B*), the initial auxin concentration in these cells is 0.5 $A_u$ /μm$^2$. In all simulations where the proximal-most row of cells has a low initial auxin concentration and an elevated rate of auxin degradation (blue cells in bottom row in *Figures 8A,B*, *11H, 16E,F*), the initial auxin concentration in these cells is 0 $A_u$ /μm$^2$ and their auxin degradation rate = 0.05/s.

In the simulation used to generate *Figure 14D*, the initial auxin concentration in the distal-most row of cells = 0.5 $A_u$ /μm$^2$, whilst in all other cells, the initial auxin concentration is as stated in *Table 3*. In the distal most row of cells, the rates of auxin import and degradation are elevated (I = 20 μm / s and μ = 0.01 /s). In the proximal most file of cells, the auxin production rate is elevated so that $\rho$ = 0.001 $A_u$ /μm$^2$.s

In the simulation used to generate *Figure 11H*, in the distal most row of cells, auxin import and degradation are elevated, as described for *Figure 14D*. When the auxin concentration exceeds a threshold, $T_{import}$, in cells other than those at the proximal and distal boundaries, the rates of auxin import and degradation are increased to be equal to those in the distal most row of cells (I = 20 μm / s and μ = 0.01 /s). $T_{import}$ = 0.08 $A_u$ /μm$^2$.

In *Figure 16E,F*, in the band of three cells with elevated auxin production the auxin production rate = 0.04 $A_u$ /μm$^2$.s. In *Figure 16F*, in the cell with elevated auxin import and removal, I = 5 μm / s and μ = 0.03 /s.

**Table 3.** Parameter values used in up-the-gradient simulations.

| Symbol | description | unit | value |
|---|---|---|---|
| $\Delta t$ | numerical time step | s (seconds) | 0.01 |
| $R$ | area of cytoplasmic compartment | $\mu m^2$ | 260 |
| $l$ | length of cell edge compartment | $\mu m$ | 15 for short cells edges; 8.6 for the each of the two compartments of a long edge * |
| $C_A$ | default initial concentration of auxin in cytoplasmic compartments | $A_u$/ $\mu m^2$ | 0.01 (**Figure 7A,D**, **Figure 7—figure supplment 1A**.) 0.005 (other Figs.) |
| $C_{PIN}$ | default initial concentration of PIN1 at cell edge compartments | $A_u$ /$\mu m$ | 0.1[†] |
| $\varepsilon_{PIN}$ | limit for noise addition during initialisation of PIN1 concentrations at cell edges | dimensionless | 0.025[†] |
| $\varepsilon_{Aux}$ | limit for noise addition during initialisation of Auxin concentrations | dimensionless | 0.025[†] |
| $T$ | PIN-dependent active auxin efflux rate | $\mu m^2/A_u$.s | 80 |
| $\rho$ | production rate of Auxin | $A_u$ /$\mu m^2$.s | 0.0003[‡] |
| $\mu$ | degradation rate of Auxin | /s | 0.005 |
| $PIN_i$ | Total amount of PIN1 in a cell available for binding to edge compartments | $A_u$ /$\mu m$ | 0.1 |
| $D$ | Passive permeation rate of auxin | $\mu m/s$ | 10 |
| $b$ | Exponentiation base for PIN1 allocation to the membrane | dimensionless | 6 |
| $I$ | Auxin import rate | $\mu m$ / s | 0 |

* For cells with hexagonal geometries, each cell edge has a length of 10 $\mu m$.

[†] Values given for the initial concentration of PIN1 at cell edge compartments, and the limit for noise addition to this concentration, are those used in simulations used to generate **Figure 7A** and **Figure 7—figure supplement 1A**. In all other simulations, $C_{PIN}$ and $\varepsilon_{PIN}$ = 0. The value given for $\varepsilon_{Aux}$ applies only to **Figure 7D**, in all other simulations, $\varepsilon_{Aux}$ = 0.

[‡] Value given for the production rate of auxin applies to the simulations used to generate **Figures 7D**, **8A, B**, **11H, 14D** and **16E,F**. In the simulation used to generate **Figure 7A** and **Figure 7—figure supplement 1A**, $\rho$ = 0.0025 $A_u$ /$\mu m^2$.s.

## Acknowledgements

We thank Youfa Cheng and Yunde Zhao for sharing the unpublished *YUC4::GFP* line, Patrick Laufs for providing the unpublished *CUC2::RFP* line and John Bowman for providing *kan1kan2+/-* seeds and for helpful discussions. Thanks to three anonymous reviewers for their help with improving the manuscript. Thanks to all to members of the Coen lab for discussions and to Xana Rebocho Christopher Whitewoods and Annis Richardson for comments on the manuscript. Additionally, special thanks to Xana Rebocho for advice on experimental work and to Samantha Fox for confirming genotypes of *kan1kan2aux/lax* mutants. Thanks to Przemysław Prusinkiewicz, Mik Cieslak, Adam Runions and Pau Formosa Jordan for comments on the manuscript and discussions on the relationships between models. Thanks to Pierre Barbier de Reuille for development of the VVe modelling environment and to Pierre Barbier de Reuille and Pascal Ferraro for preparing L-studio/VVe for distribution. We also thank Grant Calder for confocal microscope support. Thanks to the John Innes Horticultural Services team for help with plant care. Many thanks to Andrew Bangham for his support and encouragement and for many useful discussions about this work.

## Additional information

### Funding

| Funder | Grant reference number | Author |
|---|---|---|
| Biotechnology and Biological Sciences Research Council | BB/F005997/1 SABR grant | Enrico Coen |
| Gatsby Charitable Foundation | PhD studentship | Katie Abley |

| Biotechnology and Biological Sciences Research Council | BB/L008920/1 | Enrico Coen |
|---|---|---|

The funders had no role in study design, data collection and interpretation, or the decision to submit the work for publication.

### Author contributions

KA, Conception and design, Acquisition of data, Analysis and interpretation of data, Drafting or revising the article; SS-G, Acquisition of data, Contributed unpublished essential data or reagents; AFMM, Analysis and interpretation of data, Drafting or revising the article; EC, Conception and design, Analysis and interpretation of data, Drafting or revising the article

### Author ORCIDs

Katie Abley, http://orcid.org/0000-0001-5524-6786
Enrico Coen, http://orcid.org/0000-0001-8454-8767

# Additional files

### Supplementary files

• Source code 1. Code used to generate *Figure 7A*.

• Source code 2. Code used to generate *Figure 7B*.

• Source code 3. Code used to generate *Figure 7C*.

• Source code 4. Code used to generate *Figure 7D*.

• Source code 5. Code used to generate *Figure 7E*.

• Source code 6. Code used to generate *Figure 7F*.

• Source code 7. Code used to generate *Figure 8A*.

• Source code 8. Code used to generate *Figure 8B*.

• Source code 9. Code used to generate *Figure 9A*.

• Source code 10. Code used to generate *Figure 9B*.

• Source code 11. Code used to generate *Figure 9C*.

• Source code 12. Code used to generate *Figure 9D*.

• Source code 13. Code used to generate *Figure 10A and B*.

• Source code 14. Code used to generate *Figure 10C and D*.

• Source code 15. Code used to generate *Figure 11H*.

• Source code 16. Code used to generate *Figure 14D*.

• Source code 17. Code used to generate *Figure 16A*.

• Source code 18. Code used to generate *Figure 16B*.

• Source code 19. Code used to generate *Figure 16C*.

• Source code 20. Code used to generate *Figure 16D*.

• Source code 21. Code used to generate *Figure 16E*.

• Source code 22. Code used to generate *Figure 16F*.

• Supplementary file 1. Supplementary model information. Instructions on how to run models and explanation of the code for each model.

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
