## [Decision Letter]

[Editors’ note: a previous version of this study was rejected after peer review, but the authors submitted for reconsideration. The first decision letter after peer review is shown below.]

Thank you for submitting your work entitled "Formation of Polarity Convergences underlying Shoot Outgrowths" for consideration by *eLife*. Your article has been evaluated by Detlef Weigel as the Senior editor and three reviewers, one of whom, Joanne Chory, is a member of our Board of Reviewing Editors.

Our decision has been reached after consultation between the reviewers. Based on these discussions and the individual reviews below, we regret to inform you that your current study will not be considered further for publication in *eLife*. However, we want to encourage you to prepare a substantially revised manuscript and consider submission of a new version to *eLife*.

We are in principle very interested in your model because it brings new discussion to the field, taking into account the parameters of auxin import and biosynthesis. However, we felt a few more experiments would greatly strengthen your arguments. Specifically, should you decide to resubmit, please consider:

1) A more complete documentation for different PINs and *LAX*s in (*kan*) leaves, using state-of-the-art markers;

2) Testing whether a D6 kinase type activity would improve a model (see point 4 below, and comments of reviewers 1 and 2);

3) Adding more discussion about what your model can and cannot accomplish.

In particular, we would like to see increased discussion of the following specific points:

1) The authors build their work on a previous paper (Abley 2013), in which they introduced the indirect coupling model. In this context, it is laudable that the authors clarify that their earlier conclusion that both the convergence point and the flux-based model could not create single cell polarity was erroneous. This is because previously they did not allow stochastic noise in the PIN distribution, which is however the necessary starting point for all models now, just as in a classic self-organizing reaction-diffusion model. Is this correct? If so, or any other reason, this should be explicitly stated.

2) In several places, the authors discussed auxin removal and auxin degradation. It is not clear whether the two terms mean the same thing. Does auxin transport out of a cell represent auxin removal? Auxin degradation means that auxin is destroyed, not transported out? A clear definition of the terms will be helpful.

3) To explain the phenotype of wild type with respect to the convergence point model, the authors propose at one point that "PIN polarity becomes fixed early in development, preventing polarity reorientation". How would that work and is there any experimental evidence for this in the literature?

4) The characterization of regulatory mechanisms for PIN activity in recent years is not considered in the model, and maybe understandably, because it is technically difficult to assess. Nevertheless, one can no longer assume that where there's a PIN, there's auxin efflux. How robust is the model if, for instance, a parameter for non-linear PIN activation (by D6 kinases, themselves polar), for example as a function of polarity, would be introduced?

5) What are the limitations of using the mixed outgrowths of the *kanadi* mutants?

Reviewer #1:

In this paper, Abley et al. examine various models for how shoot apical meristems of plants generate auxin maxima, which become the site of new outgrowths. The most widely accepted model, the "up-the-gradient" model, is also the most parsimonious explanation as it spontaneously generates convergent PIN transporter alignments. Based on recent literature, suggesting that regulated auxin import and new auxin biosynthesis must also be involved, the authors looked at the ability of current models to predict sites of new leaf outgrowths. In a combination of methodologies that include both experiments and computer simulations, the authors challenge the simple model and conclude that less parsimonious models, ("flux model" or "indirect cell-cell coupling" model), make predictions that better explain the data.

This is a very nicely written paper that will initiate a new dialog in the field of auxin control of development-one of the deepest studied areas of plant biology. The time-lapse images are stunning, and I generally agree with the interpretation (with the exclusion of Figure 11B3). The Meyerowitz lab's "up-the gradient" model has dominated the discussion for 8 years, and I think it's time for us to consider some new ideas. Thus, I am in favor of publishing this paper. That said, some other new data has emerged in the past 8 years, in addition to the information on auxin import and auxin biosynthesis. As an example, PIN1 can localize to a side of a cell, but it is only an active transporter when it's phosphorylated by PID or one of its paralogs or one of the D6PKs. A second example is the use of *kan1 kan2* double mutant, which makes mixed tissue outgrowths ectopically that are best predicted by tandem alignment models. Why not just look at areas of serration? The authors imaged only epidermal tissue. Why not other layers?

Reviewer #2:

This paper is quite nice to read and very well explained, especially how the models work and how they relate to the data is very accessible to the non-specialist reader as compared to other studies in this field. In a nutshell, the authors perform a top-down modeling approach to test which of three alternative models for auxin convergence point formation fits the data best. The data set mostly comprises time course observations of pertinent markers, at the same time taking advantage of the unusual laminar outgrowth formation in *kanadi* mutants. From the simulations and their match with spatio-temporal maker expression, the authors conclude that the two "tandem alignment" models fit the data better than another model that has been proposed to explain convergence points. Specifically, the "indirect coupling" tandem alignment model proposed by the authors previously appears to fit the observations best. I am not a modeler and cannot judge the technical dimension of the simulations, but trust the authors and other reviewers are competent in this field. Nevertheless, I have some questions and suggestions related to the modeling, as well as the experimental support and verification.

1) The authors build their work on a previous paper (Abley 2013), in which they introduced the indirect coupling model. In this context, it is laudable that the authors clarify that their earlier conclusion that both the convergence point and the flux-based model could not create single cell polarity was erroneous. As far as I understand, this is because previously they did not allow stochastic noise in the PIN distribution, which is however the necessary starting point for all models now, just as in a classic self-organizing reaction-diffusion model. Is this correct? If so, or any other reason, this should be explicitly stated.

2) The indirect coupling model requires two substances that establish an intracellular polarity through the interplay of differentially mobile cytosolic and membrane/apoplastic isoforms that interact with, and determine polarity of the auxin transport machinery. I suspect that the appeal of this model was linked to the discovery of the supposed interaction between Rho GTPases on one side and ABP1 on the other, including the auxin feedback on ABP1 mobility/activity as outlined in Wabnik et al. 2010, Mol Sys Biol. With ABP1 now clearly defunct in its role as extracellular auxin receptor that feeds on PIN polarity by regulating endocytic recycling of PINs (Michalko et al. F1000 Research 2016), I wonder how realistic is this model still? In all consequence, one might say that maybe an even better model could be built if it's all free fantasy. In the absence of experimentally proven candidates for substances A and B at this point, the indirect coupling model loses much of its mojo, unfortunately.

3) This brings me to my maybe major concern as an experimentalist: what is the value of the model if it is too detached from the molecular experimental reality? For instance, the characterization of regulatory mechanisms for PIN activity in recent years is not considered in the model, and maybe understandably, because it is technically difficult to assess. Nevertheless, one can no longer assume that where there's a PIN, there's auxin efflux. Thus, I wonder how robust the model is, for instance if a parameter for non-linear PIN activation (by D6 kinases, themselves polar), for example as a function of polarity, would be introduced?

4) What I am concerned about is that the model is supported by (maybe unconscious) selectivity for experimental data, in the sense that data that fit are over-emphasized while data that don't are ignored. I do not imply that this is what the authors did here, but I would like them to take into account the following considerations:

5) The imaging/modeling is exclusively based on PIN1 abundance and polarity patterns as far as I can see. Are there any valid reasons for this? Are other PINs not at all expressed in leaves? And, specifically, in the *kan1 kan2* background (see below)?

6) A major point to get any of the models to work is the requirement for proximo-distally separated auxin sources and sinks. Sinks can be provided in different ways, like auxin drainage through developing vasculature, but an attractive feature of the best-fit model is that auxin importers would amplify sink strength. Apparently, both *AUX1* and *LAX1* would fit the bill. One idea here is that threshold auxin levels induce either or both of them, and I imagine that would be easy to test, and quantify experimentally.

7) Given the major role assigned to *AUX1* and *LAX1* in the model fit, the experimental data are a bit disappointing. The quadruple mutant clearly has a tangible reduction in "outgrowth success", but it is rather weak (from ~12 to ~8) overall. Weakening the suspected vascular drain by knocking out *LAX2* on top leads to a further reduction to ~5. I was wondering how these values exactly relate to the model? Are they within the models "plasticity" when these components are taken out? Also, is *AUX/LAX* polarity in the vasculature in line with the direction of drainage?

8) Regarding more experimental proof for this idea: I believe the authors could test the role of *AUX/LAX*s more directly. For example, does local induction of an *AUX/LAX* in an inducible system (maybe +/- auxin treatment) trigger laminar outgrowth? Or, does local auxin efflux inhibition after convergence point formation delay/prevent outgrowth?

9) Also, with respect to the contribution of auxin biosynthesis, could outgrowth frequency be modulated by its local pharmacological inhibition?

10) Finally, I am a little bit worried about the use of the *kan1 kan2* double mutant as a "model" for outgrowth formation. Given the genetic interaction of *KANADI*s with auxin response factor (e.g. Pekker et al. 2005 PC), and the finding that *KANADI*s are transcriptional repressors that target many auxin-related genes (notably, PIN-activating AGC kinases) (Huang et al. 2014 PC), could the observations in the *kan1 kan2* double mutants be deceiving? For instance, *LAX3* is a repressed *KAN1* target, therefore presumably up-regulated in *kan1 kan2*. This could create stochastic auxin "holes", which could then trigger outgrowth, and it appears possible that the authors' observations regarding PINs etc. follow, rather than precede this event.

11) The fact that *LAX3* has such a prominent role in reducing outgrowth frequency despite its supposed absence in the leaf could support this idea. I suggest the authors test experimentally at higher resolution whether *LAX3* is ectopically expressed in *kan1 kan2* leaves and associated with, and/or preceding outgrowth formation. The images provided in Figure 8—figure supplement 1C are difficult to interpret, maybe early *LAX3* expression is carried over during leaf organogenesis and auxin sinks are predefined at this point? Also, alternative markers (fluorescent markers) could reveal hidden expression patterns frequently missed by equivalent GUS markers.

12) If not, could the impact of the *lax3* mutation be explained by auxin accumulation in the leaf because of a lack of drainage to the stem? Maybe this could be tested experimentally.

13) To explain the discrepant phenotype of wild type with respect to the convergence point model, the authors propose at one point that "PIN polarity becomes fixed early in development, preventing polarity reorientation". How would that work and is there any experimental evidence for this in the literature?

Reviewer #3:

In this manuscript, the authors combined modeling, live-imaging, and mutant analyses to study the mechanisms underlying shoot outgrowth. Similar to previous studies, this paper also used PIN1 localization/polarity as a proxy for auxin concentrations/gradients. The interesting part of this paper is that they included the analysis of expression patterns of auxin influx carriers and auxin biosynthesis genes. Overall the paper is very well written and easy to follow. I am not an expert on modeling and cannot really judge the validity of the modeling parts. But I think that the paper can be more convincing if:

1) PIN1 polarity is analyzed in the aux lax mutants and *kan1 kan2 aux lax* mutants.

2) PIN1 polarity in *kan1 kan2 yuc1 yuc4*.

In several places, the authors discussed auxin removal and auxin degradation. It is not clear to me whether the two terms mean the same thing. Does auxin transport out of a cell represent auxin removal? Auxin degradation means that auxin is destroyed, not transported out? Clearly define the terms will be helpful.

---

## [Author Response]

[Editors’ note: the author responses to the first round of peer review follow.]

*We are in principle very interested in your model because it brings new discussion to the field, taking into account the parameters of auxin import and biosynthesis. However, we felt a few more experiments would greatly strengthen your arguments.*

Thank you for this suggestion. We have further tested models by carrying out additional experiments on the role of *CUC2*, as this gene plays a key role in outgrowth formation (subsection “*CUC2* is required for normal *kan1kan2* outgrowth development and PIN1 convergence formation”, first paragraph). The results show that, similar to leaf serrations, the development of *kan1kan2* outgrowths and their associated PIN1 convergence points requires *CUC2*. Moreover, consistent with tandem alignment models, the loss of outgrowths in *kan1 kan2cuc2* mutants correlates with loss of ectopic expression of *YUC1*, *YUC4* and *LAX1* and altered expression of *AUX1*. The comparison of this new data with model predictions further illustrates the value of combining modelling with experimental testing.

Specifically, should you decide to resubmit, please consider:

*1) A more complete documentation for different PINs and LAXs in (kan) leaves, using state-of-the-art markers;*

Our analysis focused on PIN1 as this is the predominant epidermally expressed polarized PIN in wild-type (Guenot et al., 2012). The results with PIN1 clearly demonstrate polarity reorientations and how these might relate to the proposed models. It is unclear what key new insights would be gained by introduction of additional markers. We hope the paper is acceptable without this additional work. We have added the following sentence:

“Here we focus on PIN1 since this is the predominant epidermally expressed PIN that shows polar intracellular distributions in wild-type leaves (Guenot et al., 2012).”

*2) Testing whether a D6 kinase type activity would improve a model (see point 4 below, and comments of reviewers 1 and 2);*

We have added the following to the Results section:

“D6 protein kinases are polarly localized in cells independently of PIN proteins and their phosphorylation of PIN proteins is required for PIN-mediated auxin efflux (Barbosa et al., 2014, Zourelidou et al., 2009). We added a representation of D6 kinase activity to the indirect coupling model, as cell polarity in this system does not depend on PIN activity, and found that this does notqualitatively affect model behavior (Figure 10—figure supplement 1).”

*3) Adding more discussion about what your model can and cannot accomplish.*

We have added the following clarification at the end of the Discussion, which points out that evaluation of models is not based only on what they can accomplish but on how they compare to other models:

“Our analysis illustrates the value of comparing multiple models when analysing experimental data rather than focusing on one in isolation. […] Other types of experiment will be needed to discriminate between these further levels of explanation.”

*In particular, we would like to see increased discussion of the following specific points:*

1) The authors build their work on a previous paper (Abley 2013), in which they introduced the indirect coupling model. In this context, it is laudable that the authors clarify that their earlier conclusion that both the convergence point and the flux-based model could not create single cell polarity was erroneous. This is because previously they did not allow stochastic noise in the PIN distribution, which is however the necessary starting point for all models now, just as in a classic self-organizing reaction-diffusion model. Is this correct? If so, or any other reason, this should be explicitly stated.

We thank the reviewer for pointing this ambiguity out. The reason for us overlooking intracellular partitioning for flux-based and up-the-gradient models in our previous paper was not because we didn’t allow for stochastic noise in PIN distribution as the referee suggests. In all our previous work we allowed for stochastic noise in both PIN and auxin distribution, which, as the referee correctly points out, is a necessary starting point for testing any model, and such a procedure therefore was applied in all current and previous simulations in all our work. The reason for our revision arose from the fact that no one had previously applied their models to simulate single polarisable cells surrounded by non-polarisable neighbours - all previous simulations considered cell arrays in which all cells are polarisable. When performing such simulations we found that all three models can generate polarity in an initially uniform field of auxin concentration. The polarity arises in the flux-based and up-the-gradient models because small fluctuations in PIN1 distribution create variations in auxin flux or concentrations that feed back to reinforce the polarity. We have now clarified the text in the subsection “PIN1 polarity models can be classified into two groups with different behaviours” to make our reasoning more explicit, and also pointed out a key difference between the up-the-gradient model and the other models. If auxin concentration in the medium surrounding a single polarisable cell remains fixed throughout the simulation (equivalent to voltage-clamping in neurophysiology), then the up-the-gradient model does not generate polarity (Figure 7—figure supplement 1). In contrast, both the flux-based and indirect coupling model present polarity even under such settings.

2) In several places, the authors discussed auxin removal and auxin degradation. It is not clear whether the two terms mean the same thing. Does auxin transport out of a cell represent auxin removal? Auxin degradation means that auxin is destroyed, not transported out? A clear definition of the terms will be helpful.

We thank the reviewers for pointing out this ambiguity. We have added the following:

“Net auxin removal from a region of tissue could be achieved through a decreased rate of auxin biosynthesis, an increased rate of transport away from the region, or an increased rate of auxin degradation or conjugation. Here net removal is achieved using elevated auxin degradation rates (Figure 8).”

We also now say:

“Here we simulate net auxin production in proximal regions with an increased auxin biosynthesis rate, although it could also be achieved with a reduced auxin degradation rate or increased auxin influx from (or decreased auxin efflux to) tissues beyond those represented in the simulation. Likewise, net removal from the leaf tip is simulated using elevated degradation, although it could also occur through reduction in biosynthesis, increase in conjugation, or via transport into underlying tissue layers (Bayer et al., 2009; Scarpella et al., 2006).”

3) To explain the phenotype of wild type with respect to the convergence point model, the authors propose at one point that "PIN polarity becomes fixed early in development, preventing polarity reorientation". How would that work and is there any experimental evidence for this in the literature?

We agree with the reviewers that it is unclear how fixing polarity would work and is a possible weakness of the convergent polarity model. We have made this clearer by changing the wording of this paragraph to read:

“The failure to form such a convergence centre in wild-type leaves could be explained by the PIN1 polarity pattern becoming fixed at early stages of development, preventing polarity reorientation. However, it is unclear how this could be achieved, although it has been proposed that absence of CUC2 activity may play a role in fixing polarity (Bilsborough et al., 2011).”

4) The characterization of regulatory mechanisms for PIN activity in recent years is not considered in the model, and maybe understandably, because it is technically difficult to assess. Nevertheless, one can no longer assume that where there's a PIN, there's auxin efflux. How robust is the model if, for instance, a parameter for non-linear PIN activation (by D6 kinases, themselves polar), for example as a function of polarity, would be introduced?

See response to editorial comments point 2 above.

5) What are the limitations of using the mixed outgrowths of the kanadi mutants?

All systems for studying outgrowths (primordia, serrations, *kanadi* outgrowths) have the limitation that they represent specific instances. Identifying features that are common to all of them allows general elements of outgrowth formation to be identified, irrespective of the particular system or context. To make this point clearer we have added the following text to the beginning of the Discussion:

“*kan1kan2* outgrowths arise ectopically as a consequence of reduced abaxial identity, and thus differ from leaf primordia and serrations. Despite these differences, we show that general elements of outgrowth formation apply to shoot outgrowths (*kan1kan2* outgrowths, primordia and serrations) regardless of their specific context.”

*Reviewer #1:*

*This is a very nicely written paper that will initiate a new dialog in the field of auxin control of development-one of the deepest studied areas of plant biology. The time-lapse images are stunning, and I generally agree with the interpretation (with the exclusion of Figure 11B3). The Meyerowitz lab's "up-the gradient" model has dominated the discussion for 8 years, and I think it's time for us to consider some new ideas. Thus, I am in favor of publishing this paper. That said, some other new data has emerged in the past 8 years, in addition to the information on auxin import and auxin biosynthesis. As an example, pin1 can localize to a side of a cell, but it is only an active transporter when it's phosphorylated by PID or one of its paralogs or one of the D6PKs.*

See response to editorial comments point 2 above.

*A second example is the use of kan1 kan2 double mutant, which makes mixed tissue outgrowths ectopically that are best predicted by tandem alignment models. Why not just look at areas of serration?*

See response to editorial specific point 5. We have also added the following sentence to the Introduction:

“Because of their emergence from the abaxial lamina, these outgrowths are more amenable to time- lapse imaging than serrations which are often obscured by neighbouring cotyledon tissue and curving of the leaf edge.”

The authors imaged only epidermal tissue. Why not other layers?

We focus on epidermal tissue as it can be more readily tracked through time-lapse imaging and because it displays convergence sites. Following other layers would indeed be very interesting as well but is beyond the scope of the current paper.

*Reviewer #2:*

*This paper is quite nice to read and very well explained, especially how the models work and how they relate to the data is very accessible to the non-specialist reader as compared to other studies in this field. In a nutshell, the authors perform a top-down modeling approach to test which of three alternative models for auxin convergence point formation fits the data best. The data set mostly comprises time course observations of pertinent markers, at the same time taking advantage of the unusual laminar outgrowth formation in kanadi mutants. From the simulations and their match with spatio-temporal maker expression, the authors conclude that the two "tandem alignment" models fit the data better than another model that has been proposed to explain convergence points. Specifically, the "indirect coupling" tandem alignment model proposed by the authors previously appears to fit the observations best. I am not a modeler and cannot judge the technical dimension of the simulations, but trust the authors and other reviewers are competent in this field. Nevertheless, I have some questions and suggestions related to the modeling, as well as the experimental support and verification.*

1) The authors build their work on a previous paper (Abley 2013), in which they introduced the indirect coupling model. In this context, it is laudable that the authors clarify that their earlier conclusion that both the convergence point and the flux-based model could not create single cell polarity was erroneous. As far as I understand, this is because previously they did not allow stochastic noise in the PIN distribution, which is however the necessary starting point for all models now, just as in a classic self-organizing reaction-diffusion model. Is this correct? If so, or any other reason, this should be explicitly stated.

See response to editorial specific point 1 above.

2) The indirect coupling model requires two substances that establish an intracellular polarity through the interplay of differentially mobile cytosolic and membrane/apoplastic isoforms that interact with, and determine polarity of the auxin transport machinery. I suspect that the appeal of this model was linked to the discovery of the supposed interaction between Rho GTPases on one side and ABP1 on the other, including the auxin feedback on ABP1 mobility/activity as outlined in Wabnik et al. 2010, Mol Sys Biol. With ABP1 now clearly defunct in its role as extracellular auxin receptor that feeds on PIN polarity by regulating endocytic recycling of PINs (Michalko et al. F1000 Research 2016), I wonder how realistic is this model still? In all consequence, one might say that maybe an even better model could be built if it's all free fantasy. In the absence of experimentally proven candidates for substances A and B at this point, the indirect coupling model loses much of its mojo, unfortunately.

The indirect coupling model is a hypothesis that arose from general considerations of how tissue cell polarity might be coordinated, irrespective of the molecular players (Abley et al). It is not true that any model can be built if all is fantasy because there are limitations on what can and cannot work in principle. Indeed, the comment of the referee highlights the danger with making models that are too narrowly based on particular experimental claims which may themselves be faulty. Indirect coupling depends on a mechanism for sensing extracellular auxin concentration and ABP1 initially seemed like a reasonable candidate. We agree that recent data rules out ABP1 having this role, and like the other hypotheses (up-the-gradient, flux) the molecular players involved in sensing auxin levels or flux remain to be elucidated.

3) This brings me to my maybe major concern as an experimentalist: what is the value of the model if it is too detached from the molecular experimental reality? For instance, the characterization of regulatory mechanisms for PIN activity in recent years is not considered in the model, and maybe understandably, because it is technically difficult to assess. Nevertheless, one can no longer assume that where there's a PIN, there's auxin efflux. Thus, I wonder how robust the model is, for instance if a parameter for non-linear PIN activation (by D6 kinases, themselves polar), for example as a function of polarity, would be introduced?

See response to editorial comments point 2 above.

*4) What I am concerned about is that the model is supported by (maybe unconscious) selectivity for experimental data, in the sense that data that fit are over-emphasized while data that don't are ignored. I do not imply that this is what the authors did here, but I would like them to take into account the following considerations:*

5) The imaging/modeling is exclusively based on PIN1 abundance and polarity patterns as far as I can see. Are there any valid reasons for this? Are other PINs not at all expressed in leaves? And, specifically, in the kan1 kan2 background (see below)?

See response to editorial comments point 1 above.

6) A major point to get any of the models to work is the requirement for proximo-distally separated auxin sources and sinks. Sinks can be provided in different ways, like auxin drainage through developing vasculature, but an attractive feature of the best-fit model is that auxin importers would amplify sink strength. Apparently, both AUX1 and LAX1 would fit the bill. One idea here is that threshold auxin levels induce either or both of them, and I imagine that would be easy to test, and quantify experimentally.

Models should make testable predictions and we agree that this is a testable prediction of our model. However, as we already test several predictions, including the incorporation of additional data on the role of *CUC2*, we hope that further tests such as this could be the subject of further studies and would not be required for acceptance of this one.

7) Given the major role assigned to AUX1 and LAX1 in the model fit, the experimental data are a bit disappointing. The quadruple mutant clearly has a tangible reduction in "outgrowth success", but it is rather weak (from ~12 to ~8) overall. Weakening the suspected vascular drain by knocking out LAX2 on top leads to a further reduction to ~5. I was wondering how these values exactly relate to the model? Are they within the models "plasticity" when these components are taken out? Also, is AUX/LAX polarity in the vasculature in line with the direction of drainage?

We could tweak parameters to allow the contribution of individual importers to be fitted to the data. However, our approach is not to fit the model to particular data after it has been obtained but to test specific predictions of the model. The tandem coupling models predict that there should be a mechanism for keeping extracellular auxin low at convergence centres, such as auxin import. We tested this prediction experimentally and obtained supporting data based on the expression pattern and mutants. The fact that we see a marked reduction as predicted is striking, and given the potential for redundancy of gene activity and mechanisms, it is not surprising that outgrowths were not completely eliminated. We have not looked at polarity of *LAX1* or *LAX2* in vascular tissue but the model does not have specific requirements for this.

8) Regarding more experimental proof for this idea: I believe the authors could test the role of AUX/LAXs more directly. For example, does local induction of an AUX/LAX in an inducible system (maybe +/- auxin treatment) trigger laminar outgrowth? Or, does local auxin efflux inhibition after convergence point formation delay/prevent outgrowth?

See response to Reviewer 2 point 6 above.

9) Also, with respect to the contribution of auxin biosynthesis, could outgrowth frequency be modulated by its local pharmacological inhibition?

The loss of outgrowths in *kan1kan2yuc1yuc2yuc4* mutants (Wang et al., 2011) shows the role of YUC genes more cleanly than using pharmacological inhibition. We mention this requirement of YUC genes in the Results section:

“*A. thaliana* has six YUCCA genes, three of which *(YUCCA1 (YUC1), YUCCA2 (YUC2)* and *YUCCA4 (YUC4))* are expressed in the leaf and are redundantly required for *kan1kan2* outgrowth development (Cheng et al., 2007; Cheng et al., 2006; Wang et al., 2011).

10) Finally, I am a little bit worried about the use of the kan1 kan2 double mutant as a "model" for outgrowth formation. Given the genetic interaction of KANADIs with auxin response factor (e.g. Pekker et al. 2005 PC), and the finding that KANADIs are transcriptional repressors that target many auxin-related genes (notably, PIN-activating AGC kinases) (Huang et al. 2014 PC), could the observations in the kan1 kan2 double mutants be deceiving? For instance, LAX3 is a repressed KAN1 target, therefore presumably up-regulated in kan1 kan2. This could create stochastic auxin "holes", which could then trigger outgrowth, and it appears possible that the authors' observations regarding PINs etc. follow, rather than precede this event.

Pekker et al., 2005 show that ARF3 and ARF4 expression is only slightly altered in *kan1kan2* mutant leaves where ARF3 expression appears ectopically in *kan1kan2* outgrowths. However, since ARF3 is also expressed in emerging leaf primordia, this supports the similarity of *kan1kan2* outgrowths and leaf primordia. We thank the reviewer for the suggestion of stochastic auxin holes arising as a consequence of altered *LAX* expression. This of course begs the question of how such holes would be established and would require some sort of reinforcing mechanism in itself. From the data in Huang et al., 2014, it appears that *LAX3* expression is actually promoted by induction of KAN1-GR activity (see supplementary Figure 5 and Table 2), and thus its expression would be expected to be reduced in *kan1kan2* mutants. There is some evidence that *LAX1* expression is down-regulated by KAN1 in Huang et al., 2014. However, here we show that ectopic *LAX1* expression is lost in *kan1kan2cuc2* leaves, suggesting that upregulation of *LAX1* in *kan1kan2* leaves occurs via *CUC2*. This is consistent with the tandem alignment models we present but less easily reconciled with a stochastic auxin hole hypothesis.

11) The fact that LAX3 has such a prominent role in reducing outgrowth frequency despite its supposed absence in the leaf could support this idea. I suggest the authors test experimentally at higher resolution whether LAX3 is ectopically expressed in kan1 kan2 leaves and associated with, and/or preceding outgrowth formation. The images provided in Figure 8—figure supplement 1C are difficult to interpret, maybe early LAX3 expression is carried over during leaf organogenesis and auxin sinks are predefined at this point? Also, alternative markers (fluorescent markers) could reveal hidden expression patterns frequently missed by equivalent GUS markers.

We thank the reviewer for this suggestion. *LAX3* only has a prominent role in the *aux1 lax1 lax2* background. The specific contribution of *LAX3* in relation to other importers is not critical to our model (see response to Reviewer 2 point 7 above) and investigating its specific contribution could be the subject of a further study (see also response to Reviewer 2 point 6 above).

12) If not, could the impact of the lax3 mutation be explained by auxin accumulation in the leaf because of a lack of drainage to the stem? Maybe this could be tested experimentally.

See response to Reviewer 2 points 6 and 11 above.

13) To explain the discrepant phenotype of wild type with respect to the convergence point model, the authors propose at one point that "PIN polarity becomes fixed early in development, preventing polarity reorientation". How would that work and is there any experimental evidence for this in the literature?

See response to editorial specific point 3.

*Reviewer #3:*

*In this manuscript, the authors combined modeling, live-imaging, and mutant analyses to study the mechanisms underlying shoot outgrowth. Similar to previous studies, this paper also used PIN1 localization/polarity as a proxy for auxin concentrations/gradients. The interesting part of this paper is that they included the analysis of expression patterns of auxin influx carriers and auxin biosynthesis genes. Overall the paper is very well written and easy to follow. I am not an expert on modeling and cannot really judge the validity of the modeling parts. But I think that the paper can be more convincing if:*

*1) PIN1 polarity is analyzed in the aux lax mutants and kan1 kan2 aux lax mutants.*

2) PIN1 polarity in kan1 kan2 yuc1 yuc4.

We thank the referee for raising this point. We have addressed it by incorporating new data which shows that PIN1 convergences are absent in *kan1kan2cuc2* mutants which lose both ectopic *AUX/LAX* and YUC1, YUC4 expression.

*In several places, the authors discussed auxin removal and auxin degradation. It is not clear to me whether the two terms mean the same thing. Does auxin transport out of a cell represent auxin removal? Auxin degradation means that auxin is destroyed, not transported out? Clearly define the terms will be helpful.*

See response to editorial specific point 2.